# Stochastic Variance Reduced Primal Dual Algorithms for Empirical Composition Optimization

**Adithya M. Devraj**[*]   and   **Jianshu Chen**[†]

## Abstract

We consider a generic *empirical composition optimization* problem, where there are empirical averages present both outside and inside nonlinear loss functions. Such a problem is of interest in various machine learning applications, and cannot be directly solved by standard methods such as stochastic gradient descent. We take a novel approach to solving this problem by reformulating the original minimization objective into an equivalent min-max objective, which brings out all the empirical averages that are originally inside the nonlinear loss functions. We exploit the rich structures of the reformulated problem and develop a stochastic primal-dual algorithm, SVRPDA-I, to solve the problem efficiently. We carry out extensive theoretical analysis of the proposed algorithm, obtaining the convergence rate, the computation complexity and the storage complexity. In particular, the algorithm is shown to converge at a *linear* rate when the problem is strongly convex. Moreover, we also develop an approximate version of the algorithm, named SVRPDA-II, which further reduces the memory requirement. Finally, we evaluate our proposed algorithms on several real-world benchmarks, and experimental results show that the proposed algorithms significantly outperform existing techniques.

## 1   Introduction

In this paper, we consider the following regularized *empirical composition optimization* problem:

$$\min_\theta \frac{1}{n_X} \sum_{i=0}^{n_X-1} \phi_i\left( \frac{1}{n_{Y_i}} \sum_{j=0}^{n_{Y_i}-1} f_\theta(x_i, y_{ij}) \right) + g(\theta), \tag{1}$$

where $(x_i, y_{ij}) \in \mathbb{R}^{m_x} \times \mathbb{R}^{m_y}$ is the $(i,j)$-th data sample, $f_\theta : \mathbb{R}^{m_x} \times \mathbb{R}^{m_y} \to \mathbb{R}^\ell$ is a function parameterized by $\theta \in \mathbb{R}^d$, $\phi_i : \mathbb{R}^\ell \to \mathbb{R}^+$ is a convex *merit function*, which measures a certain loss of the parametric function $f_\theta$, and $g(\theta)$ is a $\mu$-strongly convex regularization term.

Problems of the form (1) widely appear in many machine learning applications such as reinforcement learning [5, 3, 2, 13], unsupervised sequence classification [12, 21] and risk-averse learning [15, 18, 9, 10, 19] — see our detailed discussion in Section 2. Note that the cost function (1) has an empirical average (over $x_i$) outside the (nonlinear) merit function $\phi_i(\cdot)$ and an empirical average (over $y_{ij}$) inside the merit function, which makes it different from the *empirical risk minimization* problems that are common in machine learning [17]. Problem (1) can be understood as a generalized version of the one considered in [9, 10].[3] In these prior works, $y_{ij}$ and $n_{Y_i}$ are assumed to be independent of

---

[*]Department of Electrical and Computer Engineering, University of Florida, Gainesville, USA. Email: `adithyamdevraj@ufl.edu`. The work was done during an internship at Tencent AI Lab, Bellevue, WA.

[†]Tencent AI Lab, Bellevue, WA, USA. Email: `jianshuchen@tencent.com`.

[3]In addition to the term in (2), the cost function in [10] also has another convex regularization term.

$i$ and $f_\theta$ is only a function of $y_j$ so that problem (1) can be reduced to the following special case:

$$\min_\theta \frac{1}{n_X} \sum_{i=0}^{n_X-1} \phi_i \left( \frac{1}{n_Y} \sum_{j=0}^{n_Y-1} f_\theta(y_j) \right). \tag{2}$$

Our more general problem formulation (1) encompasses wider applications (see Section 2). Furthermore, different from [2, 19, 18], we focus on the finite sample setting, where we have empirical averages (instead of expectations) in (1). As we shall see below, the finite-sum structures allows us to develop efficient stochastic gradient methods that converges at linear rate.

While problem (1) is important in many machine learning applications, there are several key challenges in solving it efficiently. First, the number of samples (i.e., $n_X$ and $n_{Y_i}$) could be extremely large: they could be larger than one million or even one billion. Therefore, it is unrealistic to use batch gradient descent algorithm to solve the problem, which requires going over all the data samples at each gradient update step. Moreover, since there is an empirical average inside the nonlinear merit function $\phi_i(\cdot)$, it is not possible to directly apply the classical stochastic gradient descent (SGD) algorithm. This is because sampling from both empirical averages outside and inside $\phi_i(\cdot)$ simultaneously would make the stochastic gradients intrinsically biased (see Appendix A for a discussion).

To address these challenges, in this paper, we first reformulate the original problem (1) into an equivalent saddle point problem (i.e., min-max problem), which brings out all the empirical averages inside $\phi_i(\cdot)$ and exhibits useful dual decomposition and finite-sum structures (Section 3.1). To fully exploit these properties, we develop a stochastic primal-dual algorithm that alternates between a dual step of stochastic variance reduced coordinate ascent and a primal step of stochastic variance reduced gradient descent (Section 3.2). In particular, we develop a novel variance reduced stochastic gradient estimator for the primal step, which achieves better variance reduction with low complexity (Section 3.3). We derive the convergence rate, the finite-time complexity bound, and the storage complexity of our proposed algorithm (Section 4). In particular, it is shown that the proposed algorithms converge at a *linear* rate when the problem is strongly convex. Moreover, we also develop an approximate version of the algorithm that further reduces the storage complexity without much performance degradation in experiments. We evaluate the performance of our algorithms on several real-world benchmarks, where the experimental results show that they significantly outperform existing methods (Section 5). Finally, we discuss related works in Section 6 and conclude our paper in Section 7.

## 2 Motivation and Applications

To motivate our composition optimization problem (1), we discuss several important machine learning applications where cost functions of the form (1) arise naturally.

**Unsupervised sequence classification:** Developing algorithms that can learn classifiers from unlabeled data could benefit many machine learning systems, which could save a huge amount of human labeling costs. In [12, 21], the authors proposed such unsupervised learning algorithms by exploiting the sequential output structures. The developed algorithms are applied to optical character recognition (OCR) problems and automatic speech recognition (ASR) problems. In these works, the learning algorithms seek to learn a sequence classifier by optimizing the empirical output distribution match (Empirical-ODM) cost, which is in the following form (written in our notation):

$$\min_\theta \left\{ - \sum_{i=0}^{n_X-1} p_{\mathrm{LM}}(x_i) \log \left( \frac{1}{n_Y} \sum_{j=0}^{n_Y-1} f_\theta(x_i, y_j) \right) \right\}, \tag{3}$$

where $p_{\mathrm{LM}}$ is a known language model (LM) that describes the distribution of output sequence (e.g., $x_i$ represents different $n$-grams), and $f_\theta$ is a functional of the sequence classifier to be learned, with $\theta$ being its model parameter vector. The key idea is to learn the classifier so that its predicted output $n$-gram distribution is close to the prior $n$-gram distribution $p_{\mathrm{LM}}$ (see [12, 21] for more details). The cost function (3) can be viewed as a special case of (1) by setting $n_{Y_i} = n_Y$, $y_{ij} = y_j$ and $\phi_i(u) = -p_{LM}(x_i) \log(u)$. Note that the formulation (2) cannot be directly used here, because of the dependency of the function $f_\theta$ on both $x_i$ and $y_j$.

**Risk-averse learning:** Another application where (1) arises naturally is the risk-averse learning problem, which is common in finance [15, 18, 9, 10, 19, 20]. Let $x_i \in \mathbb{R}^d$ be a vector consisting of

the rewards from $d$ assets at the $i$-th instance, where $0 \leq i \leq n-1$. The objective in risk-averse learning is to find the optimal weights of the $d$ assets so that the average returns are maximized while the risk is minimized. It could be formulated as the following optimization problem:

$$\min_{\theta} -\frac{1}{n} \sum_{i=0}^{n-1} \langle x_i, \theta \rangle + \frac{1}{n} \sum_{i=0}^{n-1} \left( \langle x_i, \theta \rangle - \frac{1}{n} \sum_{j=0}^{n-1} \langle x_j, \theta \rangle \right)^2, \tag{4}$$

where $\theta \in \mathbb{R}^d$ denotes the weight vector. The objective function in (4) seeks a tradeoff between the mean (the first term) and the variance (the second term). It can be understood as a special case of (2) (which is a further special case of (1)) by making the following identifications:

$$n_X = n_Y = n, \ y_i \equiv x_i, \ f_\theta(y_j) = [\theta^T, \ -\langle y_j, \ \theta \rangle]^T, \ \phi_i(u) = (\langle x_i, u_{0:d-1} \rangle + u_d)^2 - \langle x_i, u_{0:d-1} \rangle, \tag{5}$$

where $u_{0:d-1}$ denotes the subvector constructed from the first $d$ elements of $u$, and $u_d$ denotes the $d$-th element. An alternative yet simpler way of dealing with (4) is to treat the second term in (4) as a special case of (1) by setting

$$n_X = n_{Y_i} = n, \ y_{ij} \equiv x_j, \ f_\theta(x_i, y_{ij}) = \langle x_i - y_{ij}, \theta \rangle, \ \phi_i(u) = u^2, u \in \mathbb{R}. \tag{6}$$

In addition, we observe that the first term in (4) is in standard empirical risk minimization form, which can be dealt with in a straightforward manner. This second formulation leads to algorithms with lower complexity due to the lower dimension of the functions: $\ell = 1$ instead of $\ell = d + 1$ in the first formulation. Therefore, we will adopt this formulation in our experiment section (Section 5).

**Other applications:** Cost functions of the form (1) also appear in reinforcement learning [5, 2, 3] and other applications [18]. In Appendix D, we demonstrate its applications in policy evaluation.

## 3 Algorithms

### 3.1 Saddle point formulation

Recall from (1) that there is an empirical average inside each (nonlinear) merit function $\phi_i(\cdot)$, which prevents the direct application of stochastic gradient descent to (1) due to the inherent bias (see Appendix A for more discussions). Nevertheless, we will show that minimizing the original cost function (1) can be transformed into an equivalent saddle point problem, which brings out all the empirical averages inside $\phi_i(\cdot)$. In what follows, we will use the machinery of *convex conjugate functions* [14]. For a function $\psi : \mathbb{R}^\ell \to \mathbb{R}$, its convex conjugate function $\psi^* : \mathbb{R}^\ell \to \mathbb{R}$ is defined as $\psi^*(y) = \sup_{x \in \mathbb{R}^\ell}(\langle x, y \rangle - \psi(x))$. Under certain mild conditions on $\psi(x)$ [14], one can also express $\psi(x)$ as a functional of its conjugate function: $\psi(x) = \sup_{y \in \mathbb{R}^\ell}(\langle x, y \rangle - \psi^*(y))$. Let $\phi_i^*(w_i)$ denote the conjugate function of $\phi_i(u)$. Then, we can express $\phi_i(u)$ as

$$\phi_i(u) = \sup_{w_i \in \mathbb{R}^\ell} (\langle u, w_i \rangle - \phi_i^*(w_i)), \tag{7}$$

where $w_i$ is the corresponding dual variable. Substituting (7) into the original minimization problem (1), we obtain its equivalent min-max problem as:

$$\min_{\theta} \max_{w} \left\{ L(\theta, w) + g(\theta) \triangleq \frac{1}{n_X} \sum_{i=0}^{n_X-1} \left[ \left\langle \frac{1}{n_{Y_i}} \sum_{j=0}^{n_{Y_i}-1} f_\theta(x_i, y_{ij}), w_i \right\rangle - \phi_i^*(w_i) \right] + g(\theta) \right\}, \tag{8}$$

where $w \triangleq \{w_0, \ldots, w_{n_X-1}\}$, is a collection of all dual variables. We note that the transformation of the original problem (1) into (8) brings out all the empirical averages that are present inside $\phi_i(\cdot)$. This new formulation allows us to develop stochastic variance reduced algorithms below.

### 3.2 Stochastic variance reduced primal-dual algorithm

One common solution for the min-max problem (8) is to alternate between the step of minimization (with respect to the *primal variable* $\theta$) and the step of maximization (with respect to the *dual variable* $w$). However, such an approach generally suffers from high computation complexity because each minimization/maximization step requires a summation over many components and requires a full

pass over all the data samples. The complexity of such a batch algorithm would be prohibitively high when the number of data samples (i.e., $n_X$ and $n_{Y_i}$) is large (e.g., they could be larger than one million or even one billion in applications like unsupervised speech recognition [21]). On the other hand, problem (8) indeed has rich structures that we can exploit to develop more efficient solutions.

To this end, we make the following observations. First, expression (8) implies that when $\theta$ is fixed, the maximization over the dual variable $w$ can be decoupled into a total of $n_X$ individual maximizations over different $w_i$'s. Second, the objective function in each individual maximization (with respect to $w_i$) contains a finite-sum structure over $j$. Third, by (8), for a fixed $w$, the minimization with respect to the primal variable $\theta$ is also performed over an objective function with a finite-sum structure. Based on these observations, we will develop an efficient stochastic variance reduced primal-dual algorithm (named SVRPDA-I). It alternates between (i) a dual step of stochastic variance reduced coordinate ascent and (ii) a primal step of stochastic variance reduced gradient descent. The full algorithm is summarized in Algorithm 1, with its key ideas explained below.

**Dual step: stochastic variance reduced coordinate ascent.** To exploit the decoupled dual maximization over $w$ in (8), we can randomly sample an index $i$, and update $w_i$ according to:

$$w_i^{(k)} = \arg\min_{w_i} \left\{ - \left\langle \frac{1}{n_{Y_i}} \sum_{j=0}^{n_{Y_i}-1} f_{\theta^{(k-1)}}(x_i, y_{ij}), w_i \right\rangle + \phi_i^*(w_i) + \frac{1}{2\alpha_w} \|w_i - w_i^{(k-1)}\|^2 \right\}, \quad (9)$$

while keeping all other $w_j$'s ($j \neq i$) unchanged, where $\alpha_w$ denotes a step-size. Note that each step of recursion (9) still requires a summation over $n_{Y_i}$ components. To further reduce the complexity, we approximate the sum over $j$ by a variance reduced stochastic estimator defined in (12) (to be discussed in Section 3.3). The dual step in our algorithm is summarized in (13), where we assume that the function $\phi_i^*(w_i)$ is in a simple form so that the argmin could be solved in closed-form. Note that we flip the sign of the objective function to change maximization to minimization and apply coordinate descent. We will still refer to the dual step as "coordinate ascent" (instead of descent).

**Primal step: stochastic variance reduced gradient descent** We now consider the minimization in (8) with respect to $\theta$ when $w$ is fixed. The gradient descent step for minimizing $L(\theta, w)$ is given by

$$\theta^{(k)} = \arg\min_{\theta} \left\{ \left\langle \sum_{i=0}^{n_X-1} \sum_{j=0}^{n_{Y_i}-1} \frac{1}{n_X n_{Y_i}} f'_{\theta^{(k-1)}}(x_i, y_{ij}) w_i^{(k)}, \theta \right\rangle + \frac{1}{2\alpha_\theta} \|\theta - \theta^{(k-1)}\|^2 \right\}, \quad (10)$$

where $\alpha_\theta$ denotes a step-size. It is easy to see that the update equation (10) has high complexity, it requires evaluating and averaging the gradient $f'_\theta(\cdot, \cdot)$ at every data sample. To reduce the complexity, we use a variance reduced gradient estimator, defined in (15), to approximate the sums in (10) (to be discussed in Section 3.3). The primal step in our algorithm is summarized in (16) in Algorithm 1.

### 3.3 Low-complexity stochastic variance reduced estimators

We now proceed to explain the design of the variance reduced gradient estimators in both the dual and the primal updates. The main idea is inspired by the stochastic variance reduced gradient (SVRG) algorithm [7]. Specifically, for a vector-valued function $h(\theta) = \frac{1}{n} \sum_{i=0}^{n-1} h_i(\theta)$, we can construct its SVRG estimator $\delta_k$ at each iteration step $k$ by using the following expression:

$$\delta_k = h_{i_k}(\theta) - h_{i_k}(\tilde{\theta}) + h(\tilde{\theta}), \quad (17)$$

where $i_k$ is a randomly sampled index from $\{0, \ldots, n-1\}$, and $\tilde{\theta}$ is a reference variable that is updated *periodically* (to be explained below). The first term $h_i(\theta)$ in (17) is an unbiased estimator of $h(\theta)$ and is generally known as the *stochastic gradient* when $h(\theta)$ is the gradient of a certain cost function. The last two terms in (17) construct a control variate that has zero mean and is negatively correlated with $h_i(\theta)$, which keeps $\delta_k$ unbiased while significantly reducing its variance. The reference variable $\tilde{\theta}$ is usually set to be a delayed version of $\theta$: for example, after every $M$ updates of $\theta$, it can be reset to the most recent iterate of $\theta$. Note that there is a trade-off in the choice of $M$: a smaller $M$ further reduces the variance of $\delta_k$ since $\tilde{\theta}$ will be closer to $\theta$ and the first two terms in (17) cancel more with each other; on the other hand, it will also require more frequent evaluations of the costly batch term $h(\tilde{\theta})$, which has a complexity of $O(n)$.

**Algorithm 1** SVRPDA-I

1: **Inputs:** data $\{(x_i, y_{ij}) : 0 \le i < n_X, 0 \le j < n_{Y_i}\}$; step-sizes $\alpha_\theta$ and $\alpha_w$; # inner iterations $M$.
2: **Initialization:** $\tilde{\theta}_0 \in \mathbb{R}^d$ and $\tilde{w}_0 \in \mathbb{R}^{\ell n_X}$.
3: **for** $s = 1, 2, \dots$ **do**
4:  Set $\tilde{\theta} = \tilde{\theta}_{s-1}$, $\theta^{(0)} = \tilde{\theta}$, $\tilde{w} = \tilde{w}_{s-1}$, $w^{(0)} = \tilde{w}_{s-1}$, and compute the batch quantities (for each $0 \le i < n_X$):

$$U_0 = \sum_{i=0}^{n_X-1} \sum_{j=0}^{n_{Y_i}-1} \frac{f'_{\tilde{\theta}}(x_i, y_{ij}) w_i^{(0)}}{n_X n_{Y_i}}, \quad \overline{f}_i(\tilde{\theta}) \triangleq \sum_{j=0}^{n_{Y_i}-1} \frac{f_{\tilde{\theta}}(x_i, y_{ij})}{n_{Y_i}}, \quad \overline{f}'_i(\tilde{\theta}) = \sum_{j=0}^{n_{Y_i}-1} \frac{f'_{\tilde{\theta}}(x_i, y_{ij})}{n_{Y_i}}. \tag{11}$$

5:  **for** $k = 1$ **to** $M$ **do**
6:   Randomly sample $i_k \in \{0, \dots, n_X - 1\}$ and then $j_k \in \{0, \dots, n_{Y_{i_k}} - 1\}$ at uniform.
7:   Compute the stochastic variance reduced gradient for dual update:

$$\delta_k^w = f_{\theta^{(k-1)}}(x_{i_k}, y_{i_k j_k}) - f_{\tilde{\theta}}(x_{i_k}, y_{i_k j_k}) + \overline{f}_{i_k}(\tilde{\theta}). \tag{12}$$

8:   Update the dual variables:

$$w_i^{(k)} = \begin{cases} \underset{w_i}{\arg\min} \left[ -\langle \delta_k^w, w_i \rangle + \phi_i^*(w_i) + \frac{1}{2\alpha_w} \|w_i - w_i^{(k-1)}\|^2 \right] & \text{if } i = i_k \\ w_i^{(k-1)} & \text{if } i \ne i_k \end{cases}. \tag{13}$$

9:   Update $U_k$ (primal batch gradient at $\tilde{\theta}$ and $w^{(k)}$) according to the following recursion:

$$U_k = U_{k-1} + \frac{1}{n_X} \overline{f}'_{i_k}(\tilde{\theta}) \big( w_{i_k}^{(k)} - w_{i_k}^{(k-1)} \big). \tag{14}$$

10:   Randomly sample $i'_k \in \{0, \dots, n_X - 1\}$ and then $j'_k \in \{0, \dots, n_{Y_{i'_k}} - 1\}$, independent of $i_k$ and $j_k$, and compute the stochastic variance reduced gradient for primal update:

$$\delta_k^\theta = f'_{\theta^{(k-1)}}(x_{i'_k}, y_{i'_k j'_k}) w_{i'_k}^{(k)} - f'_{\tilde{\theta}}(x_{i'_k}, y_{i'_k j'_k}) w_{i'_k}^{(k)} + U_k. \tag{15}$$

11:   Update the primal variable:

$$\theta^{(k)} = \underset{\theta}{\arg\min} \left[ \langle \delta_k^\theta, \theta \rangle + g(\theta) + \frac{1}{2\alpha_\theta} \|\theta - \theta^{(k-1)}\|^2 \right]. \tag{16}$$

12:  **end for**
13:  **Option I:** Set $\tilde{w}_s = w^{(M)}$ and $\tilde{\theta}_s = \theta^{(M)}$.
14:  **Option II:** Set $\tilde{w}_s = w^{(M)}$ and $\tilde{\theta}_s = \theta^{(t)}$ for randomly sampled $t \in \{0, \dots, M-1\}$.
15: **end for**
16: **Output:** $\tilde{\theta}_s$ at the last outer-loop iteration.

Based on (17), we develop two stochastic variance reduced estimators, (12) and (15), to approximate the finite-sums in (9) and (10), respectively. The dual gradient estimator $\delta_k^w$ in (12) is constructed in a standard manner using (17), where the reference variable $\tilde{\theta}$ is a delayed version of $\theta^{(k)}$[4]. On the other hand, the primal gradient estimator $\delta_k^\theta$ in (15) is constructed by using reference variables $(\tilde{\theta}, w^{(k)})$; that is, we uses the most recent $w^{(k)}$ as the dual reference variable, without any delay. As discussed earlier, such a choice leads to a smaller variance in the stochastic estimator $\delta_\theta^k$ at a potentially higher computation cost (from more frequent evaluation of the batch term). Nevertheless, we are able to show that, with the dual coordinate ascent structure in our algorithm, the batch term $U_k$ in (15), which is the summation in (10) evaluated at $(\tilde{\theta}, w^{(k)})$, can be computed efficiently. To see this, note that, after each dual update step in (13), only one term inside this summation in (10), has been changed, i.e., the one associated with $i = i_k$. Therefore, we can correct $U_k$ for this term by using recursion (14), which only requires an extra $O(d\ell)$-complexity per step (same complexity as (15)).

Note that SVRPDA-I (Algorithm 1) requires to compute and store all the $\overline{f}'_i(\tilde{\theta})$ in (11), which is $O(n_X d\ell)$-complexity in storage and could be expensive in some applications. To avoid the cost, we develop a variant of Algorithm 1, named as SVRPDA-II (see Algorithm 1 in the supplementary material), by approximating $\overline{f}_{i_k}(\tilde{\theta})$ in (14) with $f'_{\tilde{\theta}}(x_{i_k}, y_{i_k j''_k})$, where $j''_k$ is another randomly sampled index from $\{0, \dots, n_{Y_i} - 1\}$, independent of all other indexes. By doing this, we can significantly

Table 1: The total complexities of different stochastic composition optimization algorithms. For C-SAGA, $\alpha = 2/3$ in the minibatch setting and $\alpha = 1$ when batch-size=1. In the bound for ASCVRG, the dependency on $\kappa$ has been dropped since it was not reported in [10].

| Methods | SVRPDA-I (Ours) | Comp-SVRG [9] | C-SAGA [22] | MSPBE-SVRG/SAGA [5] | ASCVRG [10] |
|---|---|---|---|---|---|
| General: problem (1) | $(n_X n_Y + n_X \kappa)\ln\frac{1}{\epsilon}$ | | | | |
| Special: problem (2) | $(n_X+n_Y+n_X\kappa)\ln\frac{1}{\epsilon}$ | $(n_X+n_Y+\kappa^3)\ln\frac{1}{\epsilon}$ | $(n_X+n_Y+(n_X+n_Y)^\alpha\kappa)\ln\frac{1}{\epsilon}$ | | $(n_X+n_Y)\ln\frac{1}{\epsilon}+\frac{1}{\epsilon^3}$ |
| Special: (2) & $n_X=1$ | $(n_Y+\kappa)\ln\frac{1}{\epsilon}$ | $(n_Y+\kappa^3)\ln\frac{1}{\epsilon}$ | $(n_Y+n_Y^\alpha\kappa)\ln\frac{1}{\epsilon}$ | $(n_Y+\kappa^2)\ln\frac{1}{\epsilon}$ | $n_Y\ln\frac{1}{\epsilon}+\frac{1}{\epsilon^3}$ |

reduce the memory requirement from $O(n_X d\ell)$ in SVRPDA-I to $O(d + n_X\ell)$ in SVRPDA-II (see Section 4.2). In addition, experimental results in Section 5 will show that such an approximation only cause slight performance loss compared to that of SVRPDA-I algorithm.

## 4 Theoretical Analysis

### 4.1 Computation complexity

We now perform convergence analysis for the SVRPDA-I algorithm and also derive their complexities in computation and storage. To begin with, we first introduce the following assumptions.

**Assumption 4.1.** *The function $g(\theta)$ is $\mu$-strongly convex in $\theta$, and each $\phi_i$ is $1/\gamma$-smooth.*

**Assumption 4.2.** *The merit functions $\phi_i(u)$ are Lipschitz with a uniform constant $B_w$:*
$$|\phi_i(u) - \phi_i(u')| \le B_w\|u - u'\|, \quad \forall u, u'; \ \forall i = 0, \ldots, n_X - 1.$$

**Assumption 4.3.** *$f_\theta(x_i, y_{ij})$ is $B_\theta$-smooth in $\theta$, and has bounded gradients with constant $B_f$:*
$$\|f'_{\theta_1}(x_i, y_{ij}) - f'_{\theta_2}(x_i, y_{ij})\| \le B_\theta\|\theta_1 - \theta_2\|, \quad \|f'_\theta(x_i, y_{ij})\| \le B_f, \quad \forall\theta, \theta_1, \theta_2, \ \forall i, j.$$

**Assumption 4.4.** *For each given $w$ in its domain, the function $L(\theta, w)$ defined in (8) is convex in $\theta$:*
$$L(\theta_1, w) - L(\theta_2, w) \ge \langle L'_\theta(\theta_2, w), \ \theta_1 - \theta_2\rangle, \quad \forall\theta_1, \theta_2.$$

The above assumptions are commonly used in existing compositional optimization works [9, 10, 18, 19, 22]. Based on these assumptions, we establish the non-asymptotic error bounds for SVRPDA-I (using either Option I or Option II in Algorithm 1). The main results are summarized in the following theorems, and their proofs can be found in Appendix E.

**Theorem 4.5.** *Suppose Assumptions 4.1–4.4 hold. If in Algorithm 1 (with Option I) we choose*
$$\alpha_\theta = \frac{1}{n_X\mu(64\kappa + 1)}, \quad \alpha_w = \frac{n_X\mu}{\gamma}\alpha_\theta, \quad M = \lceil 78.8n_X\kappa + 1.3n_X + 1.3\rceil$$

*where $\lceil x\rceil$ denotes the roundup operation and $\kappa = B_f^2/\gamma\mu + B_w^2 B_\theta^2/\mu^2$, then the Lyapunov function $P_s := \mathsf{E}\|\tilde{\theta}_s - \theta^*\|^2 + \frac{\gamma}{\mu}\cdot\frac{64\kappa+3}{64n_X\kappa+n_X+1}\mathsf{E}\|\tilde{w}_s - w^*\|^2$ satisfies $P_s \le (3/4)^s P_0$. Furthermore, the overall computational cost (in number of oracle calls[5]) for reaching $P_s \le \epsilon$ is upper bounded by*
$$O\big((n_X n_Y + n_X\kappa + n_X)\ln(1/\epsilon)\big). \tag{18}$$

*where, with a slight abuse of notation, $n_Y$ is defined as $n_Y = (n_{Y_0} + \cdots + n_{Y_{n_X - 1}})/n_X$.*

**Theorem 4.6.** *Suppose Assumptions 4.1–4.4 hold. If in Algorithm 1 (with Option II) we choose*
$$\alpha_\theta = \Big(\frac{25B_f^2}{\gamma} + 10B_\theta B_w + \frac{80B_w^2 B_\theta^2}{\mu}\Big)^{-1}, \quad \alpha_w = \frac{\mu}{40B_f^2}, \quad M = \max\Big(\frac{10}{\alpha_\theta\mu}, \frac{2n_X}{\alpha_w\gamma}, 4n_X\Big),$$

*then $P_s := \mathsf{E}\|\tilde{\theta}_s - \theta^*\|^2 + \frac{\gamma}{n_X\mu}\mathsf{E}\|\tilde{w}_s - w^*\|^2 \le (5/8)^s P_0$. Furthermore, let $\kappa = \frac{B_f^2}{\gamma\mu} + \frac{B_w^2 B_\theta^2}{\mu^2}$. Then, the overall computational cost (in number of oracle calls) for reaching $P_s \le \epsilon$ is upper bounded by*
$$O\big((n_X n_Y + n_X\kappa + n_X)\ln(1/\epsilon)\big). \tag{19}$$

The above theorems show that the Lyapunov function $P_s$ for SVRPDA-I converges to zero at a linear rate when either Option I or II is used. Since $\mathsf{E}\|\tilde{\theta}_s - \theta^*\|^2 \le P_s$, they imply that the computational cost (in number of oracle calls) for reaching $\mathsf{E}\|\tilde{\theta}_s - \theta^*\|^2 \le \epsilon$ is also upper bounded by (18) and (19).

Table 2: The storage complexity of SVRPDA-I and SVRPDA-II.

| Methods | $U_0$ | $\{\overline{f}_i\}$ | $\{\overline{f}'_i\}$ | $\theta^{(k)}$ | $\tilde{\theta}$ | $\{w_i^{(k)}\}$ | $\delta_k^\theta$ | $\delta_k^w$ | Total |
|---|---|---|---|---|---|---|---|---|---|
| SVRPDA-I | $O(d)$ | $O(n_X\ell)$ | $O(n_X d\ell)$ | $O(d)$ | $O(d)$ | $O(n_X\ell)$ | $O(d)$ | $O(\ell)$ | $O(n_X d\ell)$ |
| SVRPDA-II | $O(d)$ | $O(n_X\ell)$ |  | $O(d)$ | $O(d)$ | $O(n_X\ell)$ | $O(d)$ | $O(\ell)$ | $O(d+n_X\ell)$ |

**Comparison with existing composition optimization algorithms** Table 1 summarizes the complexity bounds for our SVRPDA-I algorithm and compares them with existing stochastic composition optimization algorithms. First, to our best knowledge, none of the existing methods consider the general objective function (1) as we did. Instead, they consider its special case (2), and even in this special case, our algorithm still has better (or comparable) complexity bound than other methods. For example, our bound is better than that of [9] since $\kappa^2 > n_X$ generally holds, and it is better than that of ASCVRG, which does not achieve linear convergence rate (as no strong convexity is assumed). In addition, our method has better complexity than C-SAGA algorithm when $n_X = 1$ (regardless of mini-batch size in C-SAGA), and it is better than C-SAGA for (2) when the mini-batch size is 1.[6] However, since we have not derived our bound for mini-batch setting, it is unclear which one is better in this case, and is an interesting topic for future work. One notable fact from Table 1 is that in this special case (2), the complexity of SVRPDA-I is reduced from $O((n_X n_Y + n_X \kappa)\ln \frac{1}{\epsilon})$ to $O((n_X + n_Y + n_X\kappa)\ln \frac{1}{\epsilon})$. This is because the complexity for evaluating the batch quantities in (11) (Algorithm 1) can be reduced from $O(n_X n_Y)$ in the general case (1) to $O(n_X + n_Y)$ in the special case (2). To see this, note that $f_\theta$ and $n_{Y_i} = n_Y$ become independent of $i$ in (2) and (11), meaning that we can factor $U_0$ in (11) as $U_0 = \frac{1}{n_X n_Y} \sum_{j=0}^{n_Y-1} f'_{\tilde{\theta}}(y_j) \sum_{i=0}^{n_X} w_i^{(0)}$, where the two sums can be evaluated independently with complexity $O(n_Y)$ and $O(n_X)$, respectively. The other two quantities in (11) need only $O(n_Y)$ due to their independence of $i$. Second, we consider the further special case of (2) with $n_X = 1$, which simplifies the objective function (1) so that there is no empirical average outside $\phi_i(\cdot)$. This takes the form of the unsupervised learning objective function that appears in [12]. Note that our results $O((n_Y + \kappa)\log \frac{1}{\epsilon})$ enjoys a linear convergence rate (i.e., log-dependency on $\epsilon$) due to the variance reduction technique. In contrast, stochastic primal-dual gradient (SPDG) method in [12], which does not use variance reduction, can only have sublinear convergence rate (i.e., $O(\frac{1}{\epsilon})$).

**Relation to SPDC [23]** Lastly, we consider the case where $n_{Y_i} = 1$ for all $1 \le i \le n_X$ and $f_\theta$ is a linear function in $\theta$. This simplifies (1) to the problem considered in [23], known as the *regularized empirical risk minimization of linear predictors*. It has applications in support vector machines, regularized logistic regression, and more, depending on how the merit function $\phi_i$ is defined. In this special case, the overall complexity for SVRPDA-I becomes (see Appendix F):

$$O\big((n_X + \kappa)\ln(1/\epsilon)\big), \tag{20}$$

where the condition number $\kappa = B_f^2/\mu\gamma$. In comparison, the authors in [23] propose a *stochastic primal dual coordinate* (SPDC) algorithm for this special case and prove an overall complexity of $O\big((n_X + \sqrt{n_X \kappa})\ln\big(\frac{1}{\epsilon}\big)\big)$ to achieve an $\epsilon$-error solution. It is interesting to note that the complexity result in (20) and the complexity result in [23] only differ in their dependency on $\kappa$. This difference is most likely due to the acceleration technique that is employed in the primal update of the SPDC algorithm. We conjecture that the dependency on the condition number of SVRPDA-I can be further improved using a similar acceleration technique.

### 4.2 Storage complexity

We now briefly discuss and compare the storage complexities of both SVRPDA-I and SVRPDA-II. In Table 2, we report the itemized and total storage complexities for both algorithms, which shows that SVRPDA-II significantly reduces the memory footprint. We also observe that the batch quantities in (11), especially $\overline{f}'_i(\tilde{\theta})$, dominates the storage complexity in SVRPDA-I. On the other hand, the memory usage in SVRPDA-II is more uniformly distributed over different quantities. Furthermore, although the total complexity of SVRPDA-II, $O(d + n_X\ell)$, grows with the number of samples $n_X$, the $n_X\ell$ term is relatively small because the dimension $\ell$ is small in many practical problems (e.g., $\ell = 1$ in (3) and (4)). This is similar to the storage requirement in SPDC [23] and SAGA [4].

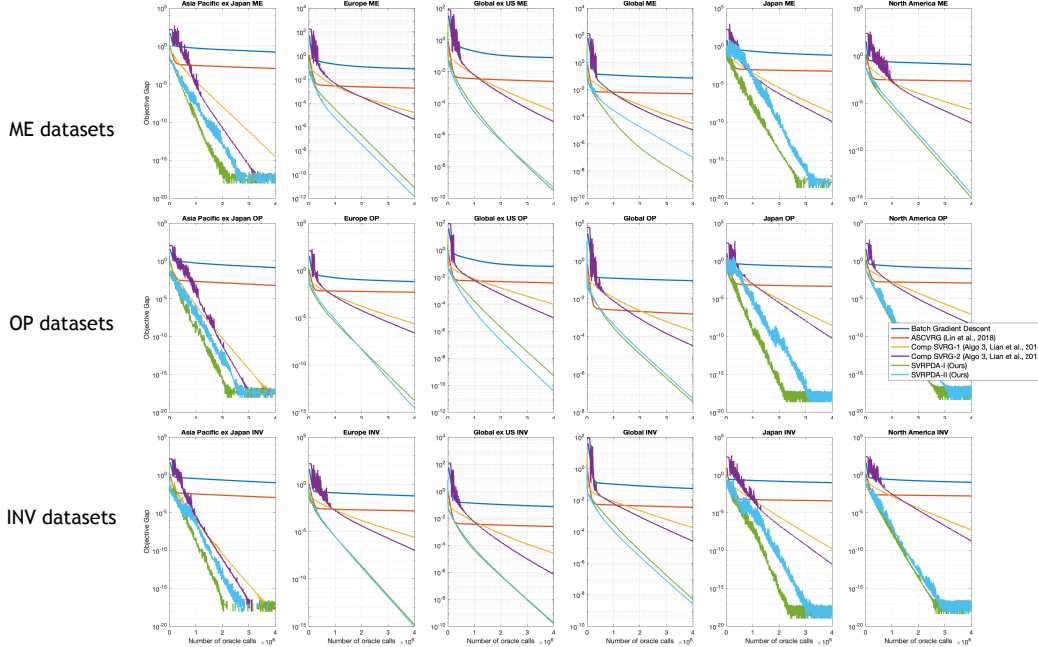

Figure 1: Performance of different algorithms on the risk-averse learning for portfolio management optimization problem. The performance is measured in terms of the number of oracle calls required to achieve a certain objective gap.

## 5 Experiments

In this section we consider the problem of risk-averse learning for portfolio management optimization [9, 10], introduced in Section 2.[7] Specifically, we want to solve the optimization problem (4) for a given set of reward vectors $\{x_i \in \mathbb{R}^d : 0 \leq i \leq n-1\}$. As we discussed in Section 2, we adopt the alternative formulation (6) for the second term so that it becomes a special case of our general problem (1). Then, we rewrite the cost function into a min-max problem by following the argument in Section 3.1 and apply our SVRPDA-I and SVRPDA-II algorithms (see Appendix C.1 for the details).

We evaluate our algorithms on 18 real-world US Research Returns datasets obtained from the Center for Research in Security Prices (CRSP) website[8], with the same setup as in [10]. In each of these datasets, we have $d = 25$ and $n = 7240$. We compare the performance of our proposed SVRPDA-I and SVRPDA-II algorithms[9] with the following state-of-the art algorithms designed to solve composition optimization problems: (i) Compositional-SVRG-1 (Algorithm 2 of [9]), (ii) Compositional-SVRG-2 (Algorithm 3 of [9]), (iii) Full batch gradient descent, and (iv) ASCVRG algorithm [10]. For the compositional-SVRG algorithms, we follow [9] to formulate it as a special case of the form (2) by using the identification (5). Note that we cannot use the identification (6) for the compositional SVRG algorithms because it will lead to the more general formulation (1) with $f_\theta$ depending on both $x_i$ and $y_{ij} \equiv x_j$. For further details, the reader is referred to [9].

As in previous works, we compare different algorithms based on the number of *oracle calls* required to achieve a certain objective gap (the difference between the objective function evaluated at the current iterate and at the optimal parameters). One *oracle call* is defined as accessing the function $f_\theta$, its derivative $f_\theta'$, or $\phi_i(u)$ for any $0 \leq i < n$ and $u \in \mathbb{R}^\ell$. The results are shown in Figure 1, which shows that our proposed algorithms significantly outperform the baseline methods on all datasets. In addition, we also observe that SVRPDA-II also converges at a linear rate, and the performance loss caused by the approximation is relatively small compared to SVRPDA-I.

# 6 Related Works

Composition optimization have attracted significant attention in optimization literature. The stochastic version of the problem (2), where the empirical averages are replaced by expectations, is studied in [18]. The authors propose a two-timescale stochastic approximation algorithm known as SCGD, and establish *sublinear* convergence rates. In [19], the authors propose the ASC-PG algorithm by using a proximal gradient method to deal with nonsmooth regularizations. The works that are more closely related to our setting are [9] and [10], which consider a finite-sum minimization problem (2) (a special case of our general formulation (1)). In [9], the authors propose the compositional-SVRG methods, which combine SCGD with the SVRG technique from [7] and obtain *linear* convergence rates. In [10], the authors propose the ASCVRG algorithms that extends to convex but non-smooth objectives. Recently, the authors in [22] propose a C-SAGA algorithm to solve the special case of (2) with $n_X = 1$, and extend to general $n_X$. Different from these works, we take an efficient primal-dual approach that fully exploits the dual decomposition and the finite-sum structures.

On the other hand, problems similar to (1) (and its stochastic versions) are also examined in different specific machine learning problems. [16] considers the minimization of the mean square projected Bellman error (MSPBE) for policy evaluation, which has an expectation inside a *quadratic loss*. The authors propose a two-timescale stochastic approximation algorithm, GTD2, and establish its asymptotic convergence. [11] and [13] independently showed that the GTD2 is a stochastic gradient method for solving an equivalent saddle-point problem. In [2] and [3], the authors derived saddle-point formulations for two other variants of costs (MSBE and MSCBE) in the policy evaluation and the control settings, and develop their stochastic primal-dual algorithms. All these works consider the stochastic version of the composition optimization and the proposed algorithms have sublinear convergence rates. In [5], different variance reduction methods are developed to solve the finite-sum version of MSPBE and achieve linear rate even without strongly convex regularization. Then the authors in [6] extends this linear convergence results to the general convex-concave problem with linear coupling and without strong convexity. Besides, problem of the form (1) was also studied in the context of unsupervised learning [12, 21] in the stochastic setting (with expectations in (1)).

Finally, our work is inspired by the stochastic variance reduction techniques in optimization [8, 7, 4, 1, 23], which considers the minimization of a cost that is a finite-sum of many component functions. Different versions of variance reduced stochastic gradients are constructed in these works to achieve linear convergence rate. In particular, our variance reduced stochastic estimators are constructed based on the idea of SVRG [7] with a novel design of the control variates. Our work is also related to the SPDC algorithm [23], which also integrates dual coordinate ascent with variance reduced primal gradient. However, our work is different from SPDC in the following aspects. First, we consider a more general composition optimization problem (1) while SPDC focuses on regularized empirical risk minimization with linear predictors, i.e., $n_{Y_i} \equiv 1$ and $f_\theta$ is linear in $\theta$. Second, because of the composition structures in the problem, our algorithms also needs SVRG in the dual coordinate ascent update, while SPDC does not. Third, the primal update in SPDC is specifically designed for linear predictors. In contrast, our work is not restricted to that by using a novel variance reduced gradient.

# 7 Conclusions and Future Work

We developed a stochastic primal-dual algorithms, SVRPDA-I to *efficiently* solve the empirical composition optimization problem. This is achieved by fully exploiting the rich structures inherent in the reformulated min-max problem, including the dual decomposition and the finite-sum structures. It alternates between (i) a dual step of stochastic variance reduced coordinate ascent and (ii) a primal step of stochastic variance reduced gradient descent. In particular, we proposed a novel variance reduced gradient for the primal update, which achieves better variance reduction with low complexity. We derive a *non-asymptotic* bound for the error sequence and show that it converges at a *linear* rate when the problem is strongly convex. Moreover, we also developed an approximate version of the algorithm named SVRPDA-II, which further reduces the storage complexity. Experimental results on several real-world benchmarks showed that both SVRPDA-I and SVRPDA-II significantly outperform existing techniques on all these tasks, and the approximation in SVRPDA-II only caused a slight performance loss. Future extensions of our work include the theoretical analysis of SVRPDA-II, the generalization of our algorithms to Bregman divergences, and applying it to large-scale machine learning problems with non-convex cost functions (e.g., unsupervised sequence classifications).

## Footnotes

[4] As in [7], we also consider Option II wherein $\tilde{\theta}$ is randomly chosen from the previous $M$ $\theta^{(k)}$'s.

[5]One *oracle call* is defined as querying $f_\theta$, $f'_\theta$, or $\phi_i(u)$ for any $0 \le i < n$ and $u \in \mathbb{R}^\ell$.

[6]In Appendix D, we also show that our algorithms outperform C-SAGA in experiments.

[7] Additional experiments on the application to policy evaluation in MDPs can be found in Appendix D.

[8] The processed data in the form of .mat file was obtained from `https://github.com/tyDLin/SCVRG`

[9] The choice of the hyper-parameters can be found in Appendix C.2, and the code will be released publicly.

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
