[Supplementary Material · NeurIPS2019_SCO_Supplementary.pdf]

# Supplemental Materials

## A   Solving (1) in the main paper directly by SGD is biased

Applying the standard chain rule, we obtain the gradient of the cost function in (1) as

$$\frac{1}{n_X} \sum_{i=0}^{n_X-1} \phi_i'\big(\overline{f}_i(\theta)\big)\overline{f}_i'(\theta) \tag{1}$$

where:

$$\overline{f}_i(\theta) \triangleq \frac{1}{n_{Y_i}} \sum_{j=0}^{n_{Y_i}-1} f_\theta(x_i, y_{ij})$$

$$\overline{f}_i'(\theta) \triangleq \frac{1}{n_{Y_i}} \sum_{j=0}^{n_{Y_i}-1} f_\theta'(x_i, y_{ij}) \tag{2}$$

and $f_\theta'(x, y)$ denotes the $d \times \ell$ matrix, with its $(i, j)^{\text{th}}$ element defined to be:

$$\big[f_\theta'(x, y)\big]_{i,j} = \frac{\partial}{\partial \theta_i}\big[(f_\theta(x, y)\big]_j \tag{3}$$

Note from (1) that there are empirical averages inside and outside $\phi'(\cdot)$. Therefore, if we sample these empirical averages simultaneously, the stochastic gradient estimator would be biased. In other words, a direct application of stochastic gradient descent to (1) would be intrinsically biased.

## B   SVRPDA-II algorithm

Algorithm 1 in this supplementary material summarizes the full details of the SVRPDA-II algorithm, which was developed in Section 3.3 of the main paper. Note that it no longer requires the computation or the storage of $\overline{f}_i(\tilde{\theta})$ in (4). Also note that the $\overline{f}_{i_k}(\tilde{\theta})$ in (7) is replaced with $f_{\tilde{\theta}}'(x_{i_k}, y_{i_k j_k''})$ now.

## C   Experiment details

### C.1   Implementation details in risk-averse learning

As we discussed in Section 2, we adopt the alternative formulation (6) for the second term so that it becomes a special case of our general problem (1). Then, using the argument in Section 3.1, the second term in (4) can be rewritten into the objective in (8). Combining it with the first term in (4), the original problem (4) can be reformulated into the following equivalent min-max form:

$$\min_{\theta \in \mathbb{R}^d} \max_{w} \frac{1}{n} \sum_{i=0}^{n-1} \Big(\Big\langle \frac{1}{n} \sum_{j=0}^{n-1} \langle x_i - x_j, \theta\rangle, w_i\Big\rangle - \phi^*(w_i) - \langle x_i, \theta\rangle\Big) \tag{10}$$

where $w_i \in \mathbb{R}$, $\phi^*(w_i) = w_i^2/4$ and $w = \{w_0, \ldots, w_{n-1}\}$. Note that the above min-max problem has an extra $\langle x_i, \theta\rangle$ term within the sum. Since it is in a standard empirical average form, we can deal with it in a straightforward manner. Notice that (10) is exactly of the form (1) in the main paper except the last term $\langle x_i, \theta\rangle$ within the summation, which as we will show next, can be dealt with in a straightforward manner.

Taking out the $\langle x_i, \theta\rangle$ term in (10), based on the discussion in Section 3 of the main paper, the batch gradients used in the algorithm are as follows. Batch gradient of (10) with respect to $w_i$, for each $0 \le i \le n-1$ can be written as:

$$\overline{f}_i(\theta) = \frac{1}{n} \sum_{j=0}^{n-1} f_\theta(x_i, x_j) = \frac{1}{n} \sum_{j=0}^{n-1} \langle x_i - x_j, \theta\rangle \tag{11}$$

**Algorithm 1** SVRPDA-II

1: **Inputs:** data $\{(x_i, y_{ij}) : 0 \leq i < n_X, 0 \leq j < n_{Y_i}\}$; step-sizes $\alpha_\theta$ and $\alpha_w$; # inner iterations $M$.

2: **Initialization:** $\tilde{\theta}_0 \in \mathbb{R}^d$ and $\tilde{w}_0 \in \mathbb{R}^{\ell n_X}$.

3: **for** $s = 1, 2, \ldots$ **do**

4:     Set $\tilde{\theta} = \tilde{\theta}_{s-1}$, $\theta^{(0)} = \tilde{\theta}$, $w^{(0)} = \tilde{w}_{s-1}$, and compute the batch quantities (for each $0 \leq i < n_X$):

$$U_0 = \sum_{i=0}^{n_X-1} \sum_{j=0}^{n_{Y_i}-1} \frac{f'_{\tilde{\theta}}(x_i, y_{ij}) w_i^{(0)}}{n_X n_{Y_i}}, \quad \overline{f}_i(\tilde{\theta}) \triangleq \sum_{j=0}^{n_{Y_i}-1} \frac{f_{\tilde{\theta}}(x_i, y_{ij})}{n_{Y_i}}. \tag{4}$$

5:     **for** $k = 1$ **to** $M$ **do**

6:         Randomly sample $i_k \in \{0, \ldots, n_X - 1\}$ and then $j_k \in \{0, \ldots, n_{Y_{i_k}} - 1\}$ at uniform.

7:         Compute the stochastic variance reduced gradient for dual update:

$$\delta_k^w = f_{\theta^{(k-1)}}(x_{i_k}, y_{i_k j_k}) - f_{\tilde{\theta}}(x_{i_k}, y_{i_k j_k}) + \overline{f}_{i_k}(\tilde{\theta}). \tag{5}$$

8:         Update the dual variables:

$$w_i^{(k)} = \begin{cases} \underset{w_i}{\arg\min} \left[ -\langle \delta_k^w, w_i \rangle + \phi_i^*(w_i) + \frac{1}{2\alpha_w} \|w_i - w_i^{(k-1)}\|^2 \right] & \text{if } i = i_k \\ w_i^{(k-1)} & \text{if } i \neq i_k \end{cases}. \tag{6}$$

9:         Update $U_k$ according to the following recursion:

$$U_k = U_{k-1} + \frac{1}{n_X} f'_{\tilde{\theta}}(x_{i_k}, y_{i_k j_k''}) \big(w_{i_k}^{(k)} - w_{i_k}^{(k-1)}\big). \tag{7}$$

10:         Randomly sample $i_k' \in \{0, \ldots, n_X - 1\}$ and then $j_k' \in \{0, \ldots, n_{Y_{i_k'}} - 1\}$, independent of $i_k$ and $j_k$, and compute the stochastic variance reduced gradient for primal update:

$$\delta_k^\theta = f'_{\theta^{(k-1)}}(x_{i_k'}, y_{i_k' j_k'}) w_{i_k'}^{(k)} - f'_{\tilde{\theta}}(x_{i_k'}, y_{i_k' j_k'}) w_{i_k'}^{(k)} + U_k. \tag{8}$$

11:         Update the primal variable:

$$\theta^{(k)} = \underset{\theta}{\arg\min} \left[ \langle \delta_k^\theta, \theta \rangle + g(\theta) + \frac{1}{2\alpha_\theta} \|\theta - \theta^{(k-1)}\|^2 \right]. \tag{9}$$

12:     **end for**

13:     **Option I:** Set $\tilde{w}_s = w^{(k)}$ and $\tilde{\theta}_s = \theta^{(k)}$.

14:     **Option II:** Set $\tilde{w}_s = w^{(k)}$ and $\tilde{\theta}_s = \theta^{(t)}$ for randomly sampled $t \in \{0, \ldots, M-1\}$.

15: **end for**

16: **Output:** $\tilde{\theta}_s$ at the last outer-loop iteration.

Batch gradient of (10) with respect to $\theta$ (without the $\langle x_i, \theta \rangle$ term) is given by:

$$\begin{aligned} L'_\theta(\theta, w) &= \frac{1}{n^2} \sum_{i=0}^{n-1} \sum_{j=0}^{n-1} f'_\theta(x_i, x_j) w_i \\ &= \frac{1}{n^2} \sum_{i=0}^{n-1} \sum_{j=0}^{n-1} (x_i - x_j) w_i \end{aligned} \tag{12}$$

For each $0 \leq i \leq n - 1$, gradient of $\overline{f}_{\theta,i}(x_i)$ is given by:

$$\overline{f}'_i(\theta) \triangleq \frac{1}{n} \sum_{j=0}^{n-1} f'_\theta(x_i, x_j) = x_i - \frac{1}{n} \sum_{j=0}^{n-1} x_j \tag{13}$$

Based on the above derivation and the expression (17) in the main paper, the stochastic variance reduced gradient for the dual update in both SVRPDA-I and SVRPDA-II is given by

$$\delta_k^w = \langle x_i - x_j, \theta \rangle + \overline{f}_{\tilde{\theta}}(x_i) \tag{14}$$

and the stochastic variance reduced gradient for the primal update is given by

$$\delta_k^\theta \triangleq (x_i - x_j)w_i - (x_i - x_j)w_i + \overline{L}_\theta'(\tilde{\theta}, w) = \overline{L}_\theta'(\tilde{\theta}, w) \tag{15}$$

Note that, since the function $f_\theta$ is linear in $\theta$, the variance reduced gradient for the primal variable is in-fact the full batch gradient.

Next, due to the additional $\langle x_i, \theta \rangle$ term in (10) (which was ommitted in the above definitions), there is an additional term that needs to be added to the variance reduced gradient in (15). Denoting $g_\theta(x_i) = \langle x_i, \theta \rangle$ and $g_\theta'(x_i) = x_i$ the correction batch term is given by:

$$\overline{g}_\theta' = \frac{1}{n} \sum_{i=0}^{n-1} g_\theta'(x_i) = \frac{1}{n} \sum_{i=0}^{n-1} x_i \tag{16}$$

which is independent of $\theta$. In summary, the final variance reduced stochastic gradient for the primal update in both SVRPDA-I and SVRPDA-II is given by:

$$\delta_k^\theta = \overline{L}_\theta'(\tilde{\theta}, w) - \frac{1}{n} \sum_{i=0}^{n-1} x_i \tag{17}$$

### C.2 Hyper-parameter choices for algorithms

In this subsection, we provide the hyper-parameters that are used in our experiments on risk-averse learning (Section 5). We first list the hyper-parameters of our methods below:

- SVRPDA-I: $M = n$, $\alpha_\theta = 0.0003$, $\alpha_w = 100$.
- SVRPDA-II: $M = n$, $\alpha_\theta = 0.0003$, $\alpha_w = 100$.

Then, we provide the hyper-parameters used in the baseline methods:

- Compositional-SVRG-1 (Algorithm 2 of [1]): $K = n$, $A = 6$, $\gamma = 0.0003$;
- Compositional-SVRG-2 (Algorithm 3 of [1]): $K = n$, $A = 3$, $B = 3$, $\gamma = 0.0004$;
- ASCVRG: The results are obtained by using their publicly released code on github: `https://github.com/tyDLin/SCVRG` with the same setting and choice of hyper-parameters.
- Batch gradient descent: step size $\alpha = 0.01$;

Note that, for the Compotional-SVRGs and batch gradient algorithms, the above choice of the hyper-parameters are obtained by sweeping through a set of hyper-parameters and choosing the ones with the best performance. For ASCVRG, we use the the publicly released code by the authors.

## D  Additional experiments on MDP policy evaluation

Consider a Markov decision process (MDP) problem with state space $\mathsf{S}$ and action space $\mathsf{A}$. We assume that both $\mathsf{S}$ and $\mathsf{A}$ are finite, and define $\mathsf{S} = \{1, \ldots, S\}$. For any $1 \le i, j \le S$, we denote by $r_{i,j}$ the reward associated with transition from state $i$ to state $j$. Given a policy $\pi : \mathsf{S} \to \mathcal{P}(\mathsf{A})$, where $\mathcal{P}(\mathsf{A})$ denote the probability space over $\mathsf{A}$, we let $P^\pi \in \mathbb{R}^{S \times S}$ denote the associated state transition probability matrix. The goal in policy evaluation is to estimate the value function $V^\pi : \mathsf{S} \to \mathbb{R}$ associated with the policy $\pi$, which is a fixed-point solution to the following Bellman equation:

$$V^\pi(i) = \sum_{j=1}^{S} P_{i,j}^\pi \left( r_{i,j} + \gamma V^\pi(j) \right), \qquad 1 \le i \le S,$$

where $0 < \gamma < 1$ denotes the discount factor. We consider a linear function approximation to the value function: $V^\pi(i) \approx \langle \Psi_i, \theta \rangle$, where $\{\Psi_i \in \mathbb{R}^d : 1 \le i \le S\}$ denotes the feature vectors, and $\theta \in \mathbb{R}^d$ denotes the weight vector to be learned. The problem of finding the optimal weight vector $\theta^*$ that best approximates $V^\pi$ can be formulated as the following optimization problem [2, 4]:

$$\theta^* = \arg\min_\theta \left\{ F(\theta) := \frac{1}{S} \sum_{i=1}^{S} \left( \langle \Psi_i, \theta \rangle - \sum_{j=1}^{S} P_{i,j}^\pi \left( r_{i,j} + \gamma \langle \Psi_j, \theta \rangle \right) \right)^2 \right\}. \tag{18}$$

Note that the above problem can be expressed as a special case of (1) by using the following identifications: For each $1 \leq i \leq S$, $n_X = n_{Y_i} = S$, $\phi_i(u) = u^2$, and

$$f_\theta(x_i, y_{ij}) = \langle \Psi_i, \theta \rangle - S \cdot P_{i,j}^\pi \big( r_{i,j} + \gamma \langle \Psi_j, \theta \rangle \big), \qquad 1 \leq i, j \leq S. \tag{19}$$

And it is also possible, although less intuitive, to rewrite (18) as a special case of (2). In existing composition optimization literature such as [4, 2], this is achieved via higher dimensional transformation, with $f_\theta : \mathbb{R}^d \to \mathbb{R}^{2S}$, $\phi_i : \mathbb{R}^{2S} \to \mathbb{R}$, $n_Y = S$, and $n_X = S$. Denote by $Q_\theta^\pi$ the Q-function:

$$Q_\theta^\pi(i) := \sum_{j=1}^{S} P_{i,j}^\pi \big( r_{i,j} + \gamma \langle \Psi_j, \theta \rangle \big), \qquad 1 \leq i \leq S.$$

Then, by defining the function $f_\theta$ and $\phi_i$ such that

$$\frac{1}{S} \sum_{j=1}^{S} f_\theta(y_j) = \Big[ \langle \Psi_1, \theta \rangle, Q_\theta^\pi(1), \ldots, \langle \Psi_S, \theta \rangle, Q_\theta^\pi(S) \Big] \tag{20}$$

$$\frac{1}{S} \sum_{i=1}^{S} \phi_i \left( \Big[ \langle \Psi_1, \theta \rangle, Q_\theta^\pi(1), \ldots, \langle \Psi_S, \theta \rangle, Q_\theta^\pi(S) \Big] \right) = \frac{1}{S} \sum_{i=1}^{S} \Big( \langle \Psi_i, \theta \rangle - Q_\theta^\pi(i) \Big)^2,$$

the problem (18) can be reformulated as (2). The reader is referred to [4, 2] for more details.

We evaluate our algorithms on two experimental settings, one with $S = 10$, $d = 5$, and another with $S = 10^4$ and $d = 10$. In each of these two cases, $\gamma$ was set to be 0.9, and both the transition probability matrix $P^\pi \in \mathbb{R}^{S \times S}$ and the feature vectors $\{\Psi_i \in \mathbb{R}^d : 1 \leq i \leq S\}$ were randomly generated. We compare the performance our algorithms SVRPDA-I and SVRPDA-II with the C-SAGA algorithm of [4], as it is the most recent composition optimization that we are aware of, and is shown to be superior to all existing algorithms on this MDP policy evaluation task [4]. The hyper-parameters for the C-SAGA algorithm were chosen as follows. For $S = 10$ case, we choose all hyper-parameters as in [4]: Mini-batch size $s = 1$ and step-size $\eta = 0.1$. For the $S = 10^4$ case, we choose mini-batch size $s = 100$, and step-size $\eta = 0.0005$ (see [4] for details on what these hyper-parameters mean). The hyper-parameters for the SVRPDA-I and SVRPDA-II are chosen as follows. For the MDP with $S = 10$, we choose $M = 150$, $\alpha_\theta = 0.1$, and $\alpha_w = 0.25$ for SVRPDA-I, and choose $M = 15$, $\alpha_\theta = 0.5$, and $\alpha_w = 1.25$ for SVRPDA-II. For the MDP with $S = 10^4$, we choose $M = 13500$, $\alpha_\theta = 0.01$ and $\alpha_w = 16 \times 10^4$ for SVRPDA-I. For SVRPDA-II, $M = 1350$, $\alpha_\theta = 0.01$ and $\alpha_w = 16 \times 10^4$.

The performance criteria was chosen to be the number of oracle calls required to achieve a certain "objective gap", defined as $F(\theta) - F(\theta^*)$. Notice that "one function call", when the function is of the form (19) *is not comparable* to one function call, when the function is defined according to (20). Due to the fact that $f_\theta$ for the C-SAGA algorithm is of dimension $2S$ (as opposed to 1 in our formulation), we count $2S$ oracle calls whenever a function of this form is called for the purpose of fair comparison. The results for MDP with $S = 10$ and $S = 10^4$ are reported in Figures 1 and 2, respectively. We observe that despite having much smaller memory requirement, SVRPDA-II has a comparable/better performance than C-SAGA, while SVRPDA-I is clearly better than C-SAGA.

Figure 1: Performance comparison of our algorithms with the C-SAGA on the MDP with $S = 10$. The performance is measured in terms of the number of oracle calls to achieve a certain objective gap.

Figure 2: Performance comparison of our algorithms with the C-SAGA on the MDP with $S = 10^4$. The performance is measured in terms of the number of oracle calls to achieve a certain objective gap.

## E    Convergence and complexity of SVRPDA-I: Proof

In this section, we derive the (non-asymptotic) bound of the SVRPDA-I algorithm and its total computation complexity. For convenience, we first repeat the saddle point formulation and the definition of several quantities below:

$$\min_{\theta} \max_{w} \frac{1}{n_X} \sum_{i=0}^{n_X-1} \frac{1}{n_{Y_i}} \sum_{j=0}^{n_{Y_i}-1} \left( \left\langle f_\theta(x_i, y_{ij}), w_i \right\rangle - \phi_i^*(w_i) \right) + g(\theta) \tag{21}$$

Also, recall the definitions of $\overline{f}_i(\theta)$ and $\overline{f}'_i(\theta)$:

$$\overline{f}_i(\theta) = \frac{1}{n_{Y_i}} \sum_{j=0}^{n_{Y_i}-1} f_\theta(x_i, y_{ij}), \quad \overline{f}'_i(\theta) = \frac{1}{n_{Y_i}} \sum_{j=0}^{n_{Y_i}-1} f'_\theta(x_i, y_{ij}) \tag{22}$$

Furthermore, we defined $L(\theta, w)$ and its gradient as

$$L(\theta, w) := \frac{1}{n_X} \sum_{i=0}^{n_X-1} \frac{1}{n_{Y_i}} \sum_{j=0}^{n_{Y_i}-1} \left( \left\langle f_\theta(x_i, y_{ij}), w_i \right\rangle - \phi_i^*(w_i) \right) \tag{23}$$

$$= \frac{1}{n_X} \sum_{i=0}^{n_X-1} \left( \left\langle \overline{f}_i(\theta), w_i \right\rangle - \phi_i^*(w_i) \right) \tag{24}$$

$$L'_\theta(\theta, w) := \frac{1}{n_X} \sum_{i=0}^{n_X-1} \overline{f}'_i(\theta) w_i \tag{25}$$

Using the above notations, the saddle point problem (21) can be rewritten as

$$\min_{\theta} \min_{w} \frac{1}{n_X} \sum_{i=0}^{n_X-1} \left( \left\langle \overline{f}_i(\theta), w_i \right\rangle - \phi_i^*(w_i) \right) + g(\theta) \tag{26}$$

## E.1 Compact Notation

Throughout this section, we introduce the following compact notation, to ease exposition of the proof: For any $\theta \in \mathbb{R}^d$, and $0 \le i \le n_X - 1$ and $0 \le j \le n_{Y_i} - 1$, we denote:

$$f_{ij}(\theta) \equiv f_\theta(x_i, y_{ij}) \tag{27}$$

Therefore, the stochastic variance reduced gradient for dual update defined in (12) of the main paper is rewritten as:

$$\delta_k^w = f_{i_k j_k}(\theta^{(k-1)}) - f_{i_k j_k}(\widetilde{\theta}) + \overline{f}_{i_k}(\widetilde{\theta}) \tag{28}$$

Similarly, the stochastic variance reduced gradient for primal update defined in (15) of the main paper is:

$$\delta_k^\theta = f'_{i'_k j'_k}(\theta^{(k-1)}) w_{i'_k}^{(k)} - f'_{i'_k j'_k}(\widetilde{\theta}) w_{i'_k}^{(k)} + L'_\theta(\widetilde{\theta}, w^{(k)}) \tag{29}$$

where we used the fact that $U_k \equiv L'_\theta(\widetilde{\theta}, w^{(k)})$. We now proceed to recall the Algorithm 1 rewritten in a simplified form, using the compact notation.

## E.2 Algorithm

Before we proceed to prove the convergence of the algorithm, we first recall the update equations of the algorithm. The following updates are at stage $s$ of the outerloop; To simplify exposition, we suppress dependency on $s$, and let $\widetilde{\theta} \equiv \widetilde{\theta}_s$ throughout.

For the dual update, at each iteration $k$, we first randomly pick an index $0 \le i_k \le n_X - 1$ at uniform, and then pick another index $0 \le j_k \le n_{Y_{i_k}} - 1$ at uniform. For the chosen $(i_k, j_k)$, we first compute the variance reduced stochastic gradient $\delta_k^w$ of $\overline{f}_i(\theta)$ using (28):

$$\delta_k^w = f_{i_k j_k}(\theta^{(k-1)}) - f_{i_k j_k}(\widetilde{\theta}) + \overline{f}_{i_k}(\widetilde{\theta})$$

Then, we update the dual variables according to the recursion (13):

$$w_i^{(k)} = \begin{cases} \underset{w_i}{\arg\min} \left[ -\left\langle \delta_k^w, \, w_i - w_i^{(k-1)} \right\rangle + \phi_i^*(w_i) + \dfrac{1}{2\alpha_w} \|w_i - w_i^{(k-1)}\|^2 \right] & \text{if } i = i_k \\ w_i^{(k-1)} & \text{if } i \neq i_k \end{cases} \tag{30}$$

For the primal update, at iteration $k$, we randomly pick another independent set of indices $(i'_k, j'_k)$ with $0 \le i'_k \le n_X - 1$ and $0 \le j'_k \le n_{Y_{i_k}} - 1$, and compute the variance reduced stochastic gradient $\delta_k^\theta$ of $L(\theta, w)$ with respect to $\theta$ using (29):

$$\delta_k^\theta = f'_{i'_k j'_k}(\theta^{(k-1)}) w_{i'_k}^{(k)} - f'_{i'_k j'_k}(\widetilde{\theta}) w_{i'_k}^{(k)} + L'_\theta(\widetilde{\theta}, w^{(k)})$$

Then, we update the primal variable $\theta$ according to the recursion (16):

$$\theta^{(k)} = \arg\min_\theta \left\{ \langle \delta_k^\theta, \theta \rangle + g(\theta) + \frac{1}{2\alpha_\theta} \|\theta - \theta^{(k-1)}\|^2 \right\} \tag{31}$$

## E.3 Assumptions

We restate the Assumptions in Section 4 here using the notation in (27) to make the reading easier:

**Assumption E.1.** *The function $g(\theta)$ is $\mu$-strongly convex in $\theta$, and each $\phi_i$ is $1/\gamma$-smooth.*

**Assumption E.2.** *The merit functions $\phi_i(u)$ are Lipschitz with a uniform constant $B_w$:*

$$|\phi_i(u) - \phi_i(u')| \le B_w \|u - u'\|, \quad \forall u, u' \in \mathbb{R}^\ell, 0 \le i \le n_X - 1.$$

**Assumption E.3.** *$f_{ij}(\theta)$ is $B_\theta$-smooth in $\theta$, and has bounded gradients with constant $B_f$: For each $0 \le i \le n_X - 1$ and $0 \le j \le n_{Y_i} - 1$,*

$$\|f'_{ij}(\theta_1) - f'_{ij}(\theta_2)\| \le B_\theta \|\theta_1 - \theta_2\|, \quad \|f'_{ij}(\theta)\| \le B_f, \quad \forall \theta, \theta_1, \theta_2 \in \mathbb{R}^d$$

**Assumption E.4.** *For each given $w$ in its domain, the function $L(\theta, w)$ defined in (8) is convex in $\theta$:*

$$L(\theta_1, w) - L(\theta_2, w) \ge \langle L'_\theta(\theta_2, w), \, \theta_1 - \theta_2 \rangle.$$

### E.4 Preliminary results

In this subsection, we introduce lemmas which lay the foundation for the proof of the main convergence result that follows. First, our proof relies on the following important lemma, which is a slightly adjusted version of Lemma 3 in [3] for our problem setting.

**Lemma E.5.** *Consider any function of the form $P(x) = f(x) + g(x)$, with $x \in \mathbb{R}^d$. Suppose $f(x)$ is linear in $x$, and $g(x)$ is $\mu_g$-strongly convex. Then, for $\alpha > 0$, the following holds for any vector $v \in \mathbb{R}^d$ and $y \in \mathbb{R}^d$:*

$$P(y) \geq P(x^{(+)}) + \frac{1}{\alpha}\langle(x - x^{(+)}), (y-x)\rangle + \langle(v - f'(x)), (x^{(+)} - y)\rangle + \frac{1}{\alpha}\|x - x^{(+)}\|^2 + \frac{\mu_g}{2}\|y - x^{(+)}\|^2$$

*where:*

$$x^{(+)} = \text{prox}_{\alpha g}\{x - \alpha v\}$$
$$= \arg\min_w \left\{g(w) + \frac{1}{2\alpha}\|w - x + \alpha v\|^2\right\}$$

*Proof.* Based on the definition of $x^{(+)}$, the optimality condition associated with the proximal operator states that there exists a sub-gradient $\xi \in \partial g(x^{(+)})$ such that

$$\frac{x^{(+)} - x + \alpha v}{\alpha} + \xi = 0 \tag{32}$$

where $\partial g(x^{(+)})$ denotes the sub-differential of $g$ at $x^{(+)}$. Next, by the linearity of $f$ and the strong convexity of $g$, we have, for any $x, y \in \mathbb{R}^d$,

$$P(y) = f(y) + g(y)$$

$$\overset{(a)}{\geq} f(x^{(+)}) + \langle f'(x), (y - x^{(+)})\rangle + g(x^{(+)}) + \langle \xi, (y - x^{(+)})\rangle + \frac{\mu_g}{2}\|y - x^{(+)}\|^2$$

$$\overset{(b)}{=} P(x^{(+)}) + \langle f'(x), (y - x^{(+)})\rangle + \langle \xi, (y - x^{(+)})\rangle + \frac{\mu_g}{2}\|y - x^{(+)}\|^2$$

$$\overset{(c)}{=} P(x^{(+)}) + \langle f'(x), (y - x^{(+)})\rangle - \frac{1}{\alpha}\langle(x^{(+)} - x + \alpha v), (y - x^{(+)})\rangle + \frac{\mu_g}{2}\|y - x^{(+)}\|^2 \tag{33}$$

$$\overset{(d)}{=} P(x^{(+)}) + \langle f'(x) - v, (y - x^{(+)})\rangle - \frac{1}{\alpha}\langle(x^{(+)} - x), (y - x^{(+)})\rangle + \frac{\mu_g}{2}\|y - x^{(+)}\|^2$$

$$\overset{(e)}{=} P(x^{(+)}) + \langle f'(x) - v, (y - x^{(+)})\rangle + \frac{1}{\alpha}\|x^{(+)} - x\|^2$$
$$- \frac{1}{\alpha}\langle(x^{(+)} - x), (y - x)\rangle + \frac{\mu_g}{2}\|y - x^{(+)}\|^2$$

where step $(a)$ follows from the linearity of $f$ and the strong convexity of the function $g$, step $(b)$ uses the definition $P(x^{(+)}) = f(x^{(+)}) + g(x^{(+)})$, step $(c)$ substitutes the expression of $\xi$ from (32), step $(d)$ rearrange the second and the third terms, and step $(e)$ completes the proof by adding and subtracting $x$ in the second inner product. $\qquad\square$

The difference between our Lemma E.5 and Lemma 3 in [3] is that our function $f(x)$ is linear (instead of being strongly convex) in $x$, which is the setting that we are mainly interested in (i.e., linear dependency on the dual variables). As a result, our $\alpha$ can be any positive number (it is constrained to be smaller than a certain positive number in Lemma 3 of [3]). This lemma is useful for deriving a bound when the update recursions are defined by a proximal mapping with an arbitrary update vector $v$. This is particularly helpful for our case as both our primal and dual updates are in proximal mapping form with the update vector $v$ being variance reduced stochastic gradient.

Next, we quote the Lemma 2 of [3] below:

**Lemma E.6.** *Let $R$ be a closed convex function on $\mathbb{R}^d$ and let $x, y \in \mathbf{dom}(R)$. Then:*

$$\|\text{prox}_R(x) - \text{prox}_R(y)\| \leq \|x - y\| \tag{34}$$

We next introduce a useful property of the conjugate function:

**Lemma E.7.** *When Assumption E.2 holds, the domain of $\phi_i^*(w_i)$, denoted as $\operatorname{domain}(\phi_i)$, satisfies*

$$\operatorname{domain}(\phi_i) \subseteq \{w_i : \|w_i\| \leq B_w\} \tag{35}$$

*That is, for any $w_i$ that satisfies $\|w_i\| > B_w$, we will have $\phi_i^*(w_i) = +\infty$. In consequence, the dual variables $w_i^{(k)}$ obtained from the dual update (30) will always be bounded by $B_w$ throughout the iterations.*

*Proof.* For any given $w_i$ that satisfies $\|w_i\| > B_w$, define $u_i^t = u_i + \frac{w_i}{\|w_i\|}t$, where $t$ is an arbitrary real scalar. Then, by the definition of conjugate function, we have

$$
\begin{aligned}
\phi_i^*(w_i) &= \sup_{u_i} \left[ \langle w_i, u_i \rangle - \phi_i(u_i) \right] \\
&\overset{(a)}{\geq} \sup_t \left[ \left\langle w_i, u_i + \frac{w_i}{\|w_i\|}t \right\rangle - \phi_i\left(u_i + \frac{w_i}{\|w_i\|}t\right) \right] \\
&= \sup_t \left[ \|w_i\|t - \phi_i\left(u_i + \frac{w_i}{\|w_i\|}t\right) + \phi_i(u_i) \right] + \langle w_i, u_i \rangle - \phi_i(u_i) \\
&\overset{(b)}{\geq} \sup_t \left[ \|w_i\|t - B_w\left\| \frac{w_i}{\|w_i\|}t \right\| \right] + \langle w_i, u_i \rangle - \phi_i(u_i) \\
&= \sup_t \left[ (\|w_i\| - B_w)t \right] + \langle w_i, u_i \rangle - \phi_i(u_i) \\
&= +\infty
\end{aligned}
\tag{36}
$$

where step (a) uses the fact that the supremum over a subset (line) is smaller, and step (b) uses the following inequality obtained from the Lipschitz property of $\phi_i(u_i)$:

$$|\phi_i(u) - \phi_i(u')| \leq B_w\|u - u'\|, \quad \forall u, u' \quad \Rightarrow \quad -\phi_i(u) + \phi_i(u') \geq -B_w\|u - u'\| \tag{37}$$

$\square$

## E.5 Dual Bound

In order to derive the bound for the dual update, we first introduce an auxiliary dummy variable $w'_{ij}$:

$$
\begin{aligned}
w'_{ij} &= \arg\min_{w_i} \left[ -\left\langle \delta_{ij}^w, w_i - w_i^{(k-1)} \right\rangle + \phi_i^*(w_i) + \frac{1}{2\alpha_w}\|w_i - w_i^{(k-1)}\|^2 \right] \\
&= \operatorname{prox}_{\alpha_w \phi_i^*}\left[ w_i^{(k-1)} - \alpha_w \delta_{ij}^w \right]
\end{aligned}
\tag{38}
$$

where,

$$\delta_{ij}^w := f_{ij}(\theta^{(k-1)}) - f_{ij}(\widetilde{\theta}) + \overline{f}_i(\widetilde{\theta}) \tag{39}$$

The variable $w'_{ij}$ can be understood as the updated value of the dual variable if $i$ and $j$ is selected.

Our analysis in this section focuses on deriving bounds for the $\|w^{(k)} - w^*\|^2$. We will first examine $\|w'_{ij} - w_i^*\|^2$ and then relate it to $\|w^{(k)} - w^*\|^2$. To begin with, for each $i$ and $j$, we have

$$
\begin{aligned}
\|w'_{ij} - w_i^*\|^2 &= \|w'_{ij} - w_i^{(k-1)} + w_i^{(k-1)} - w_i^*\|^2 \\
&= \|w'_{ij} - w_i^{(k-1)}\|^2 + \|w_i^{(k-1)} - w_i^*\|^2 + 2\langle (w'_{ij} - w_i^{(k-1)}), (w_i^{(k-1)} - w_i^*) \rangle
\end{aligned}
\tag{40}
$$

Now, we upper bound the first and the third terms in (40) together. For a given $\theta^{(k)}$ and $i$, define

$$P_{w_i}(x) := -\langle \overline{f}_i(\theta^{(k-1)}), x \rangle + \phi_i^*(x) \tag{41}$$

Note that the first part of the function is linear in $x$ and the second part of the function is $\gamma$-strongly convex (since $\phi_i$ is $1/\gamma$-smooth by Assumption E.1). Furthermore, by (38), the update rule for the dummy variables $w'_{ij}$ is defined by a proximal operator. Therefore, we can apply Lemma E.5 with $P(x) \equiv P_{w_i}(x)$ and the following identifications:

$$f(x) = -\langle \overline{f}_i(\theta^{(k-1)}), x \rangle \quad g(x) = \phi_i^*(x) \quad v = -\delta_{ij}^w \quad x = w_i^{(k-1)} \quad x^{(+)} = w'_{ij} \quad y = w_i^* \quad \alpha = \alpha_w$$

which leads to

$$-\langle \overline{f}_i(\theta^{(k-1)}), w_i^* \rangle + \phi_i^*(w_i^*) \geq -\langle \overline{f}_i(\theta^{(k-1)}), w_{ij}' \rangle + \phi_i^*(w_{ij}')$$
$$+ \frac{1}{\alpha_w} \langle (w_i^{(k-1)} - w_{ij}'), (w_i^* - w_i^{(k-1)}) \rangle$$
$$- \langle \delta_{ij}^w - \overline{f}_i(\theta^{(k-1)}), w_{ij}' - w_i^* \rangle \tag{42}$$
$$+ \frac{1}{\alpha_w} \|w_i^{(k-1)} - w_{ij}'\|^2 + \frac{\gamma}{2} \|w_i^* - w_{ij}'\|^2$$

Furthermore, by definition, since $w_i^*$ is the optimal solution to the following optimization problem,

$$w_i^* = \arg\min_{w_i} \left\{ \phi_i^*(w_i) - \langle \overline{f}_i(\theta^*), w_i \rangle \right\}$$

and by the fact that the cost function inside the above $\arg\min$ is $\gamma$-strongly convex due to $\phi_i^*(\cdot)$, we have

$$-\langle \overline{f}_i(\theta^*), w_{ij}' \rangle + \phi_i^*(w_{ij}') \geq -\langle \overline{f}_i(\theta^*), w_i^* \rangle + \phi_i^*(w_i^*) + \frac{\gamma}{2} \|w_{ij}' - w_i^*\|^2 \tag{43}$$

Adding (42) and (43) cancels the $\phi_i^*$ terms and leads to

$$\langle \overline{f}_i(\theta^{(k-1)}) - \overline{f}_i(\theta^*), w_{ij}' - w_i^* \rangle \geq \frac{1}{\alpha_w} \langle (w_i^{(k-1)} - w_{ij}'), (w_i^* - w_i^{(k-1)}) \rangle$$
$$- \langle \delta_{ij}^w - \overline{f}_i(\theta^{(k-1)}), w_{ij}' - w_i^* \rangle \tag{44}$$
$$+ \frac{1}{\alpha_w} \|w_i^{(k-1)} - w_{ij}'\|^2 + \gamma \|w_i^* - w_{ij}'\|^2$$

Multiplying both sides by $2\alpha_w$ and rearranging the terms, we obtain

$$2\alpha_w \langle \overline{f}_i(\theta^{(k-1)}) - \overline{f}_i(\theta^*), w_{ij}' - w_i^* \rangle + 2\alpha_w \langle \delta_{ij}^w - \overline{f}_i(\theta^{(k-1)}), w_{ij}' - w_i^* \rangle - 2\alpha_w \gamma \|w_{ij}' - w_i^*\|^2$$
$$\geq 2\langle (w_i^{(k-1)} - w_{ij}'), (w_i^* - w_i^{(k-1)}) \rangle\rangle + 2\|w_i^{(k-1)} - w_{ij}'\|^2 \tag{45}$$

Now, observe that inequality (45) could be used as an uppper bound for the first and third terms on the right hand side of (40). Using this, (40) becomes:

$$\|w_{ij}' - w_i^*\|^2$$
$$= \|w_i^{(k-1)} - w_i^*\|^2 + \|w_{ij}' - w_i^{(k-1)}\|^2 + 2\langle (w_{ij}' - w_i^{(k-1)}), (w_i^{(k-1)} - w_i^*) \rangle$$
$$\stackrel{(a)}{=} \|w_i^{(k-1)} - w_i^*\|^2 - \|w_{ij}' - w_i^{(k-1)}\|^2 + 2\|w_{ij}' - w_i^{(k-1)}\|^2 + 2\langle (w_{ij}' - w_i^{(k-1)}), (w_i^{(k-1)} - w_i^*) \rangle$$
$$\leq \|w_i^{(k-1)} - w_i^*\|^2 - \|w_{ij}' - w_i^{(k-1)}\|^2 + 2\alpha_w \langle \overline{f}_i(\theta^{(k-1)}) - \overline{f}_i(\theta^*), w_{ij}' - w_i^* \rangle$$
$$+ 2\alpha_w \langle \delta_{ij}^w - \overline{f}_i(\theta^{(k-1)}), w_{ij}' - w_i^* \rangle - 2\alpha_w \gamma \|w_{ij}' - w_i^*\|^2 \tag{46}$$

where step (a) added and subtracted a $\|w_i^{(k-1)} - w_{ij}'\|^2$ in order to apply (45) in the following inequality. Dividing both sides by $2\alpha_w$ and combining common terms, we get the following bound:

$$\left( \frac{1}{2\alpha_w} + \gamma \right) \|w_{ij}' - w_i^*\|^2 \leq \frac{1}{2\alpha_w} \|w_i^{(k-1)} - w_i^*\|^2 - \frac{1}{2\alpha_w} \|w_{ij}' - w_i^{(k-1)}\|^2$$
$$+ \langle \overline{f}_i(\theta^{(k-1)}) - \overline{f}_i(\theta^*), w_{ij}' - w_i^* \rangle + \langle \delta_{ij}^w - \overline{f}_i(\theta^{(k-1)}), w_{ij}' - w_i^* \rangle \tag{47}$$

Next, we will bound the last term in (47). Consider the full batch dual ascent algorithm. In this case, for each $0 \leq i \leq n_X - 1$, the update rule is given by:

$$\overline{w}_i^{(k)} = \arg\min_{w_i} \left[ -\langle \overline{f}_i(\theta^{(k-1)}), w_i \rangle + \phi_i^*(w_i) + \frac{1}{2\alpha_w} \|w_i - w_i^{(k-1)}\|^2 \right]$$
$$= \mathrm{prox}_{\alpha_w \phi_i^*} \left[ w_i^{(k-1)} - \alpha_w \overline{f}_i(\theta^{(k-1)}) \right] \tag{48}$$

The above update rule will only be used for analysis. Considering the last term in the right hand side of (47); we have:

$$
\begin{aligned}
\left\langle \delta_{ij}^w - \overline{f}_i(\theta^{(k-1)}), w_{ij}' - w_i^* \right\rangle &= \left\langle \delta_{ij}^w - \overline{f}_i(\theta^{(k-1)}), w_{ij}' - \overline{w}_i^{(k)} \right\rangle + \left\langle \delta_{ij}^w - \overline{f}_i(\theta^{(k-1)}), \overline{w}_i^{(k)} - w_i^* \right\rangle \\
&\overset{(a)}{\leq} \|\delta_{ij}^w - \overline{f}_i(\theta^{(k-1)})\| \cdot \|w_{ij}' - \overline{w}_i^{(k)}\| + \left\langle \delta_{ij}^w - \overline{f}_i(\theta^{(k-1)}), \overline{w}_i^{(k)} - w_i^* \right\rangle \\
&\overset{(b)}{\leq} \alpha_w \|\delta_{ij}^w - \overline{f}_i(\theta^{(k-1)})\|^2 + \left\langle \delta_{ij}^w - \overline{f}_i(\theta^{(k-1)}), \overline{w}_i^{(k)} - w_i^* \right\rangle
\end{aligned}
$$

(49)

where $(a)$ uses Cauchy-Schwarz inequality, and step $(b)$ substitutes the proximal expressions of $w_{ij}'$ in (38) and $\overline{w}_i^{(k)}$ in (49) followed by Lemma E.6. Averaging both sides of the inequality over all $0 \leq j \leq n_{Y_i} - 1$ and using the fact that the average of the second term on the right hand side of (49) is zero, we get:

$$
\frac{1}{n_{Y_i}} \sum_{j=0}^{n_{Y_i}-1} \left[ \left\langle \delta_{ij}^w - \overline{f}_i(\theta^{(k-1)}), w_{ij}' - w_i^* \right\rangle \right]
$$

$$
\leq \frac{\alpha_w}{n_{Y_i}} \sum_{j=0}^{n_{Y_i}-1} \left[ \|\delta_{ij}^w - \overline{f}_i(\theta^{(k-1)})\|^2 \right]
$$

$$
= \frac{\alpha_w}{n_{Y_i}} \sum_{j=0}^{n_{Y_i}-1} \left[ \|f_{ij}(\theta^{(k-1)}) - f_{ij}(\widetilde{\theta}) + \overline{f}_i(\widetilde{\theta}) - \overline{f}_i(\theta^{(k-1)})\|^2 \right]
$$

$$
\overset{(a)}{=} \frac{\alpha_w}{n_{Y_i}} \sum_{j=0}^{n_{Y_i}-1} \left\| \left(1 - \frac{1}{n_{Y_i}}\right) f_{ij}(\theta^{(k-1)}) \right.
$$
$$
\left. - \left(1 - \frac{1}{n_{Y_i}}\right) f_{ij}(\widetilde{\theta}) - \left( \overline{f}_i(\theta^{(k-1)}) - \overline{f}_i(\widetilde{\theta}) - \frac{1}{n_{Y_i}} f_{ij}(\theta^{(k-1)}) + \frac{1}{n_{Y_i}} f_{ij}(\widetilde{\theta}) \right) \right\|^2
$$

$$
\overset{(b)}{\leq} \frac{2\alpha_w}{n_{Y_i}} \sum_{j=0}^{n_{Y_i}-1} \left[ \left\| \left(1 - \frac{1}{n_{Y_i}}\right) f_{ij}(\theta^{(k-1)}) - \left(1 - \frac{1}{n_{Y_i}}\right) f_{ij}(\widetilde{\theta}) \right\|^2 \right.
$$
$$
\left. + \left\| \overline{f}_i(\theta^{(k-1)}) - \overline{f}_i(\widetilde{\theta}) - \frac{1}{n_{Y_i}} f_{ij}(\theta^{(k-1)}) + \frac{1}{n_{Y_i}} f_{ij}(\widetilde{\theta}) \right\|^2 \right]
$$

$$
= \frac{2\alpha_w}{n_{Y_i}} \sum_{j=0}^{n_{Y_i}-1} \left[ \left\| \left(1 - \frac{1}{n_{Y_i}}\right) f_{ij}(\theta^{(k-1)}) - \left(1 - \frac{1}{n_{Y_i}}\right) f_{ij}(\widetilde{\theta}) \right\|^2 \right.
$$
$$
\left. + \left\| \frac{n_{Y_i} - 1}{n_{Y_i}} \frac{1}{n_{Y_i} - 1} \sum_{\ell \neq j} \left[ f_{i\ell}(\theta^{k-1}) - f_{i\ell}(\widetilde{\theta}) \right] \right\|^2 \right]
$$

$$
\overset{(c)}{\leq} \frac{2\alpha_w}{n_{Y_i}} \sum_{j=0}^{n_{Y_i}-1} \left[ \left\| \left(1 - \frac{1}{n_{Y_i}}\right) f_{ij}(\theta^{(k-1)}) - \left(1 - \frac{1}{n_{Y_i}}\right) f_{ij}(\widetilde{\theta}) \right\|^2 \right.
$$
$$
\left. + \left( \frac{n_{Y_i} - 1}{n_{Y_i}} \right)^2 \frac{1}{n_{Y_i} - 1} \sum_{\ell \neq j} \left\| \left[ f_{i\ell}(\theta^{k-1}) - f_{i\ell}(\widetilde{\theta}) \right] \right\|^2 \right]
$$

$$
\overset{(d)}{\leq} 2\alpha_w \left[ \left(1 - \frac{1}{n_{Y_i}}\right)^2 B_f^2 \|\theta^{(k-1)} - \widetilde{\theta}\|^2 + \frac{n_{Y_i} - 1}{n_{Y_i}^2} B_f^2 \|\theta^{(k-1)} - \widetilde{\theta}\|^2 \right]
$$

$$
= 2\alpha_w B_f^2 \left(1 - \frac{1}{n_{Y_i}}\right) \|\theta^{(k-1)} - \widetilde{\theta}\|^2
$$

(50)

where step (a) follows by adding and subtracting $\frac{1}{n_{Y_i}} \left( f_{ij}(\theta^{(k-1)}) - f_{ij}(\widetilde{\theta}) \right)$, step (b) uses $\|a + b\|^2 \leq 2\|a\|^2 + 2\|b\|^2$, step (c) applies Jensen's inequality to the second term, step (d) applies Assumption E.3 (uniformly bounded gradients implies the uniform Lipschitz continuity of the functions $f_{ij}$) to both

terms. Averaging both sides of (47) over all $0 \leq j \leq n_{Y_i} - 1$, and using (50), we obtain:

$$
\left(\frac{1}{2\alpha_w} + \gamma\right) \frac{1}{n_{Y_i}} \sum_{j=0}^{n_{Y_i}-1} \|w'_{ij} - w_i^*\|^2 \leq \frac{1}{2\alpha_w} \|w_i^{(k-1)} - w_i^*\|^2 - \frac{1}{2\alpha_w} \frac{1}{n_{Y_i}} \sum_{j=0}^{n_{Y_i}-1} \|w'_{ij} - w_i^{(k-1)}\|^2
$$

$$
+ \frac{1}{n_{Y_i}} \sum_{j=0}^{n_{Y_i}-1} \langle \overline{f}_i(\theta^{(k-1)}) - \overline{f}_i(\theta^*), w'_{ij} - w_i^* \rangle + 2\alpha_w B_f^2 \left(1 - \frac{1}{n_{Y_i}}\right) \|\theta^{(k-1)} - \widetilde{\theta}\|^2 \tag{51}
$$

Finally, we relate the bound for $w'_{ij}$ back to the bound for the dual variable $w_i^{(k)}$. Recall that, for each $w_i$, there is a probability $1/n_X$ that it will be selected and updated, and a probability of $(n_X - 1)/n_X$ that it will be kept the same as $w_i^{(k-1)}$. Furthermore, conditioned on the fact that $w_i$ is selected, it will be updated to $w'_{ij}$ with probability $1/n_{Y_i}$. Therefore, for each $w_i$, there is a probability $1/n_X n_{Y_i}$ that it will be updated to $w'_{ij}$ for $j = 0, \ldots, n_{Y_i} - 1$, and a probability of $(n_X - 1)/n_X$ that it remains the same. Therefore, letting $\mathcal{F}_k$ denote the filtration of all events upto the beginning of iteration $k$ (before the dual update step), we have:

$$
\mathsf{E}\{w_i^{(k)} \mid \mathcal{F}_k\} = \frac{1}{n_X} \frac{1}{n_{Y_i}} \sum_{j=0}^{n_{Y_i}-1} w'_{ij} + \frac{n_X - 1}{n_X} w_i^{(k-1)}
$$

$$
\mathsf{E}\{w_i^{(k)} - w_i^* \mid \mathcal{F}_k\} = \frac{1}{n_X} \frac{1}{n_{Y_i}} \sum_{j=0}^{n_{Y_i}-1} (w'_{ij} - w_i^*) + \frac{n_X - 1}{n_X} (w_i^{(k-1)} - w_i^*)
$$

$$
\mathsf{E}\{\|w_i^{(k)} - w_i^*\|^2 \mid \mathcal{F}_k\} = \frac{1}{n_X} \frac{1}{n_{Y_i}} \sum_{j=0}^{n_{Y_i}-1} \|w'_{ij} - w_i^*\|^2 + \frac{n_X - 1}{n_X} \|w_i^{(k-1)} - w_i^*\|^2 \tag{52}
$$

$$
\mathsf{E}\{\|w_i^{(k)} - w_i^{(k-1)}\|^2 \mid \mathcal{F}_k\} = \frac{1}{n_X} \frac{1}{n_{Y_i}} \sum_{j=0}^{n_{Y_i}-1} \|w'_{ij} - w_i^{(k-1)}\|^2
$$

Using (52) in (51), we obtain:

$$
\left(\frac{1}{2\alpha_w} + \gamma\right) \left(n_X \mathsf{E}\{\|w_i^{(k)} - w_i^*\|^2 \mid \mathcal{F}_k\} - (n_X - 1)\|w_i^{(k-1)} - w_i^*\|^2\right)
$$

$$
\leq \frac{1}{2\alpha_w} \|w_i^{(k-1)} - w_i^*\|^2 - \frac{n_X}{2\alpha_w} \mathsf{E}\{\|w_i^{(k)} - w_i^{(k-1)}\|^2 \mid \mathcal{F}_k\}
$$

$$
+ 2\alpha_w B_f^2 \left(1 - \frac{1}{n_{Y_i}}\right) \|\theta^{(k-1)} - \widetilde{\theta}\|^2 \tag{53}
$$

$$
+ n_X \mathsf{E}\left\{\langle \overline{f}_i(\theta^{(k-1)}) - \overline{f}_i(\theta^*), w_i^{(k)} - w_i^* \rangle \mid \mathcal{F}_k\right\}
$$

$$
- (n_X - 1)\left\{\langle \overline{f}_i(\theta^{(k-1)}) - \overline{f}_i(\theta^*), w_i^{(k-1)} - w_i^* \rangle\right\}
$$

Summing both sides of (53) over $0 \leq i \leq n_X - 1$, using the fact that $\|w^{(k)} - w^*\|^2 = \sum_{i=0}^{n_X-1} \|w_i^{(k)} - w_i^*\|^2$, and then dividing by $n_X$, we get

$$
\left(\frac{1}{2\alpha_w} + \gamma\right) \left(\mathsf{E}\{\|w^{(k)} - w^*\|^2 \mid \mathcal{F}_k\} - \frac{n_X - 1}{n_X} \|w^{(k-1)} - w^*\|^2\right)
$$

$$
\leq \frac{1}{2\alpha_w n_X} \|w^{(k-1)} - w^*\|^2 - \frac{1}{2\alpha_w} \mathsf{E}\{\|w^{(k)} - w^{(k-1)}\|^2 \mid \mathcal{F}_k\}
$$

$$
+ 2\alpha_w B_f^2 \left(1 - \overline{1/n_Y}\right) \|\theta^{(k-1)} - \widetilde{\theta}\|^2 \tag{54}
$$

$$
+ n_X \mathsf{E}\left\{L(\theta^{(k-1)}, w^{(k)} - w^*) - L(\theta^*, w^{(k)} - w^*) \mid \mathcal{F}_k\right\}
$$

$$
- (n_X - 1)\left\{L(\theta^{(k-1)}, w^{(k-1)} - w^*) - L(\theta^*, w^{(k-1)} - w^*)\right\}
$$

where, we have used the notation:

$$\overline{1/n_Y} := \frac{1}{n_X} \sum_{i=0}^{n_X - 1} 1/n_{Y_i} \tag{55}$$

Rearranging and combining the common terms, we obtain the final dual bound:

$$\left( \frac{1}{2\alpha_w} + \gamma \right) \mathsf{E}\{\|w^{(k)} - w^*\|^2 \mid \mathcal{F}_k\} + \frac{1}{2\alpha_w} \mathsf{E}\{\|w^{(k)} - w^{(k-1)}\|^2 \mid \mathcal{F}_k\}$$

$$\leq \left( \frac{1}{2\alpha_w} + \frac{\gamma(n_X - 1)}{n_X} \right) \|w^{(k-1)} - w^*\|^2 + 2\alpha_w B_f^2 \left(1 - \overline{1/n_Y}\right) \|\theta^{(k-1)} - \widetilde{\theta}\|^2 \tag{56}$$

$$+ \mathsf{E}\Big\{ L(\theta^{(k-1)}, w^{(k)} - w^*) - L(\theta^*, w^{(k)} - w^*) \mid \mathcal{F}_k \Big\}$$

$$+ (n_X - 1)\mathsf{E}\Big\{ L(\theta^{(k-1)}, w^{(k)} - w^{(k-1)}) - L(\theta^*, w^{(k)} - w^{(k-1)}) \mid \mathcal{F}_k \Big\}$$

Note that the above bound still have terms related to $L(\cdot, \cdot)$. We will combine these terms together with the $L$ terms in the primal bound, and then bound them all together thereafter.

## E.6  Primal Bound

Now we proceed to derive the bound for the primal variable $\theta$. Specifically, we will focus on examining $\|\theta^{(k)} - \theta^*\|^2$, which can be written as

$$\begin{aligned} \|\theta^{(k)} - \theta^*\|^2 &= \|\theta^{(k)} - \theta^{(k-1)} + \theta^{(k-1)} - \theta^*\|^2 \\ &= \|\theta^{(k)} - \theta^{(k-1)}\|^2 + \|\theta^{(k-1)} - \theta^*\|^2 + 2\langle(\theta^{(k)} - \theta^{(k-1)}), (\theta^{(k-1)} - \theta^*)\rangle \end{aligned} \tag{57}$$

Similar to the dual bound, we now bound the first term and the third term together. Introduce the following function of $x$ (for fixed $\theta^{(k-1)}$ and $w^{(k)}$):

$$P_\theta(x) := \big\langle L'_\theta(\theta^{(k-1)}, w^{(k)}), x \big\rangle + g(x) \tag{58}$$

The first part of the function is linear in $x$ (and hence convex), and the second part of the function is $\mu$-strongly convex (Assumption E.1). Recall the primal update rule in (31), which can be written in the following proximal mapping form:

$$\begin{aligned} \theta^{(k)} &= \arg\min_\theta \left\{ \langle \delta_k^\theta, \theta \rangle + g(\theta) + \frac{1}{2\alpha_\theta} \|\theta - \theta^{(k-1)}\|^2 \right\} \\ &= \text{prox}_{\alpha_\theta g} \left\{ \theta^{(k-1)} - \alpha_\theta \delta_k^\theta \right\} \end{aligned} \tag{59}$$

We now apply Lemma E.5 with $P(x) \equiv P_\theta(x)$ and the following identifications:

$$f(x) = \big\langle L'_\theta(\theta^{(k-1)}, w^{(k)}), x \big\rangle \quad g(x) = g(x) \quad v = \delta_k^\theta \quad x = \theta^{(k-1)} \quad x^{(+)} = \theta^{(k)} \quad y = \theta^* \quad \alpha = \alpha_\theta$$

which leads to

$$\begin{aligned} \big\langle L'_\theta(\theta^{(k-1)}, w^{(k)}), \theta^* \big\rangle + g(\theta^*) \geq &\big\langle L'_\theta(\theta^{(k-1)}, w^{(k)}), \theta^{(k)} \big\rangle + g(\theta^{(k)}) \\ &+ \frac{1}{\alpha_\theta} \langle(\theta^{(k-1)} - \theta^{(k)}), (\theta^* - \theta^{(k-1)})\rangle \\ &- \langle(\delta_k^\theta - L'_\theta(\theta^{(k-1)}, w^{(k)})), (\theta^{(k)} - \theta^*)\rangle \\ &+ \frac{1}{\alpha_\theta} \|\theta^{(k-1)} - \theta^{(k)}\|^2 + \frac{\mu}{2} \|\theta^* - \theta^{(k)}\|^2 \end{aligned} \tag{60}$$

Rearranging the terms in the above inequality, we obtain

$$\|\theta^{(k-1)} - \theta^{(k)}\|^2 + \langle \theta^{(k)} - \theta^{(k-1)}, \theta^{(k-1)} - \theta^*\rangle$$
$$\leq \alpha_\theta \langle L'_\theta(\theta^{(k-1)}, w^{(k)}), \theta^*\rangle + \alpha_\theta g(\theta^*) - \alpha_\theta g(\theta^{(k)}) - \alpha_\theta \langle L'_\theta(\theta^{(k-1)}, w^{(k)}), \theta^{(k)}\rangle$$

$$+ \alpha_\theta \langle \delta_k^\theta - L'(\theta^{(k-1)}, w^{(k)}), \theta^{(k)} - \theta^* \rangle - \frac{\alpha_\theta \mu}{2} \|\theta^* - \theta^{(k)}\|^2 \tag{61}$$

Using (61) to bound the first and the third term in (57), we obtain:

$$
\begin{aligned}
\|\theta^{(k)} - \theta^*\|^2 &= \|\theta^{(k)} - \theta^{(k-1)}\|^2 + \|\theta^{(k-1)} - \theta^*\|^2 + 2\langle (\theta^{(k)} - \theta^{(k-1)}), (\theta^{(k-1)} - \theta^*) \rangle \\
&\overset{(a)}{=} \|\theta^{(k)} - \theta^{(k-1)}\|^2 - \|\theta^{(k-1)} - \theta^*\|^2 + 2\|\theta^{(k-1)} - \theta^*\|^2 \\
&\quad + 2\langle (\theta^{(k)} - \theta^{(k-1)}), (\theta^{(k-1)} - \theta^*) \rangle \\
&\leq \|\theta^{(k-1)} - \theta^*\|^2 - \|\theta^{(k)} - \theta^{(k-1)}\|^2 - \alpha_\theta \mu \|\theta^{(k)} - \theta^*\|^2 \\
&\quad + 2\alpha_\theta \Big[ \Big\langle L'_\theta(\theta^{(k-1)}, w^{(k)}), \theta^* \Big\rangle + g(\theta^*) - \Big\langle L'_\theta(\theta^{(k-1)}, w^{(k)}), \theta^{(k)} \Big\rangle - g(\theta^{(k)}) \Big] \\
&\quad + 2\alpha_\theta \langle (\delta_k^\theta - L'_\theta(\theta^{(k-1)}, w^{(k)})), (\theta^{(k)} - \theta^*) \rangle
\end{aligned}
\tag{62}
$$

where step (a) subtracts and adds the second term. Furthermore, note that $\theta^*$ is the optimal solution to the following optimization problem:

$$\theta^* = \arg\min_\theta \Big[ L(\theta, w^*) + g(\theta) \Big]$$

which implies (from the fact that $g$ is $\mu$-strongly convex):

$$L(\theta^{(k)}, w^*) + g(\theta^{(k)}) \geq L(\theta^*, w^*) + g(\theta^*) + \frac{\mu}{2}\|\theta^{(k)} - \theta^*\|^2 \tag{63}$$

Multiplying both sides of the above inequality by $2\alpha_\theta$ and then adding it to (62), we obtain:

$$
\begin{aligned}
(1 + 2\alpha_\theta \mu)\|\theta^{(k)} - \theta^*\|^2 &\leq \|\theta^{(k-1)} - \theta^*\|^2 - \|\theta^{(k)} - \theta^{(k-1)}\|^2 \\
&\quad + 2\alpha_\theta \Big[ L(\theta^{(k)}, w^*) - L(\theta^*, w^*) \Big] \\
&\quad + 2\alpha_\theta \Big[ \Big\langle L'_\theta(\theta^{(k-1)}, w^{(k)}), \theta^* \Big\rangle - \Big\langle L'_\theta(\theta^{(k-1)}, w^{(k)}), \theta^{(k)} \Big\rangle \Big] \\
&\quad + 2\alpha_\theta \langle (\delta_k^\theta - L'_\theta(\theta^{(k-1)}, w^{(k)})), (\theta^{(k)} - \theta^*) \rangle
\end{aligned}
\tag{64}
$$

Next, we bound the last term in (64). To this end, we first introduce the following auxiliary variable $\overline{\theta}^{(k)}$, which is the updated primal variable if *full batch gradient* were used:

$$
\begin{aligned}
\overline{\theta}^{(k)} &= \arg\min_\theta \Big[ \Big\langle L'_\theta(\theta^{(k-1)}, w^{(k)}), \theta \Big\rangle + g(\theta) + \frac{1}{2\alpha_\theta}\|\theta - \theta^{(k-1)}\|^2 \Big] \\
&= \mathrm{prox}_{\alpha_\theta g}\Big\{ \theta^{(k-1)} - \alpha_\theta L'_\theta(\theta^{(k-1)}, w^{(k)}) \Big\}
\end{aligned}
\tag{65}
$$

Note that both (59) and (65) are written in proximal mapping form. We now bound the last term (64):

$$
\begin{aligned}
&\langle (\delta_k^\theta - L'_\theta(\theta^{(k-1)}, w^{(k)})), (\theta^{(k)} - \theta^*) \rangle \\
&\overset{(a)}{=} \langle (\delta_k^\theta - L'_\theta(\theta^{(k-1)}, w^{(k)})), (\theta^{(k)} - \overline{\theta}^{(k)}) \rangle + \langle (\delta_k^\theta - L'_\theta(\theta^{(k-1)}, w^{(k)})), (\overline{\theta}^{(k)} - \theta^*) \rangle \\
&\overset{(b)}{\leq} \|\delta_k^\theta - L'_\theta(\theta^{(k-1)}, w^{(k)})\| \cdot \|\theta^{(k)} - \overline{\theta}^{(k)}\| + \langle (\delta_k^\theta - L'_\theta(\theta^{(k-1)}, w^{(k)})), (\overline{\theta}^{(k)} - \theta^*) \rangle \\
&\overset{(c)}{\leq} \alpha_\theta \|\delta_k^\theta - L'_\theta(\theta^{(k-1)}, w^{(k)})\|^2 + \langle (\delta_k^\theta - L'_\theta(\theta^{(k-1)}, w^{(k)})), (\overline{\theta}^{(k)} - \theta^*) \rangle
\end{aligned}
\tag{66}
$$

where step (a) adds and subtracts $\overline{\theta}^{(k)}$, step (b) uses Cauchy-Schwartz inequality, and step (c) substitutes (59) and (65) and then applies Lemma E.6. Let $\mathcal{F}_k^{(+)}$ denote the filtration of all events up to and including the dual update in the $k$-th iteration. Applying expectation to both sides of (66) conditioned on $\mathcal{F}_k^{(+)}$, we have:

$$
\begin{aligned}
&\mathsf{E}\Big\{ \langle (\delta_k^\theta - L'_\theta(\theta^{(k-1)}, w^{(k)})), (\theta^{(k)} - \theta^*) \rangle \big| \mathcal{F}_k^{(+)} \Big\} \\
&\leq \alpha_\theta \mathsf{E}\Big\{ \|\delta_k^\theta - L'_\theta(\theta^{(k-1)}, w^{(k)})\|^2 \big| \mathcal{F}_k^{(+)} \Big\} + \mathsf{E}\Big\{ \langle \delta_k^\theta - L'_\theta(\theta^{(k-1)}, w^{(k)}), \overline{\theta}^{(k)} - \theta^* \rangle \big| \mathcal{F}_k^{(+)} \Big\} \\
&\overset{(a)}{=} \alpha_\theta \mathsf{E}\Big\{ \|\delta_k^\theta - L'_\theta(\theta^{(k-1)}, w^{(k)})\|^2 \big| \mathcal{F}_k^{(+)} \Big\} + \Big\langle \mathsf{E}[\delta_k^\theta | \mathcal{F}_k^{(+)}] - L'_\theta(\theta^{(k-1)}, w^{(k)}), \overline{\theta}^{(k)} - \theta^* \Big\rangle
\end{aligned}
$$

$$\overset{(b)}{=} \alpha_\theta \mathsf{E}\Big\{ \|\delta_k^\theta - L_\theta'(\theta^{(k-1)}, w^{(k)})\|^2 \big| \mathcal{F}_k^{(+)} \Big\} \tag{67}$$

where step $(a)$ uses the fact that $\overline{\theta}^{(k)}$, $\theta^{(k-1)}$ and $w^{(k)}$ are deterministic conditioned on $\mathcal{F}_k^{(+)}$, and step $(b)$ uses the fact that the conditional expectation of $\delta_k^\theta$ is the batch gradient.

We will now upper bound the right hand side of (67). To this end, we have:

$$
\begin{aligned}
&\mathsf{E}\Big\{ \|\delta_k^\theta - L_\theta'(\theta^{(k-1)}, w^{(k)})\|^2 \big| \mathcal{F}_k^{(+)} \Big\} \\
&\overset{(a)}{=} \mathsf{E}\Big\{ \|f_{i_k j_k}'(\theta^{(k-1)}) w_{i_k}^{(k)} - f_{i_k j_k}'(\widetilde{\theta}) w_{i_k}^{(k)} + L_\theta'(\widetilde{\theta}, w^{(k)}) - L_\theta'(\theta^{(k-1)}, w^{(k)})\|^2 \big| \mathcal{F}_k^{(+)} \Big\} \\
&\overset{(b)}{\leq} \mathsf{E}\Big\{ \|f_{i_k j_k}'(\theta^{(k-1)}) w_{i_k}^{(k)} - f_{i_k j_k}'(\widetilde{\theta}) w_{i_k}^{(k)}\|^2 \big| \mathcal{F}_k^{(+)} \Big\} \\
&\overset{(c)}{\leq} B_w^2 \mathsf{E}\Big\{ \|f_{i_k j_k}'(\theta^{(k-1)}) - f_{i_k j_k}'(\widetilde{\theta})\|^2 \big| \mathcal{F}_k^{(+)} \Big\} \\
&\overset{(d)}{\leq} B_w^2 B_\theta^2 \|\theta^{(k-1)} - \widetilde{\theta}\|^2
\end{aligned}
\tag{68}
$$

where step $(a)$ uses the definition of $\delta_k^\theta$, step $(b)$ uses $\mathsf{E}[X - \mathsf{E}[X]]^2 \leq \mathsf{E}[X^2]$, step $(c)$ uses Lemma E.7, and step $(d)$ uses the Lipschitz continuity of the gradients (Assumption E.3). Substituting (68) into (67), we get:

$$\mathsf{E}\Big\{ \langle (\delta_{i_k' j_k'}^\theta - L_\theta'(\theta^{(k-1)}, w^{(k)})), (\theta^{(k)} - \theta^*) \rangle \big| \mathcal{F}_k^{(+)} \Big\} \leq \alpha_\theta B_w^2 B_\theta^2 \|\theta^{(k-1)} - \widetilde{\theta}\|^2 \tag{69}$$

Finally, substituting (69) into (64) and then further applying expectation conditioned on $\mathcal{F}_k$, we obtain:

$$
\begin{aligned}
(1 + 2\alpha_\theta \mu) \mathsf{E}\Big\{ \|\theta^{(k)} - \theta^*\|^2 \big| \mathcal{F}_k \Big\} \leq\ & \|\theta^{(k-1)} - \theta^*\|^2 - \mathsf{E}\Big\{ \|\theta^{(k)} - \theta^{(k-1)}\|^2 \big| \mathcal{F}_k \Big\} \\
& + 2\alpha_\theta \mathsf{E}\Big\{ \big[ L(\theta^{(k)}, w^*) - L(\theta^*, w^*) \big] \big| \mathcal{F}_k \Big\} \\
& + 2\alpha_\theta \mathsf{E}\Big\{ \big[ \big\langle L_\theta'(\theta^{(k-1)}, w^{(k)}), \theta^* \big\rangle - \big\langle L_\theta'(\theta^{(k-1)}, w^{(k)}), \theta^{(k)} \big\rangle \big] \big| \mathcal{F}_k \Big\} \\
& + 2\alpha_\theta^2 B_w^2 B_\theta^2 \|\theta^{(k-1)} - \widetilde{\theta}\|^2
\end{aligned}
\tag{70}
$$

Dividing both sides by $2\alpha_\theta$ and combining common terms, we obtain the final bound for the primal variable:

$$
\begin{aligned}
&\Big( \frac{1}{2\alpha_\theta} + \mu \Big) \mathsf{E}\Big\{ \|\theta^{(k)} - \theta^*\|^2 \big| \mathcal{F}_k \Big\} + \frac{1}{2\alpha_\theta} \mathsf{E}\Big\{ \|\theta^{(k)} - \theta^{(k-1)}\|^2 \big| \mathcal{F}_k \Big\} \\
&\leq \frac{1}{2\alpha_\theta} \|\theta^{(k-1)} - \theta^*\|^2 + \alpha_\theta B_w^2 B_\theta^2 \|\theta^{(k-1)} - \widetilde{\theta}\|^2 + \mathsf{E}\Big\{ \big[ L(\theta^{(k)}, w^*) - L(\theta^*, w^*) \big] \big| \mathcal{F}_k \Big\} \\
&\quad + \mathsf{E}\Big\{ \big[ \big\langle L_\theta'(\theta^{(k-1)}, w^{(k)}), \theta^* \big\rangle - \big\langle L_\theta'(\theta^{(k-1)}, w^{(k)}), \theta^{(k)} \big\rangle \big] \big| \mathcal{F}_k \Big\}
\end{aligned}
\tag{71}
$$

### E.7 Convergence for Option I

Based on the derived primal and dual bounds above, we now proceed to prove the convergence of SVRPDA-I with Option I: updating $\widetilde{\theta}$ using the most recent $\theta^{(k)}$ (see Algorithm 1).

Adding (56) and (71) we obtain the total bound for the primal and dual variable updates:

$$
\left(\frac{1}{2\alpha_w} + \gamma\right) \mathsf{E}\{\|w^{(k)} - w^*\|^2 \mid \mathcal{F}_k\} + \frac{1}{2\alpha_w} \mathsf{E}\{\|w^{(k)} - w^{(k-1)}\|^2 \mid \mathcal{F}_k\}
$$

$$
\left(\frac{1}{2\alpha_\theta} + \mu\right) \mathsf{E}\left\{\|\theta^{(k)} - \theta^*\|^2 \mid \mathcal{F}_k\right\} + \frac{1}{2\alpha_\theta} \mathsf{E}\left\{\|\theta^{(k)} - \theta^{(k-1)}\|^2 \mid \mathcal{F}_k\right\}
$$

$$
\leq \left(\frac{1}{2\alpha_w} + \frac{\gamma(n_X - 1)}{n_X}\right) \|w^{(k-1)} - w^*\|^2 + \frac{1}{2\alpha_\theta} \|\theta^{(k-1)} - \theta^*\|^2
$$

$$
+ \left(2\alpha_w B_f^2 \left(1 - \overline{1/n_Y}\right) + \alpha_\theta B_w^2 B_\theta^2\right) \|\theta^{(k-1)} - \widetilde{\theta}\|^2
$$

$$
+ \mathsf{E}\left\{ L(\theta^{(k-1)}, w^{(k)} - w^*) - L(\theta^*, w^{(k)} - w^*) \mid \mathcal{F}_k \right\}
$$

$$
+ \mathsf{E}\left\{ \left[ L(\theta^{(k)}, w^*) - L(\theta^*, w^*) \right] \mid \mathcal{F}_k \right\} + \mathsf{E}\left\{ \left[ \left\langle L_\theta'(\theta^{(k-1)}, w^{(k)}), \theta^* \right\rangle - \left\langle L_\theta'(\theta^{(k-1)}, w^{(k)}), \theta^{(k)} \right\rangle \right] \mid \mathcal{F}_k \right\}
$$

$$
+ (n_X - 1)\mathsf{E}\left\{ L(\theta^{(k-1)}, w^{(k)} - w^{(k-1)}) - L(\theta^*, w^{(k)} - w^{(k-1)}) \mid \mathcal{F}_k \right\}
\tag{72}
$$

Next, we need to upper bound the $L$ terms on the right-hand side of the above inequality. To this end, we first show the following inequality:

$$
- L(\theta^*, w^{(k)} - w^*) + L(\theta^{(k)}, w^*) - L(\theta^*, w^*) + \left\langle L_\theta'(\theta^{(k-1)}, w^{(k)}), \theta^* \right\rangle
$$

$$
- \left\langle L_\theta'(\theta^{(k-1)}, w^{(k)}), \theta^{(k)} \right\rangle + L(\theta^{(k)}, w^{(k)} - w^*)
$$

$$
= -L(\theta^*, w^{(k)}) + L(\theta^{(k)}, w^{(k)}) + \left\langle L_\theta'(\theta^{(k-1)}, w^{(k)}), \theta^* - \theta^{(k)} \right\rangle
$$

$$
\overset{(a)}{\leq} -\left\langle L_\theta'(\theta^{(k)}, w^{(k)}), \theta^* - \theta^{(k)} \right\rangle + \left\langle L_\theta'(\theta^{(k-1)}, w^{(k)}), \theta^* - \theta^{(k)} \right\rangle
$$

$$
= \left\langle L_\theta'(\theta^{(k)}, w^{(k)}) - L_\theta'(\theta^{(k-1)}, w^{(k)}), \theta^{(k)} - \theta^* \right\rangle
$$

$$
\overset{(b)}{=} \frac{1}{n_X} \sum_{i=0}^{n_X - 1} \left\langle \left( \overline{f}_i'(\theta^{(k)}) - \overline{f}_i'(\theta^{(k-1)}) \right) w_i^{(k)}, \theta^{(k)} - \theta^* \right\rangle
$$

$$
\leq \left| \frac{1}{n_X} \sum_{i=0}^{n_X - 1} \left\langle \left( \overline{f}_i'(\theta^{(k)}) - \overline{f}_i'(\theta^{(k-1)}) \right) w_i^{(k)}, \theta^{(k)} - \theta^* \right\rangle \right|
$$

$$
\overset{(c)}{\leq} \frac{1}{n_X} \sum_{i=0}^{n_X - 1} \left| \left\langle \left( \overline{f}_i'(\theta^{(k)}) - \overline{f}_i'(\theta^{(k-1)}) \right) w_i^{(k)}, \theta^{(k)} - \theta^* \right\rangle \right|
$$

$$
\overset{(d)}{\leq} \frac{1}{n_X} \sum_{i=0}^{n_X - 1} \left\| \left( \overline{f}_i'(\theta^{(k)}) - \overline{f}_i'(\theta^{(k-1)}) \right) w_i^{(k)} \right\| \cdot \|\theta^{(k)} - \theta^*\|
$$

$$
\leq \frac{1}{n_X} \sum_{i=0}^{n_X - 1} \left\| \overline{f}_i'(\theta^{(k)}) - \overline{f}_i'(\theta^{(k-1)}) \right\| \cdot \|w_i^{(k)}\| \cdot \|\theta^{(k)} - \theta^*\|
$$

$$
\overset{(e)}{=} \frac{1}{n_X} \sum_{i=0}^{n_X - 1} \left\| \frac{1}{n_{Y_i}} \sum_{j=0}^{n_{Y_i} - 1} \left( f_{\theta^{(k)}}'(x_i, y_{ij}) - f_{\theta^{(k-1)}}'(x_i, y_{ij}) \right) \right\| \cdot \|w_i^{(k)}\| \cdot \|\theta^{(k)} - \theta^*\|
$$

$$
\overset{(f)}{\leq} \frac{1}{n_X} \sum_{i=0}^{n_X - 1} \frac{1}{n_{Y_i}} \sum_{j=0}^{n_{Y_i} - 1} \left\| f_{\theta^{(k)}}'(x_i, y_{ij}) - f_{\theta^{(k-1)}}'(x_i, y_{ij}) \right\| \cdot \|w_i^{(k)}\| \cdot \|\theta^{(k)} - \theta^*\|
$$

$$
\overset{(g)}{\leq} \frac{1}{n_X} \sum_{i=0}^{n_X - 1} \frac{1}{n_{Y_i}} \sum_{j=0}^{n_{Y_i} - 1} B_\theta B_w \|\theta^{(k)} - \theta^{(k-1)}\| \cdot \|\theta^{(k)} - \theta^*\|
$$

$$
= B_\theta B_w \|\theta^{(k)} - \theta^{(k-1)}\| \cdot \|\theta^{(k)} - \theta^*\|
$$

$$\overset{(h)}{\le} \frac{B_\theta B_w}{\beta_0}\|\theta^{(k)} - \theta^{(k-1)}\|^2 + \beta_0 B_\theta B_w \|\theta^{(k)} - \theta^*\|^2 \tag{73}$$

where step (a) uses the fact that $L(\theta, w)$ is convex with respect to the $\theta$, step (b) substitutes the expression of $L'_\theta$, step (c) uses Jensen's inequality, step (d) uses Cauchy-Schwartz inequality, step (e) substitutes the expression for $\overline{f}'_i$, step (f) uses Jensen's inequality, step (g) uses the Lipschitz gradient property of $f'_\theta$ and Lemma E.7, step (h) uses $ab \le \frac{1}{\beta_0}a^2 + \beta_0 b^2$. In consequence, the above inequality implies that

$$- L(\theta^*, w^{(k)} - w^*) + L(\theta^{(k)}, w^*) - L(\theta^*, w^*) + \left\langle L'_\theta(\theta^{(k-1)}, w^{(k)}), \theta^* \right\rangle - \left\langle L'_\theta(\theta^{(k-1)}, w^{(k)}), \theta^{(k)} \right\rangle$$

$$\le - L(\theta^{(k)}, w^{(k)} - w^*) + \frac{B_\theta B_w}{\beta_0}\|\theta^{(k)} - \theta^{(k-1)}\|^2 + \beta_0 B_\theta B_w \|\theta^{(k)} - \theta^*\|^2 \tag{74}$$

Using (74), the $L$ terms in (72) becomes (notice that we keep the first and last $L$ terms in (72) intact)

$$L(\theta^{(k-1)}, w^{(k)} - w^*) - L(\theta^*, w^{(k)} - w^*) + L(\theta^{(k)}, w^*) - L(\theta^*, w^*) + \left\langle L'_\theta(\theta^{(k-1)}, w^{(k)}), \theta^* \right\rangle$$

$$- \left\langle L'_\theta(\theta^{(k-1)}, w^{(k)}), \theta^{(k)} \right\rangle + (n_X - 1)\left[ L(\theta^{(k-1)}, w^{(k)} - w^{(k-1)}) - L(\theta^*, w^{(k)} - w^{(k-1)}) \right]$$

$$\overset{(a)}{\le} L(\theta^{(k-1)}, w^{(k)} - w^*) - L(\theta^{(k)}, w^{(k)} - w^*) + (n_X - 1)\left[ L(\theta^{(k-1)}, w^{(k)} - w^{(k-1)}) - L(\theta^*, w^{(k)} - w^{(k-1)}) \right]$$

$$+ \frac{B_\theta B_w}{\beta_0}\|\theta^{(k)} - \theta^{(k-1)}\|^2 + \beta_0 B_\theta B_w \|\theta^{(k)} - \theta^*\|^2$$

$$= L\left( \theta^{(k-1)}, w^{(k)} - w^* + (n_X - 1)(w^{(k)} - w^{(k-1)}) \right) - L\left( \theta^{(k)}, w^{(k)} - w^* + (n_X - 1)(w^{(k)} - w^{(k-1)}) \right)$$

$$+ \frac{B_\theta B_w}{\beta_0}\|\theta^{(k)} - \theta^{(k-1)}\|^2 + \beta_0 B_\theta B_w \|\theta^{(k)} - \theta^*\|^2$$

$$\overset{(b)}{\le} \left\langle L'_\theta \left( \theta^{(k-1)}, w^{(k)} - w^* + (n_X - 1)(w^{(k)} - w^{(k-1)}) \right), \theta^{(k-1)} - \theta^{(k)} \right\rangle$$

$$+ \frac{B_\theta B_w}{\beta_0}\|\theta^{(k)} - \theta^{(k-1)}\|^2 + \beta_0 B_\theta B_w \|\theta^{(k)} - \theta^*\|^2$$

$$= \left\langle \frac{1}{n_X}\sum_{i=0}^{n_X - 1} \overline{f}'_i(\theta^{(k-1)})\left( w_i^{(k)} - w_i^* + (n_X - 1)(w_i^{(k)} - w_i^{(k-1)}) \right), \theta^{(k-1)} - \theta^{(k)} \right\rangle$$

$$+ \frac{B_\theta B_w}{\beta_0}\|\theta^{(k)} - \theta^{(k-1)}\|^2 + \beta_0 B_\theta B_w \|\theta^{(k)} - \theta^*\|^2$$

$$= \left\langle \frac{1}{n_X}\sum_{i=0}^{n_X - 1} \overline{f}'_i(\theta^{(k-1)})\left( w_i^{(k-1)} + n_X(w_i^{(k)} - w_i^{(k-1)}) - w_i^* \right), \theta^{(k-1)} - \theta^{(k)} \right\rangle$$

$$+ \frac{B_\theta B_w}{\beta_0}\|\theta^{(k)} - \theta^{(k-1)}\|^2 + \beta_0 B_\theta B_w \|\theta^{(k)} - \theta^*\|^2$$

$$\overset{(c)}{\le} \beta_1 \left\| \frac{1}{n_X}\sum_{i=0}^{n_X - 1} \overline{f}'_i(\theta^{(k-1)})\left( w_i^{(k-1)} + n_X(w_i^{(k)} - w_i^{(k-1)}) - w_i^* \right) \right\|^2 + \frac{1}{\beta_1}\|\theta^{(k-1)} - \theta^{(k)}\|^2$$

$$+ \frac{B_\theta B_w}{\beta_0}\|\theta^{(k)} - \theta^{(k-1)}\|^2 + \beta_0 B_\theta B_w \|\theta^{(k)} - \theta^*\|^2$$

$$= \beta_1 \left\| \frac{1}{n_X}\sum_{i=0}^{n_X - 1} \overline{f}'_i(\theta^{(k-1)})\left( w_i^{(k-1)} - w_i^* \right) + \sum_{i=0}^{n_X - 1} \overline{f}'_i(\theta^{(k-1)})(w_i^{(k)} - w_i^{(k-1)}) \right\|^2$$

$$+ \frac{1}{\beta_1}\|\theta^{(k-1)} - \theta^{(k)}\|^2 + \frac{B_\theta B_w}{\beta_0}\|\theta^{(k)} - \theta^{(k-1)}\|^2 + \beta_0 B_\theta B_w \|\theta^{(k)} - \theta^*\|^2$$

$$\overset{(d)}{\le} 2\beta_1 \left\| \frac{1}{n_X}\sum_{i=0}^{n_X - 1} \overline{f}'_i(\theta^{(k-1)})\left( w_i^{(k-1)} - w_i^* \right) \right\|^2 + 2\beta_1 \left\| \sum_{i=0}^{n_X - 1} \overline{f}'_i(\theta^{(k-1)})(w_i^{(k)} - w_i^{(k-1)}) \right\|^2$$

$$+ \frac{1}{\beta_1}\|\theta^{(k-1)} - \theta^{(k)}\|^2 + \frac{B_\theta B_w}{\beta_0}\|\theta^{(k)} - \theta^{(k-1)}\|^2 + \beta_0 B_\theta B_w \|\theta^{(k)} - \theta^*\|^2$$

$$\overset{(e)}{\leq} \frac{2\beta_1 B_f^2}{n_X} \sum_{i=0}^{n_X-1} \|w_i^{(k-1)} - w_i^*\|^2 + 2\beta_1 \left\| \sum_{i=0}^{n_X-1} \overline{f}'_i(\theta^{(k-1)})(w_i^{(k)} - w_i^{(k-1)}) \right\|^2$$

$$+ \left( \frac{1}{\beta_1} + \frac{B_\theta B_w}{\beta_0} \right) \|\theta^{(k-1)} - \theta^{(k)}\|^2 + \beta_0 B_\theta B_w \|\theta^{(k)} - \theta^*\|^2 \tag{75}$$

where step (a) applies (74), step (b) uses convexity of $L$ in $\theta$, step (c) uses $\langle a, b \rangle \leq \beta_1 \|a\|^2 + \frac{1}{\beta_1}\|b\|^2$, for some $\beta_1 > 0$ to be chosen later, step (d) uses $\|a + b\|^2 \leq 2\|a\|^2 + 2\|b\|^2$, and step (e) applies Jensen's inequality and bounded gradient assumption to the first term. Before we proceed, we note that, by taking expectation of the second term conditioned on $\mathcal{F}_k$, we get

$$\mathsf{E}\left\{ \left\| \sum_{i=0}^{n_X-1} \overline{f}'_i(\theta^{(k-1)})(w_i^{(k)} - w_i^{(k-1)}) \right\|^2 \Big| \mathcal{F}_k \right\}$$

$$= \sum_{i=0}^{n_X-1} \sum_{j=0}^{n_{Y_i}-1} \frac{1}{n_X n_{Y_i}} \left\| \overline{f}'_i(\theta^{(k-1)})(w'_{ij} - w_i^{(k-1)}) \right\|^2$$

$$\leq B_f^2 \sum_{i=0}^{n_X-1} \sum_{j=0}^{n_{Y_i}-1} \frac{1}{n_X n_{Y_i}} \left\| w'_{ij} - w_i^{(k-1)} \right\|^2$$

$$= B_f^2 \sum_{i=0}^{n_X-1} \mathsf{E}\left\{ \|w_i^{(k)} - w_i^{(k-1)}\|^2 \big| \mathcal{F}_k \right\} \tag{76}$$

Therefore, the conditional expectation of all the $L$ terms are bounded by

$$\frac{2\beta_1 B_f^2}{n_X} \sum_{i=0}^{n_X-1} \|w_i^{(k-1)} - w_i^*\|^2 + 2\beta_1 B_f^2 \sum_{i=0}^{n_X-1} \mathsf{E}\left\{ \|w_i^{(k)} - w_i^{(k-1)}\|^2 \big| \mathcal{F}_k \right\}$$

$$+ \left( \frac{1}{\beta_1} + \frac{B_\theta B_w}{\beta} \right) \mathsf{E}\left\{ \|\theta^{(k-1)} - \theta^{(k)}\|^2 \big| \mathcal{F}_k \right\} + \beta_0 B_\theta B_w \mathsf{E}\left\{ \|\theta^{(k)} - \theta^*\|^2 \big| \mathcal{F}_k \right\}$$

$$= \frac{2\beta_1 B_f^2}{n_X} \|w^{(k-1)} - w^*\|^2 + 2\beta_1 B_f^2 \mathsf{E}\left\{ \|w^{(k)} - w^{(k-1)}\|^2 \big| \mathcal{F}_k \right\}$$

$$+ \left( \frac{1}{\beta_1} + \frac{B_\theta B_w}{\beta_0} \right) \mathsf{E}\left\{ \|\theta^{(k-1)} - \theta^{(k)}\|^2 \big| \mathcal{F}_k \right\} + \beta_0 B_\theta B_w \mathsf{E}\left\{ \|\theta^{(k)} - \theta^*\|^2 \big| \mathcal{F}_k \right\} \tag{77}$$

Therefore, the total bound (72) becomes

$$\left( \frac{1}{2\alpha_w} + \gamma \right) \mathsf{E}\{\|w^{(k)} - w^*\|^2 \mid \mathcal{F}_k\} + \frac{1}{2\alpha_w} \mathsf{E}\{\|w^{(k)} - w^{(k-1)}\|^2 \mid \mathcal{F}_k\}$$

$$+ \left( \frac{1}{2\alpha_\theta} + \mu \right) \mathsf{E}\left\{ \|\theta^{(k)} - \theta^*\|^2 \big| \mathcal{F}_k \right\} + \frac{1}{2\alpha_\theta} \mathsf{E}\left\{ \|\theta^{(k)} - \theta^{(k-1)}\|^2 \big| \mathcal{F}_k \right\}$$

$$\leq \left( \frac{1}{2\alpha_w} + \frac{\gamma(n_X - 1)}{n_X} \right) \|w^{(k-1)} - w^*\|^2 + \frac{1}{2\alpha_\theta} \|\theta^{(k-1)} - \theta^*\|^2$$

$$+ \left( 2\alpha_w B_f^2 \left( 1 - \overline{1/n_Y} \right) + \alpha_\theta B_w^2 B_\theta^2 \right) \|\theta^{(k-1)} - \widetilde{\theta}\|^2$$

$$+ \frac{2\beta_1 B_f^2}{n_X} \|w^{(k-1)} - w^*\|^2 + 2\beta_1 B_f^2 \mathsf{E}\{\|w^{(k)} - w^{(k-1)}\|^2 | \mathcal{F}_k\}$$

$$+ \left( \frac{1}{\beta_1} + \frac{B_\theta B_w}{\beta_0} \right) \mathsf{E}\left\{ \|\theta^{(k-1)} - \theta^{(k)}\|^2 \big| \mathcal{F}_k \right\} + \beta_0 B_\theta B_w \mathsf{E}\{\|\theta^{(k)} - \theta^*\|^2 | \mathcal{F}_k\} \tag{78}$$

By combining the common terms, we obtain

$$\left( \frac{1}{2\alpha_\theta} + \mu - \beta_0 B_\theta B_w \right) \mathsf{E}\left\{ \|\theta^{(k)} - \theta^*\|^2 \big| \mathcal{F}_k \right\} + \left( \frac{1}{2\alpha_w} + \gamma \right) \mathsf{E}\{\|w^{(k)} - w^*\|^2 \mid \mathcal{F}_k\}$$

$$+ \left(\frac{1}{2\alpha_\theta} - \frac{1}{\beta_1} - \frac{B_\theta B_w}{\beta_0}\right)\mathsf{E}\left\{\|\theta^{(k)} - \theta^{(k-1)}\|^2\big|\mathcal{F}_k\right\} + \left(\frac{1}{2\alpha_w} - 2\beta_1 B_f^2\right)\mathsf{E}\left\{\|w^{(k)} - w^{(k-1)}\|^2 \mid \mathcal{F}_k\right\}$$

$$\leq \frac{1}{2\alpha_\theta}\|\theta^{(k-1)} - \theta^*\|^2 + \left(\frac{1}{2\alpha_w} + \gamma - \frac{\gamma}{n_X} + \frac{2\beta_1 B_f^2}{n_X}\right)\|w^{(k-1)} - w^*\|^2$$

$$+ \left(2\alpha_w B_f^2\left(1 - \overline{1/n_Y}\right) + \alpha_\theta B_w^2 B_\theta^2\right)\|\theta^{(k-1)} - \widetilde{\theta}\|^2 \tag{79}$$

Applying inequality $\|x + y\|^2 \leq 2\|x\|^2 + 2\|y\|^2$ to the last term in (79), we obtain

$$\left(\frac{1}{2\alpha_\theta} + \mu - \beta_0 B_\theta B_w\right)\mathsf{E}\left\{\|\theta^{(k)} - \theta^*\|^2\big|\mathcal{F}_k\right\} + \left(\frac{1}{2\alpha_w} + \gamma\right)\mathsf{E}\left\{\|w^{(k)} - w^*\|^2 \mid \mathcal{F}_k\right\}$$

$$+ \left(\frac{1}{2\alpha_\theta} - \frac{1}{\beta_1} - \frac{B_\theta B_w}{\beta_0}\right)\mathsf{E}\left\{\|\theta^{(k)} - \theta^{(k-1)}\|^2\big|\mathcal{F}_k\right\} + \left(\frac{1}{2\alpha_w} - 2\beta_1 B_f^2\right)\mathsf{E}\left\{\|w^{(k)} - w^{(k-1)}\|^2 \mid \mathcal{F}_k\right\}$$

$$\leq \left(\frac{1}{2\alpha_\theta} + 4\alpha_w B_f^2\left(1 - \overline{1/n_Y}\right) + 2\alpha_\theta B_w^2 B_\theta^2\right)\|\theta^{(k-1)} - \theta^*\|^2$$

$$+ \left(\frac{1}{2\alpha_w} + \gamma - \frac{\gamma}{n_X} + \frac{2\beta_1 B_f^2}{n_X}\right)\|w^{(k-1)} - w^*\|^2 + \left(4\alpha_w B_f^2\left(1 - \overline{1/n_Y}\right) + 2\alpha_\theta B_w^2 B_\theta^2\right)\|\widetilde{\theta} - \theta^*\|^2 \tag{80}$$

Taking full expectation of the above inequality, we obtain:

$$\left(\frac{1}{2\alpha_\theta} + \mu - \beta_0 B_\theta B_w\right)\mathsf{E}\|\theta^{(k)} - \theta^*\|^2 + \left(\frac{1}{2\alpha_w} + \gamma\right)\mathsf{E}\|w^{(k)} - w^*\|^2$$

$$+ \left(\frac{1}{2\alpha_\theta} - \frac{1}{\beta_1} - \frac{B_\theta B_w}{\beta_0}\right)\mathsf{E}\|\theta^{(k)} - \theta^{(k-1)}\|^2 + \left(\frac{1}{2\alpha_w} - 2\beta_1 B_f^2\right)\mathsf{E}\|w^{(k)} - w^{(k-1)}\|^2$$

$$\leq \left(\frac{1}{2\alpha_\theta} + 4\alpha_w B_f^2\left(1 - \overline{1/n_Y}\right) + 2\alpha_\theta B_w^2 B_\theta^2\right)\mathsf{E}\|\theta^{(k-1)} - \theta^*\|^2$$

$$+ \left(\frac{1}{2\alpha_w} + \gamma - \frac{\gamma}{n_X} + \frac{2\beta_1 B_f^2}{n_X}\right)\mathsf{E}\|w^{(k-1)} - w^*\|^2$$

$$+ \left(4\alpha_w B_f^2\left(1 - \overline{1/n_Y}\right) + 2\alpha_\theta B_w^2 B_\theta^2\right)\mathsf{E}\|\widetilde{\theta} - \theta^*\|^2 \tag{81}$$

In order for the above inequality to converge, the hyperparameters need to be chosen to satisfy the following conditions:

$$\frac{1}{2\alpha_\theta} \geq \frac{1}{\beta_1} + \frac{B_\theta B_w}{\beta_0}$$

$$\alpha_w \leq \frac{1}{4B_f^2\beta_1}$$

$$\beta_1 < \frac{\gamma}{2B_f^2}$$

$$4\alpha_w B_f^2(1 - \overline{1/n_Y}) + 2\alpha_\theta B_w^2 B_\theta^2 < \mu - \beta_0 B_\theta B_w \tag{82}$$

which simplifies the recursion to be

$$\left(\frac{1}{2\alpha_\theta} + \mu - \beta_0 B_\theta B_w\right)\mathsf{E}\|\theta^{(k)} - \theta^*\|^2 + \left(\frac{1}{2\alpha_w} + \gamma\right)\mathsf{E}\|w^{(k)} - w^*\|^2$$

$$\leq \left(\frac{1}{2\alpha_\theta} + 4\alpha_w B_f^2\left(1 - \overline{1/n_Y}\right) + 2\alpha_\theta B_w^2 B_\theta^2\right)\mathsf{E}\|\theta^{(k-1)} - \theta^*\|^2$$

$$+ \left(\frac{1}{2\alpha_w} + \gamma - \frac{\gamma}{n_X} + \frac{2\beta_1 B_f^2}{n_X}\right)\mathsf{E}\|w^{(k-1)} - w^*\|^2$$

$$+ \left(4\alpha_w B_f^2\left(1 - \overline{1/n_Y}\right) + 2\alpha_\theta B_w^2 B_\theta^2\right)\mathsf{E}\|\widetilde{\theta} - \theta^*\|^2 \tag{83}$$

Inequality (83) can also be further written as

$$
\mathsf{E}\|\theta^{(k)} - \theta^*\|^2 + \frac{\frac{1}{2\alpha_w} + \gamma}{\frac{1}{2\alpha_\theta} + \mu - \beta_0 B_\theta B_w}\mathsf{E}\|w^{(k)} - w^*\|^2
$$

$$
\leq r_P \mathsf{E}\|\theta^{(k-1)} - \theta^*\|^2 + r_D \cdot \frac{\frac{1}{2\alpha_w} + \gamma}{\frac{1}{2\alpha_\theta} + \mu - \beta_0 B_\theta B_w}\mathsf{E}\|w^{(k-1)} - w^*\|^2
$$

$$
+ \frac{4\alpha_w B_f^2\big(1 - \overline{1/n_Y}\big) + 2\alpha_\theta B_w^2 B_\theta^2}{\frac{1}{2\alpha_\theta} + \mu - \beta_0 B_\theta B_w}\mathsf{E}\|\widetilde{\theta} - \theta^*\|^2 \tag{84}
$$

where $r_P$ and $r_D$ are the primal and the dual ratios, defined as

$$
r_P = \frac{\frac{1}{2\alpha_\theta} + 4\alpha_w B_f^2\big(1 - \overline{1/n_Y}\big) + 2\alpha_\theta B_w^2 B_\theta^2}{\frac{1}{2\alpha_\theta} + \mu - \beta_0 B_\theta B_w}
$$

$$
r_D = 1 - \frac{1}{n_X}\frac{2\alpha_w(\gamma - 2\beta_1 B_f^2)}{1 + 2\alpha_w \gamma}
$$

We choose $\beta_0$, $\beta_1$, and the primal and the dual step-sizes to be

$$
\beta_0 = \frac{\mu}{2 B_\theta B_w}, \quad \beta_1 = \frac{\gamma}{4 B_f^2}
$$

$$
\alpha_\theta = \frac{\frac{1}{\mu}}{64 n_X\big(\frac{B_f^2}{\mu\gamma} + \frac{B_\theta^2 B_w^2}{\mu^2}\big) + n_X} = \frac{1}{\mu}\cdot\frac{1}{64 n_X \kappa + n_X}
$$

$$
\alpha_w = \frac{\frac{1}{\gamma}}{64\big(\frac{B_f^2}{\mu\gamma} + \frac{B_\theta^2 B_w^2}{\mu^2}\big) + 1} = \frac{1}{\gamma}\cdot\frac{1}{64\kappa + 1}
$$

where $\kappa$ is the condition number defined as

$$
\kappa = \frac{B_f^2}{\mu\gamma} + \frac{B_\theta^2 B_w^2}{\mu^2} \tag{85}
$$

It can be verified that the above choice of step-sizes satisfies the condition (82). With our choice of the parameters, we also have

$$
\frac{\frac{1}{2\alpha_w} + \gamma}{\frac{1}{2\alpha_\theta} + \mu - \beta_0 B_\theta B_w} = \frac{\frac{1}{2\alpha_w} + \gamma}{\frac{1}{2\alpha_\theta} + \frac{\mu}{2}} = \frac{\gamma}{\mu}\cdot\frac{64\kappa + 3}{64 n_X \kappa + n_X + 1} \tag{86}
$$

and

$$
\frac{4\alpha_w B_f^2\big(1 - \overline{1/n_Y}\big) + 2\alpha_\theta B_w^2 B_\theta^2}{\frac{1}{2\alpha_\theta} + \mu - \beta_0 B_\theta B_w} = \frac{4\alpha_w B_f^2\big(1 - \overline{1/n_Y}\big) + 2\alpha_\theta B_w^2 B_\theta^2}{\frac{1}{2\alpha_\theta} + \frac{\mu}{2}}
$$

$$
= \frac{\frac{8 B_f^2\big(1 - \overline{1/n_Y}\big)}{\mu\gamma} + \frac{4 B_w^2 B_\theta^2}{\mu^2}\cdot\frac{1}{n_X}}{(64\kappa + 1)(64 n_X \kappa + n_X + 1)} \tag{87}
$$

Substituting (86)–(87) into (84), we obtain

$$
\mathsf{E}\|\theta^{(k)} - \theta^*\|^2 + \frac{\gamma}{\mu}\cdot\frac{64\kappa + 3}{64 n_X \kappa + n_X + 1}\mathsf{E}\|w^{(k)} - w^*\|^2
$$

$$
\leq r_P \mathsf{E}\|\theta^{(k-1)} - \theta^*\|^2 + r_D \cdot \frac{\gamma}{\mu}\cdot\frac{64\kappa + 3}{64 n_X \kappa + n_X + 1}\mathsf{E}\|w^{(k-1)} - w^*\|^2
$$

$$
+ \frac{\frac{8 B_f^2\big(1 - \overline{1/n_Y}\big)}{\mu\gamma} + \frac{4 B_w^2 B_\theta^2}{\mu^2}\cdot\frac{1}{n_X}}{(64\kappa + 1)(64 n_X \kappa + n_X + 1)}\mathsf{E}\|\widetilde{\theta} - \theta^*\|^2 \tag{88}
$$

Furthermore, the primal and the dual ratios can be upper bounded as

$$
\begin{aligned}
r_P &= \frac{1 + 8\alpha_\theta \alpha_w B_f^2 \left(1 - \overline{1/n_Y}\right) + 4\alpha_\theta^2 B_w^2 B_\theta^2}{1 + 2\alpha_\theta \mu - 2\alpha_\theta \beta_0 B_\theta B_w} \\
&= \frac{1 + 8\alpha_\theta \alpha_w B_f^2 \left(1 - \overline{1/n_Y}\right) + 4\alpha_\theta^2 B_w^2 B_\theta^2}{1 + \alpha_\theta \mu} \\
&= 1 - \frac{\alpha_\theta \mu - 8\alpha_\theta \alpha_w B_f^2 \left(1 - \overline{1/n_Y}\right) - 4\alpha_\theta^2 B_w^2 B_\theta^2}{1 + \alpha_\theta \mu} \\
&\leq 1 - \frac{1}{\frac{1024}{13} n_X \kappa + \frac{16}{13} n_X + \frac{16}{13}} \\
&\leq 1 - \frac{1}{78.8 n_X \kappa + 1.3 n_X + 1.3} \qquad (89) \\
r_D &= 1 - \frac{1}{n_X} \frac{2\alpha_w(\gamma - 2\beta_1 B_f^2)}{1 + 2\alpha_w \gamma} \\
&= 1 - \frac{1}{n_X} \frac{\alpha_w \gamma}{1 + 2\alpha_w \gamma} \\
&= 1 - \frac{1}{64 n_X \kappa + 3 n_X} < r_P \qquad (90)
\end{aligned}
$$

Therefore, inequality (88) can be further upper bounded as

$$
\begin{aligned}
&\mathsf{E}\|\theta^{(k)} - \theta^*\|^2 + \frac{\gamma}{\mu} \cdot \frac{64\kappa + 3}{64 n_X \kappa + n_X + 1} \mathsf{E}\|w^{(k)} - w^*\|^2 \\
&\leq r_P \left( \mathsf{E}\|\theta^{(k-1)} - \theta^*\|^2 + \frac{\gamma}{\mu} \cdot \frac{64\kappa + 3}{64 n_X \kappa + n_X + 1} \mathsf{E}\|w^{(k-1)} - w^*\|^2 \right) \\
&\quad + \frac{\frac{8 B_f^2 \left(1 - \overline{1/n_Y}\right)}{\mu\gamma} + \frac{4 B_w^2 B_\theta^2}{\mu^2} \cdot \frac{1}{n_X}}{(64\kappa + 1)(64 n_X \kappa + n_X + 1)} \mathsf{E}\|\widetilde{\theta} - \theta^*\|^2 \\
&\leq r_P \left( \mathsf{E}\|\theta^{(k-1)} - \theta^*\|^2 + \frac{\gamma}{\mu} \cdot \frac{64\kappa + 3}{64 n_X \kappa + n_X + 1} \mathsf{E}\|w^{(k-1)} - w^*\|^2 \right) \\
&\quad + \frac{\frac{8 B_f^2 \left(1 - \overline{1/n_Y}\right)}{\mu\gamma} + \frac{4 B_w^2 B_\theta^2}{\mu^2} \cdot \frac{1}{n_X}}{(64\kappa + 1)(64 n_X \kappa + n_X + 1)} \left( \mathsf{E}\|\widetilde{\theta} - \theta^*\|^2 + \frac{\gamma}{\mu} \cdot \frac{64\kappa + 3}{64 n_X \kappa + n_X + 1} \mathsf{E}\|\tilde{w} - w^*\|^2 \right) \\
&\overset{(a)}{=} r_P \left( \mathsf{E}\|\theta^{(k-1)} - \theta^*\|^2 + \frac{\gamma}{\mu} \cdot \frac{64\kappa + 3}{64 n_X \kappa + n_X + 1} \mathsf{E}\|w^{(k-1)} - w^*\|^2 \right) \\
&\quad + \frac{\frac{8 B_f^2 \left(1 - \overline{1/n_Y}\right)}{\mu\gamma} + \frac{4 B_w^2 B_\theta^2}{\mu^2} \cdot \frac{1}{n_X}}{(64\kappa + 1)(64 n_X \kappa + n_X + 1)} \left( \mathsf{E}\|\widetilde{\theta}_{s-1} - \theta^*\|^2 + \frac{\gamma}{\mu} \cdot \frac{64\kappa + 3}{64 n_X \kappa + n_X + 1} \mathsf{E}\|\tilde{w}_{s-1} - w^*\|^2 \right)
\end{aligned}
$$
$$(91)$$

where step (a) uses the fact that the $\widetilde{\theta} = \widetilde{\theta}_{s-1}$ and $\tilde{w} = \tilde{w}_{s-1}$ when we are considering the $s$-th stage/outer-loop (see Algorithm 1 in the main paper). Define the following Lyapunov functions:

$$
P_{s,k} = \mathsf{E}\|\theta^{(k)} - \theta^*\|^2 + \frac{\gamma}{\mu} \cdot \frac{64\kappa + 3}{64 n_X \kappa + n_X + 1} \mathsf{E}\|w^{(k)} - w^*\|^2
$$
$$
P_s = \mathsf{E}\|\widetilde{\theta}_s - \theta^*\|^2 + \frac{\gamma}{\mu} \cdot \frac{64\kappa + 3}{64 n_X \kappa + n_X + 1} \mathsf{E}\|\tilde{w}_s - w^*\|^2
$$

As a result, we can rewrite inequality (91) as

$$
P_{s,k} \leq r_P \cdot P_{s,k-1} + \frac{\frac{8 B_f^2 \left(1 - \overline{1/n_Y}\right)}{\mu\gamma} + \frac{4 B_w^2 B_\theta^2}{\mu^2} \cdot \frac{1}{n_X}}{(64\kappa + 1)(64 n_X \kappa + n_X + 1)} P_{s-1}
$$

$$
\leq \left(1 - \frac{1}{78.8 n_X \kappa + 1.3 n_X + 1.3}\right) \cdot P_{s,k-1} + \frac{\frac{8B_f^2\left(1-\overline{1/n_Y}\right)}{\mu\gamma} + \frac{4B_w^2 B_\theta^2}{\mu^2}\cdot\frac{1}{n_X}}{(64\kappa+1)(64 n_X\kappa + n_X + 1)} P_{s-1} \quad (92)
$$

Furthermore, at the $s$-th stage (outer loop iteration), when Option I is used in Algorithm 1 in the main paper, we have $\widetilde{\theta}_s = \theta^{(M)}$ and $\widetilde{w}_s = w^{(M)}$. Therefore, it holds that

$$
P_s = P_{s,M}
$$

$$
\overset{(a)}{\leq} \left(1 - \frac{1}{78.8 n_X \kappa + 1.3 n_X + 1.3}\right)^M P_{s,0}
$$

$$
+ \sum_{k=0}^{M-1} \left(1 - \frac{1}{78.8 n_X \kappa + 1.3 n_X + 1.3}\right)^k \times \frac{\frac{8B_f^2\left(1-\overline{1/n_Y}\right)}{\mu\gamma} + \frac{4B_w^2 B_\theta^2}{\mu^2}\cdot\frac{1}{n_X}}{(64\kappa+1)(64 n_X\kappa + n_X + 1)} P_{s-1}
$$

$$
\leq \left(1 - \frac{1}{78.8 n_X \kappa + 1.3 n_X + 1.3}\right)^M P_{s,0}
$$

$$
+ \sum_{k=0}^{+\infty} \left(1 - \frac{1}{78.8 n_X \kappa + 1.3 n_X + 1.3}\right)^k \times \frac{\frac{8B_f^2\left(1-\overline{1/n_Y}\right)}{\mu\gamma} + \frac{4B_w^2 B_\theta^2}{\mu^2}\cdot\frac{1}{n_X}}{(64\kappa+1)(64 n_X\kappa + n_X + 1)} P_{s-1}
$$

$$
= \left(1 - \frac{1}{78.8 n_X \kappa + 1.3 n_X + 1.3}\right)^M P_{s,0}
$$

$$
+ \left(\frac{8B_f^2\left(1-\overline{1/n_Y}\right)}{\mu\gamma} + \frac{4B_w^2 B_\theta^2}{\mu^2}\cdot\frac{1}{n_X}\right)\frac{78.8 n_X\kappa + 1.3 n_X + 1.3}{(64\kappa+1)(64 n_X\kappa + n_X + 1)} P_{s-1}
$$

$$
\leq \left(1 - \frac{1}{78.8 n_X \kappa + 1.3 n_X + 1.3}\right)^M P_{s,0} + \left(\frac{8B_f^2\left(1-\overline{1/n_Y}\right)}{\mu\gamma} + \frac{4B_w^2 B_\theta^2}{\mu^2}\cdot\frac{1}{n_X}\right)\frac{1.3}{64\kappa+1} P_{s-1}
$$

$$
\leq \left(1 - \frac{1}{78.8 n_X \kappa + 1.3 n_X + 1.3}\right)^M P_{s,0} + \frac{\frac{16B_f^2\left(1-\overline{1/n_Y}\right)}{\mu\gamma} + \frac{8B_w^2 B_\theta^2}{\mu^2}\cdot\frac{1}{n_X}}{64\kappa+1} P_{s-1}
$$

$$
\overset{(b)}{=} \left[\left(1 - \frac{1}{78.8 n_X \kappa + 1.3 n_X + 1.3}\right)^M + \frac{\frac{16B_f^2\left(1-\overline{1/n_Y}\right)}{\mu\gamma} + \frac{8B_w^2 B_\theta^2}{\mu^2}\cdot\frac{1}{n_X}}{64\kappa+1}\right] P_{s-1}
$$

$$
\leq \left[e^{-\frac{M}{78.8 n_X \kappa + 1.3 n_X + 1.3}} + \frac{1}{4}\right] P_{s-1} \quad (93)
$$

where step (a) iteratively applies inequality (91), and step (b) uses the fact $P_{s,0} = P_{s-1}$ (because $\theta^{(0)} = \widetilde{\theta}_{s-1}$ and $w^{(0)} = \widetilde{w}_{s-1}$ as shown in Algorithm 1). Choosing $M = \lceil 78.8 n_X\kappa + 1.3 n_X + 1.3 \rceil$, where $\lceil \cdot \rceil$ denotes the roundup operation, we have

$$
P_s \leq (e^{-1} + 1/4) P_{s-1} < \frac{3}{4} P_{s-1} \leq \left(\frac{3}{4}\right)^s P_0 \quad (94)
$$

Therefore, $P_s$ converges to zero at a linear rate of $3/4$. Furthermore, we requires a total of $\ln\frac{1}{\epsilon}$ outer loop iterations to reach $\epsilon$-solution. And for each outer loop iteration, it requires $M$ steps of inner-loop primal-dual updates, which is $O(1)$ per step (in number oracle calls), and $O(n_X n_Y)$ for evaluating the batch gradients for the control variates, where $n_Y = (n_{Y_0} + \cdots + n_{Y_{n_X-1}})/n_X$. Therefore, the complexity per outer loop iteration is $O(n_X n_Y + M)$, and the total complexity is

$$
O\left((n_X n_Y + n_X\kappa + n_X)\ln\frac{1}{\epsilon}\right) \quad (95)
$$

Noting that $\mathsf{E}\|\widetilde{\theta}_s - \theta^*\|^2 \leq P_s$, the above bound (94) also implies that $\mathsf{E}\|\widetilde{\theta}_s - \theta^*\|^2$ also converges to zero at a linear rate of $3/4$ and the total complexity to reach $\mathsf{E}\|\widetilde{\theta}_s - \theta^*\|^2 \leq \epsilon$ is also given by (95).

### E.8 Convergence for Option II

Next, we move on to analyze the Option II case of the algorithm, wherein $\widetilde{\theta}$ at the end of each state is chosen to be one of the $M$ previous $\theta^{(k)}$ values (see Algorithm 1 in the main paper).

Adding (56) and (71) we obtain the total bound for the primal and dual variable updates:

$$
\left(\frac{1}{2\alpha_w} + \gamma\right) \mathsf{E}\{\|w^{(k)} - w^*\|^2 \mid \mathcal{F}_k\} + \frac{1}{2\alpha_w}\mathsf{E}\{\|w^{(k)} - w^{(k-1)}\|^2 \mid \mathcal{F}_k\}
$$

$$
\left(\frac{1}{2\alpha_\theta} + \mu\right) \mathsf{E}\left\{\|\theta^{(k)} - \theta^*\|^2 \big| \mathcal{F}_k\right\} + \frac{1}{2\alpha_\theta}\mathsf{E}\left\{\|\theta^{(k)} - \theta^{(k-1)}\|^2 \big| \mathcal{F}_k\right\}
$$

$$
\leq \left(\frac{1}{2\alpha_w} + \frac{\gamma(n_X - 1)}{n_X}\right)\|w^{(k-1)} - w^*\|^2 + \frac{1}{2\alpha_\theta}\|\theta^{(k-1)} - \theta^*\|^2
$$

$$
+ \left(2\alpha_w B_f^2\left(1 - \overline{1/n_Y}\right) + \alpha_\theta B_w^2 B_\theta^2\right)\|\theta^{(k-1)} - \widetilde{\theta}\|^2 \tag{96}
$$

$$
+ \mathsf{E}\left\{L(\theta^{(k-1)}, w^{(k)} - w^*) - L(\theta^*, w^{(k)} - w^*) \mid \mathcal{F}_k\right\}
$$

$$
+ (n_X - 1)\mathsf{E}\left\{L(\theta^{(k-1)}, w^{(k)} - w^{(k-1)}) - L(\theta^*, w^{(k)} - w^{(k-1)}) \mid \mathcal{F}_k\right\}
$$

$$
+ \mathsf{E}\left\{\left[L(\theta^{(k)}, w^*) - L(\theta^*, w^*)\right]\big|\mathcal{F}_k\right\}
$$

$$
+ \mathsf{E}\left\{\left[\left\langle L_\theta'(\theta^{(k-1)}, w^{(k)}), \theta^*\right\rangle - \left\langle L_\theta'(\theta^{(k-1)}, w^{(k)}), \theta^{(k)}\right\rangle\right]\big|\mathcal{F}_k\right\}
$$

We will now bound the $L$ terms in (96). First consider the second $L$ term in (96):

$$
(n_X - 1)\left[L(\theta^{(k-1)}, w^{(k)} - w^{(k-1)}) - L(\theta^*, w^{(k)} - w^{(k-1)})\right]
$$

$$
\overset{(a)}{\leq}(n_X - 1)\left[\left\langle L'(\theta^{(k-1)}, w^{(k)} - w^{(k-1)}), \theta^{(k-1)} - \theta^*\right\rangle\right]
$$

$$
\overset{(b)}{\leq}\beta_2(n_X - 1)^2\|L'(\theta^{(k-1)}, w^{(k)} - w^{(k-1)})\|^2 + \frac{1}{\beta_2}\|\theta^{(k-1)} - \theta^*\|^2 \tag{97}
$$

for some $\beta_2 > 0$ to be determined later, where step $(a)$ uses convexity of the function $L$ in its first variable, and $(b)$ uses $a^2 + b^2 \geq 2ab \geq ab$.

We further lower bound the first term in (97). Ignoring the scaling factors, this can be rewritten as follows:

$$
\|L'(\theta^{(k-1)}, w^{(k)} - w^{(k-1)})\|^2 \overset{(a)}{=} \left\|\frac{1}{n_X}\sum_{i=0}^{n_X-1}\overline{f}_i'(\theta)(w_i^{(k)} - w_i^{(k-1)})\right\|^2
$$

$$
\overset{(b)}{=} \frac{1}{n_X^2}\left\|\sum_{i=0}^{n_X-1}\overline{f}_i'(\theta)(w_i^{(k)} - w_i^{(k-1)})\right\|^2 \tag{98}
$$

$$
\overset{(c)}{\leq} \frac{B_f^2}{n_X^2}\left\|\sum_{i=0}^{n_X-1}w_i^{(k)} - w_i^{(k-1)}\right\|^2
$$

where $(a)$ follows from the definition (25), $(b)$ is direct, by removing the $1/n_X$ outside the $\|\cdot\|^2$ operator, and $(c)$ uses the bounded gradients assumption (Assumption E.3).

Notice that, conditioned on $\mathcal{F}_k$, $w^{(k)}$ is the only random variable in (98). Furthermore, for each $i$ and $j$, using (38) we have,

$$
\sum_{i=0}^{n_X-1}w_i^{(k)} - w_i^{(k-1)} = w_{ij}' - w_i^{(k-1)} \qquad \text{with probability} \quad 1/n_X n_{Y_i} \tag{99}
$$

Taking expectation, conditioned on $\mathcal{F}_k$ on both sides of (98),

$$\mathsf{E}\left\{\|L'(\theta^{(k-1)}, w^{(k)} - w^{(k-1)})\|^2 \Big| \mathcal{F}_k\right\} \leq \frac{B_f^2}{n_X^2} \mathsf{E}\left\{\left\|\sum_{i=0}^{n_X-1} w_i^{(k)} - w_i^{(k-1)}\right\|^2 \Big| \mathcal{F}_k\right\}$$

$$\stackrel{(a)}{=} \frac{B_f^2}{n_X^2} \frac{1}{n_X n_{Y_i}} \sum_{i=0}^{n_X-1} \sum_{j=0}^{n_{Y_i}-1} \left\|w_{ij}' - w_i^{(k-1)}\right\|^2$$

$$\stackrel{(b)}{=} \frac{B_f^2}{n_X^2} \sum_{i=0}^{n_X-1} \mathsf{E}\left\{\left\|w_i^{(k)} - w_i^{(k-1)}\right\|^2 \Big| \mathcal{F}_k\right\}$$

$$\stackrel{(c)}{=} \frac{B_f^2}{n_X^2} \mathsf{E}\left\{\left\|w^{(k)} - w^{(k-1)}\right\|^2 \Big| \mathcal{F}_k\right\}$$

(100)

where $(a)$ follows from (99), $(b)$ follows from the last identity in (52), and $(c)$ is just definition. It follows from (97) and (100) that:

$$(n_X - 1)\mathsf{E}\left\{\left[L(\theta^{(k-1)}, w^{(k)} - w^{(k-1)}) - L(\theta^*, w^{(k)} - w^{(k-1)})\right] \Big| \mathcal{F}_k\right\}$$

$$\leq \frac{\beta_2 B_f^2 (n_X - 1)^2}{n_X^2} \mathsf{E}\left\{\left\|w^{(k)} - w^{(k-1)}\right\|^2 \Big| \mathcal{F}_k\right\} + \frac{1}{\beta_2}\|\theta^{(k-1)} - \theta^*\|^2$$

(101)

Next consider the remaining $L$ terms (lines $4$ and $5$ of (96)). We have:

$$L(\theta^{(k-1)}, w^{(k)} - w^*) - L(\theta^*, w^{(k)} - w^*) + L(\theta^{(k)}, w^*) - L(\theta^*, w^*)$$
$$= L(\theta^{(k-1)}, w^{(k)}) - L(\theta^*, w^{(k)}) + L(\theta^*, w^*) - L(\theta^{(k-1)}, w^*) + L(\theta^{(k)}, w^*) - L(\theta^*, w^*)$$
$$= L(\theta^{(k-1)}, w^{(k)}) - L(\theta^*, w^{(k)}) + L(\theta^{(k)}, w^*) - L(\theta^{(k-1)}, w^*)$$

(102)

Therefore, the remaining $L$ terms of (96) can be bounded as:

$$L(\theta^{(k)}, w^*) - L(\theta^*, w^*) + L(\theta^{(k-1)}, w^{(k)} - w^*) - L(\theta^*, w^{(k)} - w^*) - \langle L_\theta'(\theta^{(k-1)}, w^{(k)}), \theta^{(k)} - \theta^*\rangle$$

$$\stackrel{(a)}{=} L(\theta^{(k-1)}, w^{(k)}) - L(\theta^*, w^{(k)}) + L(\theta^{(k)}, w^*) - L(\theta^{(k-1)}, w^*) - \langle L_\theta'(\theta^{(k-1)}, w^{(k)}), \theta^{(k)} - \theta^*\rangle$$

$$\stackrel{(b)}{\leq} \langle L_\theta'(\theta^{(k-1)}, w^{(k)}), \theta^{(k-1)} - \theta^*\rangle + \langle L_\theta'(\theta^{(k)}, w^*), \theta^{(k)} - \theta^{(k-1)}\rangle - \langle L_\theta'(\theta^{(k-1)}, w^{(k)}), \theta^{(k)} - \theta^*\rangle$$

$$\stackrel{(c)}{=} -\langle L_\theta'(\theta^{(k-1)}, w^{(k)}), \theta^{(k)} - \theta^{(k-1)}\rangle + \langle L_\theta'(\theta^{(k)}, w^*), \theta^{(k)} - \theta^{(k-1)}\rangle$$

$$\stackrel{(d)}{=} -\langle L_\theta'(\theta^{(k-1)}, w^{(k)}), \theta^{(k)} - \theta^{(k-1)}\rangle + \langle L_\theta'(\theta^{(k-1)}, w^*), \theta^{(k)} - \theta^{(k-1)}\rangle$$
$$\quad - \langle L_\theta'(\theta^{(k-1)}, w^*), \theta^{(k)} - \theta^{(k-1)}\rangle + \langle L_\theta'(\theta^{(k)}, w^*), \theta^{(k)} - \theta^{(k-1)}\rangle$$

$$\stackrel{(e)}{=} \langle L_\theta'(\theta^{(k-1)}, w^*) - L_\theta'(\theta^{(k-1)}, w^{(k)}), \theta^{(k)} - \theta^{(k-1)}\rangle$$
$$\quad + \langle L_\theta'(\theta^{(k)}, w^*) - L_\theta'(\theta^{(k-1)}, w^*), \theta^{(k)} - \theta^{(k-1)}\rangle$$

$$\stackrel{(f)}{=} -\langle L_\theta'(\theta^{(k-1)}, w^{(k)} - w^*), \theta^{(k)} - \theta^{(k-1)}\rangle$$
$$\quad + \langle L_\theta'(\theta^{(k)}, w^*) - L_\theta'(\theta^{(k-1)}, w^*), \theta^{(k)} - \theta^{(k-1)}\rangle$$

(103)

where step $(a)$ substitutes (102), step $(b)$ uses the convexity of $L(\theta, w)$ in $\theta$ by applying $f(x) - f(y) \leq \langle f'(x), x - y\rangle$, step $(c)$ merges the first and the third terms in line $(b)$, step $(d)$ adds and subtracts the same term (i.e., the second and the third terms), step $(e)$ merges the first term with the second term and also merges the third term and the fourth term, step $(f)$ uses the linearity of $L(\theta, w)$ in $w$. We now proceed to bound the two terms in (103). For a $\beta_1 > 0$ (to be determined later), the first term in (103) can be upper bounded as

$$\left|\langle L_\theta'(\theta^{(k-1)}, w^{(k)} - w^*), \theta^{(k)} - \theta^{(k-1)}\rangle\right|$$

$$\stackrel{(a)}{\leq} \frac{1}{\beta_1}\|L_\theta'(\theta^{(k-1)}, w^{(k)} - w^*)\|^2 + \beta_1\|\theta^{(k)} - \theta^{(k-1)}\|^2$$

$$\overset{(b)}{\leq} \frac{B_f^2}{\beta_1 n_X}\big\|w^{(k)} - w^*\big\|^2 + \beta_1\big\|\theta^{(k)} - \theta^{(k-1)}\big\|^2 \tag{104}$$

where $(a)$ uses Cauchy-Schwartz inequality and the fact that $a^2 + b^2 \geq 2ab \geq ab$, and step $(b)$ uses the definition of $L_\theta'$ in (25) and Jensen's inequality for $\|\cdot\|^2$. Next, the second term in (103) can be upper bounded as

$$\big|\big\langle L_\theta'(\theta^{(k)}, w^*) - L_\theta'(\theta^{(k-1)}, w^*), \theta^{(k)} - \theta^{(k-1)}\big\rangle\big|$$

$$\overset{(a)}{\leq} \big\|L_\theta'(\theta^{(k)}, w^*) - L_\theta'(\theta^{(k-1)}, w^*)\big\| \cdot \big\|\theta^{(k)} - \theta^{(k-1)}\big\|$$

$$\overset{(b)}{\leq} B_\theta B_w \big\|\theta^{(k)} - \theta^{(k-1)}\big\|^2 \tag{105}$$

where step $(a)$ uses Cauchy-Schwartz inequality and step $(b)$ uses Lipschitz condition of the gradient $f_\theta'$ together with the boundedness of $w^*$ (Lemma E.7). Substituting (104)–(105) into (103), we obtain

$$L(\theta^{(k)}, w^*) - L(\theta^*, w^*) + L\big(\theta^{(k-1)}, w^{(k)} - w^*\big) - L\big(\theta^*, w^{(k)} - w^*\big) - \big\langle L_\theta'(\theta^{(k-1)}, w^{(k)}), \theta^{(k)} - \theta^*\big\rangle$$

$$\leq \frac{B_f^2}{\beta_1 n_X}\big\|w^{(k)} - w^*\big\|^2 + \beta_1\big\|\theta^{(k)} - \theta^{(k-1)}\big\|^2 + B_\theta B_w\big\|\theta^{(k)} - \theta^{(k-1)}\big\|^2$$

$$= \frac{B_f^2}{\beta_1 n_X}\big\|w^{(k)} - w^*\big\|^2 + (\beta_1 + B_\theta B_w)\big\|\theta^{(k)} - \theta^{(k-1)}\big\|^2 \tag{106}$$

We have now bounded all the $L$ terms in (96).

Substituting both (101) and (106) in (96), we get the final bound, without the $L$ terms as follows:

$$\left(\frac{1}{2\alpha_w} + \gamma\right)\mathsf{E}\{\|w^{(k)} - w^*\|^2 \mid \mathcal{F}_k\} + \frac{1}{2\alpha_w}\mathsf{E}\{\|w^{(k)} - w^{(k-1)}\|^2 \mid \mathcal{F}_k\}$$

$$\left(\frac{1}{2\alpha_\theta} + \mu\right)\mathsf{E}\big\{\|\theta^{(k)} - \theta^*\|^2\big|\mathcal{F}_k\big\} + \frac{1}{2\alpha_\theta}\mathsf{E}\big\{\|\theta^{(k)} - \theta^{(k-1)}\|^2\big|\mathcal{F}_k\big\}$$

$$\leq \left(\frac{1}{2\alpha_w} + \frac{\gamma(n_X - 1)}{n_X}\right)\|w^{(k-1)} - w^*\|^2 + \frac{1}{2\alpha_\theta}\|\theta^{(k-1)} - \theta^*\|^2 \tag{107}$$

$$+ \left(2\alpha_w B_f^2\big(1 - \overline{1/n_Y}\big) + \alpha_\theta B_w^2 B_\theta^2\right)\|\theta^{(k-1)} - \widetilde{\theta}\|^2$$

$$+ \frac{\beta_2 B_f^2(n_X - 1)^2}{n_X^2}\mathsf{E}\big\{\|w^{(k)} - w^{(k-1)})\|^2\big|\mathcal{F}_k\big\} + \frac{1}{\beta_2}\|\theta^{(k-1)} - \theta^*\|^2$$

$$+ \frac{B_f^2}{\beta_1 n_X}\mathsf{E}\big\{\|w^{(k)} - w^*\|^2\big|\mathcal{F}_k\big\} + (\beta_1 + B_\theta B_w)\mathsf{E}\big\{\|\theta^{(k)} - \theta^{(k-1)}\|^2\big|\mathcal{F}_k\big\}$$

Combining common terms and rearranging,

$$\left(\frac{1}{2\alpha_\theta} + \mu\right)\mathsf{E}\big\{\|\theta^{(k)} - \theta^*\|^2\big|\mathcal{F}_k\big\} + \left(\frac{1}{2\alpha_w} + \gamma - \frac{B_f^2}{\beta_1 n_X}\right)\mathsf{E}\{\|w^{(k)} - w^*\|^2 \mid \mathcal{F}_k\}$$

$$\leq \left(\frac{1}{2\alpha_w} + \frac{\gamma(n_X - 1)}{n_X}\right)\|w^{(k-1)} - w^*\|^2$$

$$+ \left(\frac{1}{2\alpha_\theta} + \frac{1}{\beta_2} + 4\alpha_w B_f^2\big(1 - \overline{1/n_Y}\big) + 2\alpha_\theta B_w^2 B_\theta^2\right)\|\theta^{(k-1)} - \theta^*\|^2 \tag{108}$$

$$+ \left(\beta_1 + B_\theta B_w - \frac{1}{2\alpha_\theta}\right)\mathsf{E}\big\{\|\theta^{(k)} - \theta^{(k-1)}\|^2\big|\mathcal{F}_k\big\}$$

$$+ \left(\frac{\beta_2 B_f^2(n_X - 1)^2}{n_X^2} - \frac{1}{2\alpha_w}\right)\mathsf{E}\{\|w^{(k)} - w^{(k-1)}\|^2 \mid \mathcal{F}_k\}$$

$$+ \left(4\alpha_w B_f^2\big(1 - \overline{1/n_Y}\big) + 2\alpha_\theta B_w^2 B_\theta^2\right)\|\widetilde{\theta} - \theta^*\|^2$$

where we have also used the inequality $(a + b)^2 \le 2a^2 + 2b^2$ for the $\|\theta^{(k-1)} - \widetilde{\theta}\|^2$ term.

We now choose the step-sizes $\alpha_\theta$, $\alpha_w$ and $M$ as in Theorem 4.6:

$$\alpha_\theta = \left(\frac{25B_f^2}{\gamma} + 10B_\theta B_w + \frac{80B_w^2 B_\theta^2}{\mu}\right)^{-1} \tag{109}$$

$$\alpha_w = \frac{\mu}{40B_f^2} \tag{110}$$

$$M = \max\left(\frac{10}{\alpha_\theta \mu}, \frac{2n_X}{\alpha_w \gamma}, 4n_X\right) \tag{111}$$

The choice of $\alpha_\theta$ in (109) ensures the following three bounds:

$$\alpha_\theta \le \frac{\gamma}{25B_f^2} \quad \text{or} \quad \alpha_\theta \le \frac{1}{10B_\theta B_w} \quad \text{or} \quad \alpha_\theta \le \frac{\mu}{80B_w^2 B_\theta^2} \tag{112}$$

Furthermore, we choose $\beta_1$ and $\beta_2$ as follows:

$$\beta_1 = -B_\theta B_w + \frac{1}{2\alpha_\theta} \tag{113}$$

$$\beta_2 = \frac{1}{2\alpha_w B_f^2} \tag{114}$$

The second inequality in (112) will ensure positivity of $\beta_1$.

**Applying the above choice of hyper-parameters:** Based on (113) we have:

$$\beta_1 + B_\theta B_w - \frac{1}{2\alpha_\theta} = 0 \tag{115}$$

and using (114):

$$\begin{aligned}
\frac{\beta_2 B_f^2 (n_X - 1)^2}{n_X^2} - \frac{1}{2\alpha_w} &= \frac{1}{2\alpha_w B_f^2} \frac{B_f^2 (n_X - 1)^2}{n_X^2} - \frac{1}{2\alpha_w} \\
&= \frac{1}{2\alpha_w} \frac{(n_X - 1)^2}{n_X^2} - \frac{1}{2\alpha_w} \\
&< 0
\end{aligned} \tag{116}$$

Equations (115) and (116) will ensure that the third and fourth terms on the right hand side of (108) are either 0 or negative, and therefore can be ignored, reducing the bound in (108) to:

$$\begin{aligned}
&\left(\frac{1}{2\alpha_\theta} + \mu\right) \mathsf{E}\left\{\|\theta^{(k)} - \theta^*\|^2 \big| \mathcal{F}_k\right\} + \left(\frac{1}{2\alpha_w} + \gamma - \frac{B_f^2}{\beta_1 n_X}\right) \mathsf{E}\left\{\|w^{(k)} - w^*\|^2 \mid \mathcal{F}_k\right\} \\
&\le \left(\frac{1}{2\alpha_w} + \frac{\gamma(n_X - 1)}{n_X}\right) \|w^{(k-1)} - w^*\|^2 \\
&\quad + \left(\frac{1}{\beta_2} + \frac{1}{2\alpha_\theta} + 4\alpha_w B_f^2 + 2\alpha_\theta B_w^2 B_\theta^2\right) \|\theta^{(k-1)} - \theta^*\|^2 \\
&\quad + \left(4\alpha_w B_f^2 + 2\alpha_\theta B_w^2 B_\theta^2\right) \|\widetilde{\theta} - \theta^*\|^2
\end{aligned} \tag{117}$$

where we have also used $(1 - \overline{1/n_Y}) \le 1$.

The $\widetilde{\theta}$ in the (117) is $\widetilde{\theta}_{s-1}$, the fixed primal variable at the beginning of stage $s$. Denote $\widetilde{\theta}_s$ to be the primal variable randomly chosen among $\theta^{(k)}$ for $1 \le k \le M$ at the end of stage $s$. We define $\widetilde{w}_{s-1}$ and $\widetilde{w}_s$ in a similar manner (though neither of them are used in the algorithm), and also note that $\theta^{(0)}$

and $w^{(0)}$ are initialized to $\widetilde{\theta}_{s-1}$ and $\widetilde{w}_{s-1}$ at the beginning of stage $s$. Summing both sides of (117) over all $1 \leq k \leq M$,

$$
\begin{aligned}
&\left(\frac{1}{2\alpha_\theta} + \mu\right) \mathsf{E}\left\{\|\theta^{(M)} - \theta^*\|^2 \big| \mathcal{F}_s\right\} + \left(\frac{1}{2\alpha_w} + \gamma - \frac{B_f^2}{\beta_1 n_X}\right) \mathsf{E}\{\|w^{(M)} - w^*\|^2 \mid \mathcal{F}_s\} \\
&+ \left(\mu - \frac{1}{\beta_2} - 4\alpha_w B_f^2 - 2\alpha_\theta B_w^2 B_\theta^2\right) \sum_{k=1}^{M-1} \mathsf{E}\left\{\|\theta^{(k)} - \theta^*\|^2 \big| \mathcal{F}_s\right\} \\
&+ \left(\frac{\gamma}{n_X} - \frac{B_f^2}{\beta_1 n_X}\right) \sum_{k=1}^{M-1} \mathsf{E}\{\|w^{(k)} - w^*\|^2 \mid \mathcal{F}_s\} \\
&\leq \left(\frac{1}{2\alpha_w} + \frac{\gamma(n_X - 1)}{n_X}\right) \|w^{(0)} - w^*\|^2 + \left(\frac{1}{\beta_2} + \frac{1}{2\alpha_\theta}\right) \|\theta^{(0)} - \theta^*\|^2 \\
&+ M\left(4\alpha_w B_f^2 + 2\alpha_\theta B_w^2 B_\theta^2\right) \|\widetilde{\theta} - \theta^*\|^2
\end{aligned}
\tag{118}
$$

Substituting $w^{(0)} = \widetilde{w}_{s-1}$ and $\theta^{(0)} = \widetilde{\theta}_{s-1}$, and also noting that the first two terms on the left hand side of (118) can be combined with the second two terms (note that the difference in coefficients are positive, and positive terms on the left hand side of the inequality can be ignored):

$$
\begin{aligned}
&\left(\mu - \frac{1}{\beta_2} - 4\alpha_w B_f^2 - 2\alpha_\theta B_w^2 B_\theta^2\right) \sum_{k=1}^{M} \mathsf{E}\left\{\|\theta^{(k)} - \theta^*\|^2 \big| \mathcal{F}_s\right\} \\
&+ \left(\frac{\gamma}{n_X} - \frac{B_f^2}{\beta_1 n_X}\right) \sum_{k=1}^{M} \mathsf{E}\{\|w^{(k)} - w^*\|^2 \mid \mathcal{F}_s\} \\
&\leq \left(\frac{1}{2\alpha_w} + \frac{\gamma(n_X - 1)}{n_X}\right) \|\widetilde{w}_{s-1} - w^*\|^2 \\
&+ \left(\frac{1}{\beta_2} + \frac{1}{2\alpha_\theta} + M\left(4\alpha_w B_f^2 + 2\alpha_\theta B_w^2 B_\theta^2\right)\right) \|\widetilde{\theta}_{s-1} - \theta^*\|^2
\end{aligned}
\tag{119}
$$

Dividing both sides of (119) by $M$ and applying Jensen's inequality on the left hand side, we obtain:

$$
\begin{aligned}
&\left(\mu - \frac{1}{\beta_2} - 4\alpha_w B_f^2 - 2\alpha_\theta B_w^2 B_\theta^2\right) \mathsf{E}\left\{\|\widetilde{\theta}_s - \theta^*\|^2 \big| \mathcal{F}_s\right\} \\
&+ \left(\frac{\gamma}{n_X} - \frac{B_f^2}{\beta_1 n_X}\right) \mathsf{E}\{\|\widetilde{w}_s - w^*\|^2 \mid \mathcal{F}_s\} \\
&\leq \left(\frac{1}{2M\alpha_w} + \frac{\gamma(n_X - 1)}{M n_X}\right) \|\widetilde{w}_{s-1} - w^*\|^2 \\
&+ \left(\frac{1}{M\beta_2} + \frac{1}{2M\alpha_\theta} + 4\alpha_w B_f^2 + 2\alpha_\theta B_w^2 B_\theta^2\right) \|\widetilde{\theta}_{s-1} - \theta^*\|^2
\end{aligned}
\tag{120}
$$

We now substitute the hyper-parameter values in (109) (110) (111) (113) (114) in (120) to obtain a linear rate.

Substituting these values in the coefficient of the first term on the left hand side of (120) (we're using the third bound for $\alpha_\theta$ in (112) here):

$$
\begin{aligned}
\mu - \frac{1}{\beta_2} - 4\alpha_w B_f^2 - 2\alpha_\theta B_w^2 B_\theta^2 &= \mu - 2\alpha_w B_f^2 - 4\alpha_w B_f^2 - 2\alpha_\theta B_w^2 B_\theta^2 \\
&\geq \mu - 2\frac{\mu}{40 B_f^2} B_f^2 - 4\frac{\mu}{40 B_f^2} B_f^2 - 2\frac{\mu}{80 B_w^2 B_\theta^2} B_w^2 B_\theta^2 \\
&= \mu - \frac{\mu}{20} - \frac{\mu}{10} - \frac{\mu}{40} \\
&= \mu - \frac{7\mu}{40} \\
&= \frac{33\mu}{40} \\
&\geq \frac{4\mu}{5}
\end{aligned}
\tag{121}
$$

Next, substituting for the coefficient of the second term on the left hand side of (120) (here we use the first and second bounds for $\alpha_\theta$ in (112), both in the inequality in the fourth line):

$$
\begin{aligned}
\frac{\gamma}{n_X} - \frac{B_f^2}{\beta_1 n_X} &= \frac{\gamma}{n_X} - \frac{B_f^2}{n_X}\left(-B_\theta B_w + \frac{1}{2\alpha_\theta}\right)^{-1} \\
&= \frac{\gamma}{n_X} - \frac{B_f^2}{n_X}\left(\frac{1 - 2\alpha_\theta B_\theta B_w}{2\alpha_\theta}\right)^{-1} \\
&= \frac{\gamma}{n_X} - \frac{B_f^2}{n_X}\left(\frac{2\alpha_\theta}{1 - 2\alpha_\theta B_\theta B_w}\right) \\
&\geq \frac{\gamma}{n_X} - \frac{B_f^2}{n_X} \times \frac{2\gamma}{25 B_f^2} \times \frac{1}{1 - \frac{2}{10 B_\theta B_w} B_\theta B_w} \\
&= \frac{\gamma}{n_X} - \frac{2\gamma}{25 n_X} \times \frac{5}{4} \\
&= \frac{\gamma}{n_X} - \frac{\gamma}{10 n_X} \\
&\geq \frac{4\gamma}{5 n_X}
\end{aligned}
\tag{122}
$$

Next consider the coefficient of the first term on the right hand side of (120) (we use the second and third values of $M$ in (111), both in the second line):

$$
\begin{aligned}
\frac{1}{2M\alpha_w} + \frac{\gamma(n_X - 1)}{M n_X} &\leq \frac{1}{2M\alpha_w} + \frac{\gamma}{M} \\
&\leq \frac{\gamma}{4n_X} + \frac{\gamma}{4n_X} \\
&= \frac{\gamma}{2n_X}
\end{aligned}
\tag{123}
$$

Finally, we consider the coefficient of the second term on the right hand side of (120) (we use the third bound for $\alpha_\theta$ in (112), and the first and third definitions of $M$ in (111)):

$$
\begin{aligned}
\frac{1}{M\beta_2} + \frac{1}{2M\alpha_\theta} + 4\alpha_w B_f^2 + 2\alpha_\theta B_w^2 B_\theta^2 &\leq \frac{2\alpha_w B_f^2}{M} + \frac{\mu}{20} + 4\frac{\mu}{40 B_f^2} B_f^2 + \frac{\mu}{80 B_w^2 B_\theta^2} 2 B_w^2 B_\theta^2 \\
&\leq \frac{\mu}{40 B_f^2} 2 B_f^2 + \frac{\mu}{20} + \frac{\mu}{10} + \frac{\mu}{40} \\
&= \frac{9\mu}{40} \\
&= \frac{\mu}{4}
\end{aligned}
\tag{124}
$$

Using (121), (122), (123), and (124) in (120), we have:

$$\frac{4\mu}{5}\mathsf{E}\Big\{\|\widetilde{\theta}_s - \theta^*\|^2 \big| \mathcal{F}_s\Big\} + \frac{4\gamma}{5n_X}\mathsf{E}\{\|\widetilde{w}_s - w^*\|^2 \mid \mathcal{F}_s\}$$
$$\leq \frac{\gamma}{2n_X}\|\widetilde{w}_{s-1} - w^*\|^2 + \frac{\mu}{4}\|\widetilde{\theta}_{s-1} - \theta^*\|^2 \tag{125}$$

Applying full expectation to both sides of the above inequality, we obtain

$$\frac{4\mu}{5}\mathsf{E}\|\widetilde{\theta}_s - \theta^*\|^2 + \frac{4\gamma}{5n_X}\mathsf{E}\|\widetilde{w}_s - w^*\|^2$$
$$\leq \frac{\gamma}{2n_X}\mathsf{E}\|\widetilde{w}_{s-1} - w^*\|^2 + \frac{\mu}{4}\mathsf{E}\|\widetilde{\theta}_{s-1} - \theta^*\|^2. \tag{126}$$

Dividing both sides by $4\mu/5$, we have

$$\mathsf{E}\|\widetilde{\theta}_s - \theta^*\|^2 + \frac{\gamma}{n_X\mu}\mathsf{E}\|\widetilde{w}_s - w^*\|^2$$
$$\leq \frac{5}{16}\mathsf{E}\|\widetilde{w}_{s-1} - w^*\|^2 + \frac{5\gamma}{8\mu n_X}\mathsf{E}\|\widetilde{\theta}_{s-1} - \theta^*\|^2, \tag{127}$$

which can be further bounded as

$$\mathsf{E}\|\widetilde{\theta}_s - \theta^*\|^2 + \frac{\gamma}{n_X\mu}\mathsf{E}\|\widetilde{w}_s - w^*\|^2$$
$$\leq \frac{5}{8}\Big(\mathsf{E}\|\widetilde{w}_{s-1} - w^*\|^2 + \frac{\gamma}{\mu n_X}\mathsf{E}\|\widetilde{\theta}_{s-1} - \theta^*\|^2\Big). \tag{128}$$

Define the Lyapunov function $P_s$ to be

$$P_s = \mathsf{E}\|\widetilde{\theta}_s - \theta^*\|^2 + \frac{\gamma}{n_X\mu}\mathsf{E}\|\widetilde{w}_s - w^*\|^2.$$

Then, inequality (126) can be expressed as

$$P_s \leq \frac{5}{8}P_{s-1} \leq \Big(\frac{5}{8}\Big)^s P_0. \tag{129}$$

Therefore, $P_s$ converges to zero at a linear rate of $5/8$. In order to achieve $\epsilon$-precision solution (i.e., $P_s \leq \epsilon$), it requires a total of $O(\ln\frac{1}{\epsilon})$ outer-loop iterations (stages). And for each outer loop iteration, it requires $M$ steps of inner-loop primal-dual updates, which is $O(1)$ per step (in number of oracle calls), and $O(n_X n_Y)$ for evaluating the batch gradients for the control variates, where $n_Y = (n_{Y_0} + \cdots + n_{Y_{n_X - 1}})/n_X$. Therefore, the complexity per outer loop iteration is $O(n_X n_Y + M)$ so that the total complexity can be written as:

$$O\Big((n_X n_Y + M)\ln\Big(\frac{P_0}{\epsilon}\Big)\Big). \tag{130}$$

Recall from (111) that $M$ is given by

$$M = 10/\mu\alpha_\theta + \frac{2n_X}{\alpha_w\gamma} + 4n_X.$$

where, by (109) and (110), the step-sizes $\alpha_\theta$ and $\theta_w$ are given by

$$\alpha_\theta = \Big(\frac{25B_f^2}{\gamma} + 10B_\theta B_w + \frac{80B_w^2 B_\theta^2}{\mu}\Big)^{-1}, \quad \alpha_w = \frac{\mu}{40B_f^2}.$$

This implies that $M = O(B_f^2/\mu\gamma + B_w^2 B_\theta^2/\mu^2 + (B_f^2/\mu\gamma)n_X + n_X)$. In consequence, the total complexity is

$$O\Big((n_X n_Y + n_X \kappa + n_X)\ln\frac{1}{\epsilon}\Big), \tag{131}$$

where

$$\kappa = B_f^2/\gamma\mu + B_w^2 B_\theta^2/\mu^2. \tag{132}$$

Noting that $\mathsf{E}\|\widetilde{\theta}_s - \theta^*\|^2 \leq P_s$, the bound (129) implies that $\mathsf{E}\|\widetilde{\theta}_s - \theta^*\|^2$ also converges to zero at a linear rate of $5/8$ and the total complexity to reach $\mathsf{E}\|\widetilde{\theta}_s - \theta^*\|^2 \leq \epsilon$ is also given by (131).

# F  Special case: SVRPDA-I with $n_{Y_i} \equiv 1$, $f_\theta$ linear in $\theta$ and no outer loop

First, we observe that in this special case, our SVRPDA-I algorithm will become a single-loop algorithm, and that the outer-loop in Algorithm 1 is no longer needed. To see this, first note that when $n_{Y_i} \equiv 1$, $\delta_k^w$ is independent of $\widetilde\theta$ because the last two terms in (12) would cancel each other. Second, when $f_\theta$ is linear in $\theta$, the term $\overline{f}'_{i_k}(\widetilde\theta)$ in (14) and $U_0$ in (11) are independent of $\widetilde\theta$, which further implies that $U_k$ (that is recursively defined in (14)) is also independent of $\widetilde\theta$. Finally, we also note that, with linear $f_\theta$, the first two terms in (15) cancel with each other, so that $\delta_k^\theta \equiv U_k$ is independent of $\widetilde\theta$. As a result, the inner loop in Algorithm 1 does not require an outer-loop to update the reference variable $\widetilde\theta$.

The following theorem establishes the complexity bound for the SVRPDA-I algorithm in this special case.

**Theorem F.1.** *Suppose Assumptions 4.1–4.4 hold. Furthermore, suppose $n_{Y_i} = 1$, $1 \le i \le n_X$, and $f_\theta$ is a linear function of $\theta$. Consider just the the inner loop of Algorithm 1, with s = 1 fixed, and*

$$\alpha_\theta = \frac{\gamma}{16B_f^2 + 4n_X\mu\gamma}, \quad and \quad \alpha_w = \frac{1}{2\gamma}\frac{3n_X + \kappa + 1}{n_X + \kappa + 1}$$

*where $\kappa = B_f^2/\gamma\mu$ is the condition number. Then, the Lyapunov function*

$$\Delta^{(k)} := \left(\frac{1}{2\alpha_\theta}+\mu\right)\mathsf{E}\Big\{\|\theta^{(k)}-\theta^*\|^2\Big|\mathcal{F}_k\Big\} + \left(\frac{1}{2\alpha_w}+\gamma\right)\mathsf{E}\Big\{\|w^{(k)}-w^*\|^2\Big|\mathcal{F}_k\Big\}$$

*satisfies $\Delta^{(k)} \le \big(1-1/(1+2\kappa+2n_X)\big)^k \Delta^{(0)}$. Furthermore, the overall computational cost (in number of oracle calls) for reaching $\Delta^{(k)} \le \epsilon$ is upper bounded by*

$$O\Big(\big(n_X + \kappa\big)\ln\Big(\frac{1}{\epsilon}\Big)\Big). \tag{133}$$

In comparison, the authors in [5] propose a stochastic primal dual coordinate (SPDC) algorithm for this special case and prove an overall complexity of $O\big((n_X + \sqrt{n_X\kappa})\ln\big(\frac{1}{\epsilon}\big)\big)$ to achieve an $\epsilon$-error solution, where the condition number $\kappa = B_f^2/\mu\gamma$. This is by far the best complexity for this class of problems. It is interesting to note that the complexity result in (133) and the complexity result in [5] only differ in their dependency on $\kappa$. This difference is most likely due to the acceleration technique that is employed in the primal update of the SPDC algorithm. We conjecture that the dependency on the condition number of SVRPDA-I can be further improved using a similar acceleration technique.

## F.1  Proof of Theorem F.1

It is useful to first discuss the main implications of choosing $f_\theta$ to be linear in $\theta$ and $n_{Y_i} = 1$ for all $i$. First, based on Assumption E.3 (or equivalently **Assumption 4.3**), we have $B_\theta = 0$, since $f_\theta'$ is independent of $\theta$. This also implies that $L_\theta'$ is independent of $\theta$, and therefore, Assumption E.4 (or equivalently **Assumption 4.4**) holds with equality:

$$L(\theta_1, w) - L(\theta_2, w) = \langle L_\theta'(\theta_2, w),\ \theta_1 - \theta_2 \rangle. \tag{134}$$

In particular, for any $\theta \in \mathbb{R}^d$ and $w \in \mathbb{R}^\ell$, $L(\theta, w) = \langle L_\theta'(\theta, w),\ \theta \rangle$. Finally, $n_{Y_i} = 1$ implies $\overline{1/n_Y} = \frac{1}{n_X}\sum_{i=0}^{n_X-1} 1/n_{Y_i} = 1$.

Using the above implications in the primal bound (71) (in particular, letting $B_\theta = 0$, and using linearity of $L(\theta, w)$), we obtain the primal bound for the special case as follows:

$$\left(\frac{1}{2\alpha_\theta}+\mu\right)\mathsf{E}\Big\{\|\theta^{(k)}-\theta^*\|^2\big|\mathcal{F}_k\Big\} + \frac{1}{2\alpha_\theta}\mathsf{E}\Big\{\|\theta^{(k)}-\theta^{(k-1)}\|^2\big|\mathcal{F}_k\Big\}$$

$$\le \frac{1}{2\alpha_\theta}\|\theta^{(k-1)}-\theta^*\|^2 - \mathsf{E}\Big\{\Big[L(\theta^{(k)}, w^{(k)}-w^*) - L(\theta^*, w^{(k)}-w^*)\Big]\big|\mathcal{F}_k\Big\} \tag{135}$$

Similarly, using the above implications in the dual bound (56) (in particular, letting $(1 - \overline{1/n_Y}) = 0$), the dual bound for the special case becomes:

$$\left(\frac{1}{2\alpha_w}+\gamma\right)\mathsf{E}\{\|w^{(k)}-w^*\|^2 \mid \mathcal{F}_k\} + \frac{1}{2\alpha_w}\mathsf{E}\{\|w^{(k)}-w^{(k-1)}\|^2 \mid \mathcal{F}_k\}$$

$$\leq \left( \frac{1}{2\alpha_w} + \frac{\gamma(n_X - 1)}{n_X} \right) \|w^{(k-1)} - w^*\|^2$$

$$+ \mathsf{E}\Big\{ L(\theta^{(k-1)}, w^{(k-1)} - w^*) - L(\theta^*, w^{(k-1)} - w^*) \mid \mathcal{F}_k \Big\}$$

$$+ n_X \mathsf{E}\Big\{ L(\theta^{(k-1)}, w^{(k)} - w^{(k-1)}) - L(\theta^*, w^{(k)} - w^{(k-1)}) \mid \mathcal{F}_k \Big\} \qquad (136)$$

Adding (135) and (136), we obtain the combined primal-dual bound:

$$\left( \frac{1}{2\alpha_\theta} + \mu \right) \mathsf{E}\Big\{ \|\theta^{(k)} - \theta^*\|^2 \big| \mathcal{F}_k \Big\} + \frac{1}{2\alpha_\theta} \mathsf{E}\Big\{ \|\theta^{(k)} - \theta^{(k-1)}\|^2 \big| \mathcal{F}_k \Big\}$$

$$+ \left( \frac{1}{2\alpha_w} + \gamma \right) \mathsf{E}\{ \|w^{(k)} - w^*\|^2 \mid \mathcal{F}_k \} + \frac{1}{2\alpha_w} \mathsf{E}\{ \|w^{(k)} - w^{(k-1)}\|^2 \mid \mathcal{F}_k \}$$

$$\leq \frac{1}{2\alpha_\theta} \|\theta^{(k-1)} - \theta^*\|^2 + \left( \frac{1}{2\alpha_w} + \frac{\gamma(n_X - 1)}{n_X} \right) \|w^{(k-1)} - w^*\|^2$$

$$- \mathsf{E}\Big\{ \Big[ L(\theta^{(k)}, w^{(k)} - w^*) - L(\theta^*, w^{(k)} - w^*) \Big] \big| \mathcal{F}_k \Big\}$$

$$+ \mathsf{E}\Big\{ L(\theta^{(k-1)}, w^{(k-1)} - w^*) - L(\theta^*, w^{(k-1)} - w^*) \mid \mathcal{F}_k \Big\}$$

$$+ n_X \mathsf{E}\Big\{ L(\theta^{(k-1)}, w^{(k)} - w^{(k-1)}) - L(\theta^*, w^{(k)} - w^{(k-1)}) \mid \mathcal{F}_k \Big\} \qquad (137)$$

As done in the previous proofs, we will first consider all the $L$ terms that appear on the right hand side of (137). Following exactly the same steps as (73), (74), (75) and (76), we obtain the final bound for the $L$ terms as given in (77), but with $B_\theta = 0$. We still write out the whole simplification details here for completeness, and also to show how this special case is much simpler than the more general case.

Considering all the $L$ terms in (137), we have:

$$- \Big[ L(\theta^{(k)}, w^{(k)} - w^*) - L(\theta^*, w^{(k)} - w^*) \Big] + \Big[ L(\theta^{(k-1)}, w^{(k-1)} - w^*) - L(\theta^*, w^{(k-1)} - w^*) \Big]$$

$$+ n_X \Big[ L(\theta^{(k-1)}, w^{(k)} - w^{(k-1)}) - L(\theta^*, w^{(k)} - w^{(k-1)}) \Big]$$

$$= \Big[ L(\theta^{(k-1)}, w^{(k-1)} - w^*) - L(\theta^{(k)}, w^{(k-1)} - w^*) \Big]$$

$$+ \Big[ L(\theta^{(k)}, w^{(k-1)} - w^{(k)}) - L(\theta^*, w^{(k-1)} - w^{(k)}) \Big]$$

$$+ n_X \Big[ L(\theta^{(k-1)}, w^{(k)} - w^{(k-1)}) - L(\theta^*, w^{(k)} - w^{(k-1)}) \Big]$$

$$= \Big[ L(\theta^{(k-1)}, w^{(k-1)} - w^*) - L(\theta^{(k)}, w^{(k-1)} - w^*) \Big]$$

$$+ \Big[ L(\theta^{(k)}, w^{(k-1)} - w^{(k)}) - L(\theta^{(k-1)}, w^{(k-1)} - w^{(k)}) \Big]$$

$$+ (n_X - 1) \Big[ L(\theta^{(k-1)}, w^{(k)} - w^{(k-1)}) - L(\theta^*, w^{(k)} - w^{(k-1)}) \Big]$$

$$= \Big[ L(\theta^{(k-1)}, w^{(k)} - w^*) - L(\theta^{(k)}, w^{(k)} - w^*) \Big]$$

$$+ (n_X - 1) \Big[ L(\theta^{(k-1)}, w^{(k)} - w^{(k-1)}) - L(\theta^*, w^{(k)} - w^{(k-1)}) \Big]$$

$$= L\Big( \theta^{(k-1)}, w^{(k)} - w^* + (n_X - 1)(w^{(k)} - w^{(k-1)}) \Big) - L\Big( \theta^{(k)}, w^{(k)} - w^* + (n_X - 1)(w^{(k)} - w^{(k-1)}) \Big)$$

$$\overset{(a)}{\leq} \Big\langle L'_\theta \Big( \theta^{(k-1)}, w^{(k)} - w^* + (n_X - 1)(w^{(k)} - w^{(k-1)}) \Big), \theta^{(k-1)} - \theta^{(k)} \Big\rangle$$

$$= \Big\langle \frac{1}{n_X} \sum_{i=0}^{n_X - 1} \overline{f}'_i(\theta^{(k-1)}) \Big( w_i^{(k)} - w_i^* + (n_X - 1)(w_i^{(k)} - w_i^{(k-1)}) \Big), \theta^{(k-1)} - \theta^{(k)} \Big\rangle$$

$$= \Big\langle \frac{1}{n_X} \sum_{i=0}^{n_X - 1} \overline{f}'_i(\theta^{(k-1)}) \Big( w_i^{(k-1)} + n_X(w_i^{(k)} - w_i^{(k-1)}) - w_i^* \Big), \theta^{(k-1)} - \theta^{(k)} \Big\rangle$$

$$\overset{(b)}{\leq} \beta_1 \Big\| \frac{1}{n_X} \sum_{i=0}^{n_X-1} \overline{f}'_i(\theta^{(k-1)}) \Big( w_i^{(k-1)} + n_X(w_i^{(k)} - w_i^{(k-1)}) - w_i^* \Big) \Big\|^2 + \frac{1}{\beta_1} \Big\| \theta^{(k-1)} - \theta^{(k)} \Big\|^2$$

$$= \beta_1 \Big\| \frac{1}{n_X} \sum_{i=0}^{n_X-1} \overline{f}'_i(\theta^{(k-1)})(w_i^{(k-1)} - w_i^*) + \sum_{i=0}^{n_X-1} \overline{f}'_i(\theta^{(k-1)})(w_i^{(k)} - w_i^{(k-1)}) \Big\|^2 + \frac{1}{\beta_1} \Big\| \theta^{(k-1)} - \theta^{(k)} \Big\|^2$$

$$\overset{(c)}{\leq} 2\beta_1 \Big\| \frac{1}{n_X} \sum_{i=0}^{n_X-1} \overline{f}'_i(\theta^{(k-1)}) \Big( w_i^{(k-1)} - w_i^* \Big) \Big\|^2 + 2\beta_1 \Big\| \sum_{i=0}^{n_X-1} \overline{f}'_i(\theta^{(k-1)})(w_i^{(k)} - w_i^{(k-1)}) \Big\|^2$$

$$+ \frac{1}{\beta_1} \Big\| \theta^{(k-1)} - \theta^{(k)} \Big\|^2$$

$$\overset{(d)}{\leq} \frac{2\beta_1 B_f^2}{n_X} \sum_{i=0}^{n_X-1} \| w_i^{(k-1)} - w_i^* \|^2 + 2\beta_1 \Big\| \sum_{i=0}^{n_X-1} \overline{f}'_i(\theta^{(k-1)})(w_i^{(k)} - w_i^{(k-1)}) \Big\|^2 + \frac{1}{\beta_1} \Big\| \theta^{(k-1)} - \theta^{(k)} \Big\|^2$$

$$\tag{138}$$

where step (a) uses convexity of $L$ in $\theta$, step (b) uses $\langle a, b \rangle \leq \beta_1 \|a\|^2 + \frac{1}{\beta_1}\|b\|^2$, step (c) uses $\|a + b\|^2 \leq 2\|a\|^2 + 2\|b\|^2$, and step (d) applies Jensen's inequality and bounded gradient assumption to the first term. The second term in (138) can be simplified as follows:

$$\mathsf{E}\Big\{ \Big\| \sum_{i=0}^{n_X-1} \overline{f}'_i(\theta^{(k-1)})(w_i^{(k)} - w_i^{(k-1)}) \Big\|^2 \Big| \mathcal{F}_k \Big\} = \sum_{i=0}^{n_X-1} \sum_{j=0}^{n_{Y_i}-1} \frac{1}{n_X n_{Y_i}} \Big\| \overline{f}'_i(\theta^{(k-1)})(w'_{ij} - w_i^{(k-1)}) \Big\|^2$$

$$\leq B_f^2 \sum_{i=0}^{n_X-1} \sum_{j=0}^{n_{Y_i}-1} \frac{1}{n_X n_{Y_i}} \Big\| w'_{ij} - w_i^{(k-1)} \Big\|^2$$

$$= B_f^2 \sum_{i=0}^{n_X-1} \mathsf{E}\{ \| w_i^{(k)} - w_i^{(k-1)} \|^2 \} \tag{139}$$

Therefore, using (138) and (139), the conditional expectation of all the $L$ terms in (137) are bounded by:

$$\frac{2\beta_1 B_f^2}{n_X} \sum_{i=0}^{n_X-1} \| w_i^{(k-1)} - w_i^* \|^2 + 2\beta_1 B_f^2 \sum_{i=0}^{n_X-1} \mathsf{E}\{ \| w_i^{(k)} - w_i^{(k-1)} \|^2 | \mathcal{F}_k \} + \frac{1}{\beta_1} \mathsf{E}\{ \| \theta^{(k-1)} - \theta^{(k)} \|^2 | \mathcal{F}_k \}$$

$$= \frac{2\beta_1 B_f^2}{n_X} \| w^{(k-1)} - w^* \|^2 + 2\beta_1 B_f^2 \mathsf{E}\{ \| w^{(k)} - w^{(k-1)} \|^2 | \mathcal{F}_k \} + \frac{1}{\beta_1} \mathsf{E}\{ \| \theta^{(k-1)} - \theta^{(k)} \|^2 | \mathcal{F}_k \} \tag{140}$$

Substituting the above upper bound for the $L$ terms in (137) leads to

$$\Big( \frac{1}{2\alpha_\theta} + \mu \Big) \mathsf{E}\{ \| \theta^{(k)} - \theta^* \|^2 | \mathcal{F}_k \} + \Big( \frac{1}{2\alpha_w} + \gamma \Big) \mathsf{E}\{ \| w^{(k)} - w^* \|^2 \mid \mathcal{F}_k \}$$

$$+ \Big( \frac{1}{2\alpha_\theta} - \frac{1}{\beta_1} \Big) \mathsf{E}\{ \| \theta^{(k)} - \theta^{(k-1)} \|^2 | \mathcal{F}_k \} + \Big( \frac{1}{2\alpha_w} - 2\beta_1 B_f^2 \Big) \mathsf{E}\{ \| w^{(k)} - w^{(k-1)} \|^2 \mid \mathcal{F}_k \}$$

$$\leq \frac{1}{2\alpha_\theta} \| \theta^{(k-1)} - \theta^* \|^2 + \Big( \frac{1}{2\alpha_w} + \gamma - \frac{\gamma}{n_X} + \frac{2\beta_1 B_f^2}{n_X} \Big) \| w^{(k-1)} - w^* \|^2 \tag{141}$$

which will be the final bound we will analyze.

**Substituting hyper-parameter choices**

Recall the choice of step-sizes $\alpha_\theta$ and $\alpha_w$ defined in Theorem F.1:

$$\alpha_\theta = \frac{\gamma}{16 B_f^2 + 4 n_X \mu \gamma}$$

$$\alpha_w = \frac{1}{2\gamma} \frac{3 n_X + \kappa + 1}{n_X + \kappa + 1} \tag{142}$$

where the condition number $\kappa$ is also defined in Theorem F.1:

$$\kappa = B_f^2/\gamma\mu$$

Furthermore, choosing $\beta_1$ such that

$$\frac{1}{2\alpha_\theta} = \frac{1}{\beta_1}, \tag{143}$$

we have:

$$\begin{aligned}
\alpha_\theta\alpha_w &= \frac{\gamma}{16B_f^2 + 4n_X\mu\gamma} \times \frac{1}{2\gamma}\frac{3n_X + \kappa + 1}{n_X + \kappa + 1} \\
&= \frac{1}{32B_f^2 + 8n_X\mu\gamma} \times \frac{3n_X + \kappa + 1}{n_X + \kappa + 1} \\
&< \frac{1}{32B_f^2 + 8n_X\mu\gamma} \times \frac{3n_X + 3\kappa + 3}{n_X + \kappa + 1} \\
&= \frac{3}{32B_f^2 + 8n_X\mu\gamma} \\
&< \frac{1}{8B_f^2 + 2n_X\mu\gamma} \\
&< \frac{1}{8B_f^2}
\end{aligned} \tag{144}$$

so that,

$$\begin{aligned}
\frac{1}{2\alpha_w} &> 4\alpha_\theta B_f^2 \\
\implies \frac{1}{2\alpha_w} &> 2\beta_1 B_f^2
\end{aligned} \tag{145}$$

Identities (143) and (145) make sure that the third and the fourth terms on the left hand side of (141) are either zero or positive, so that they can be ignored, resulting in:

$$\begin{aligned}
&\left(\frac{1}{2\alpha_\theta} + \mu\right)\mathsf{E}\left\{\|\theta^{(k)} - \theta^*\|^2\big|\mathcal{F}_k\right\} + \left(\frac{1}{2\alpha_w} + \gamma\right)\mathsf{E}\{\|w^{(k)} - w^*\|^2 \mid \mathcal{F}_k\} \\
&\leq \frac{1}{2\alpha_\theta}\|\theta^{(k-1)} - \theta^*\|^2 + \left(\frac{1}{2\alpha_w} + \gamma - \frac{\gamma}{n_X} + \frac{2\beta_1 B_f^2}{n_X}\right)\|w^{(k-1)} - w^*\|^2
\end{aligned} \tag{146}$$

We further look at the coefficients of other error terms in (146). To this end, we have:

$$\begin{aligned}
\frac{1}{2\alpha_\theta\mu} &= \frac{16B_f^2 + 4n_X\mu\gamma}{2\mu\gamma} \\
&= \frac{8B_f^2}{\mu\gamma} + 2n_X \\
&= 2\kappa + 2n_X
\end{aligned} \tag{147}$$

and therefore, letting $r_P$ to denote the ratio of the coefficients of $\|\theta^{(k-1)} - \theta^*\|^2$ and $\mathsf{E}\left\{\|\theta^{(k)} - \theta^*\|^2\big|\mathcal{F}_k\right\}$, we have:

$$\begin{aligned}
r_P &:= \left(\frac{1}{2\alpha_\theta}\right)\Big/\left(\frac{1}{2\alpha_\theta} + \mu\right) \\
&= 1 - \frac{2\alpha_\theta\mu}{1 + 2\alpha_\theta\mu}, \\
&= 1 - \frac{1}{1 + \frac{1}{2\alpha_\theta\mu}} \\
&= 1 - \frac{1}{1 + 2\kappa + 2n_X}
\end{aligned} \tag{148}$$

Also, for the dual terms we have $r_D$ denoting the ratio of the coefficients of $\|w^{(k-1)} - w^*\|^2$ and $\mathsf{E}\{\|w^{(k)} - w^*\|^2 \mid \mathcal{F}_k\}$, so that:

$$
\begin{aligned}
r_D &:= \left( \frac{1}{2\alpha_w} + \gamma - \frac{\gamma}{n_X} + \frac{2\beta_1 B_f^2}{n_X} \right) \Big/ \left( \frac{1}{2\alpha_w} + \gamma \right) \\
&= 1 - \frac{1}{n_X} \frac{2\alpha_w\gamma - 8\alpha_w\alpha_\theta B_f^2}{1 + 2\alpha_w\gamma} \\
&\stackrel{(a)}{\leq} 1 - \frac{1}{n_X} \frac{2\alpha_w\gamma - 1}{2\alpha_w\gamma + 1} \\
&\stackrel{(b)}{=} 1 - \frac{1}{n_X} \frac{2n_X}{4n_X + 2\kappa + 2} \\
&= 1 - \frac{1}{2n_X + \kappa + 1}
\end{aligned}
\tag{149}
$$

where step (a) follows from (144), and step (b) follows from (150) below:

$$
\begin{aligned}
\frac{2\alpha_w\gamma - 1}{2\alpha_w\gamma + 1} &= \frac{\frac{3n_X + \kappa + 1}{n_X + \kappa + 1} - 1}{\frac{3n_X + \kappa + 1}{n_X + \kappa + 1} + 1} \\
&= \frac{3n_X + \kappa + 1 - n_X - \kappa - 1}{3n_X + \kappa + 1 + n_X + \kappa + 1} \\
&= \frac{2n_X}{4n_X + 2\kappa + 2}
\end{aligned}
\tag{150}
$$

From (148), the above bound (149) on $r_D$ implies $r_D \leq r_P$.

Recalling the definition of $\Delta^{(k)}$, the Lyapunov function defined in Theorem F.1; Substituting for the step-size values in (142), we have:

$$
\begin{aligned}
\Delta^{(k)} &= \left( \frac{1}{2\alpha_\theta} + \mu \right) \mathsf{E}\{\|\theta^{(k)} - \theta^*\|^2 \big| \mathcal{F}_k\} + \left( \frac{1}{2\alpha_w} + \gamma \right) \mathsf{E}\{\|w^{(k)} - w^*\|^2 \big| \mathcal{F}_k\} \\
&= \left( \mu(2\kappa + 2n_X + 1) \right) \mathsf{E}\{\|\theta^{(k)} - \theta^*\|^2 \big| \mathcal{F}_k\} + \left( \gamma\left( \frac{n_X + \kappa + 1}{3n_X + \kappa + 1} + 1 \right) \right) \mathsf{E}\{\|w^{(k)} - w^*\|^2 \big| \mathcal{F}_k\} \\
&= \left( \mu(2\kappa + 2n_X + 1) \right) \mathsf{E}\{\|\theta^{(k)} - \theta^*\|^2 \big| \mathcal{F}_k\} + \left( \gamma\left( \frac{4n_X + 2\kappa + 2}{3n_X + \kappa + 1} \right) \right) \mathsf{E}\{\|w^{(k)} - w^*\|^2 \big| \mathcal{F}_k\}
\end{aligned}
$$

Based on (148) and (149), and the fact that $r_D \leq r_P$, the inequality (141) then implies:

$$
\Delta^{(k)} \leq r\Delta^{(k-1)}
\tag{151}
$$

where,

$$
r = r_P = 1 - 1/(1 + 2\kappa + 2n_X).
\tag{152}
$$

The bound (151) implies that after $k$ iterations, the error $\Delta^{(k)}$ satisfies:

$$
\Delta^{(k)} \leq r^k \Delta^{(0)}
$$

Therefore, for $\Delta^{(k)} < \epsilon$, it suffices to have $r^k\Delta^{(0)} < \epsilon$, so that the number of iterations $k$ satisfies:

$$
\begin{aligned}
& k \ln r \leq \ln\left( \frac{\epsilon}{\Delta^{(0)}} \right) \\
\implies & k \ln\left( 1 - 1/(1 + 2\kappa + 2n_X) \right) \leq \ln\left( \frac{\epsilon}{\Delta^{(0)}} \right) \\
\implies & -k/(1 + 2\kappa + 2n_X) \leq \ln\left( \frac{\epsilon}{\Delta^{(0)}} \right) \\
\implies & k \geq \left( 1 + 2\kappa + 2n_X \right) \ln\left( \frac{\Delta^{(0)}}{\epsilon} \right)
\end{aligned}
\tag{153}
$$

where we have used $-\ln(1 - x) \geq x$. (153) implies the final complexity result of Theorem F.1. $\quad\square$