[Reviews · NeurIPS 2019]

Reviewer 1



This paper proposes a new method for empirical composition optimization problems to which the vanilla SGD is not applicable because of a finite-sum structure inside non-linear loss functions. The method is a type of primal-dual methods with variance reduction for saddle-point problems which is a reformulation of the original problem. In a theoretical analysis part, a linear convergence rate of the method is provided under the strong convexity. In experiments, the superior performance of the method is verified empirically over competitors on portfolio management optimization problems. Clarity: The paper is clear and well written. Quality: The work is of good quality and is technically sound. Originality and significance: The problem (1) treated in this paper is important and contains several applications as mentioned in the paper. However, there seems to be no method that can converge at the linear rate for this problem. As for sub-problems (2), SCGD with SVRG [8] exhibits the linear convergence, but there are some important machine learning tasks not to be covered by (2) as explained in the paper. In addition, it is confirmed in experiments that the proposed method can significantly outperform existing methods including SCGD with SVRG. Hence, this paper makes certain contributions to both theorists and practitioners. If the authors can show a theoretical advantage of the proposed method over SCGD with SVRG [8] on the problem (2) besides the empirical performance, it will make the paper stronger. Minor comments: A regularization term $g$ should be added to equation (10). ----- I have read the author's response and I keep the score.

Reviewer 2



The author response addressed my doubts and questions to satisfactory and hence I raised my score. ------------------------------------------------------------------------------------ The motivation of this paper is good, while the algorithmic contribution is also solid and seems to have wide applications in many popular fields of machine learning. However, I feel the paper being weak in several aspects: Firstly, the discussion of the theoretical results are highly lacking: how does the theoretical convergence result compare to the related state-of-the-art algorithms such as the ones in the cited papers [1, 5, 8, 9]? How realistic is the bounded gradient assumption B_f ? (standard SVRG-type methods does not require bounded gradient). In addition, the algorithmic idea of combining the stochastic variance-reduced gradient and primal-dual reformulation is not brand new [1, 5], but seems that the discussion of the relevance of these work is lacking. The experiments are limited only for portfolio management optimization with very small datasets (d = 25, n < 10000), which is a bit frustrating in my sight. I would expect much larger-scale experiments, and ideally with some more applications such as policy evaluation [5] in reinforcement learning, which the proposed algorithm can be also applied. I wish also to see the experimental comparison of the very recent C-SAGA algorithm (Junyu Zhang and Lin Xiao. A composite randomized incremental gradient method. ICML 2019). In short, I think the current version of this paper can be massively improved and wish to encourage the authors to continue working on this.

Reviewer 3



The authors have satisfactorily answered my concerns and I am happy to raise my score. The complexity comparison table is interesting and should be included in the final version. ========= This paper studies a variance reduced primal dual gradient algorithm for solving strongly convex composite optimization problems (with a finite sum structure). The algorithm is shown to enjoy a linear rate of convergence, both theoretically and empirically, and the variance reduction structure has allowed for efficient implementation. The paper also provides a few convincing experiments on real dataset compared to state-of-the-art algorithms. Overall, the paper has tackled a challenging problem with an efficient algorithm and the proof appears to be correct. Here are a few comments from the reviewer: 1. The complexity bound in Theorem 1,2 The theoretical convergence rate of the algorithms grow with the order of O(n_X (n_Y + kappa) log(1/eps)) - since both n_X, n_Y are large, it seems to be undesirable. Particularly, since the epoch size M has to be chosen at the same order as O(n_X), it seems that after 1 epoch, the algorithm would only improve its optimality of the order of (1 - O(1/n_Y)). May I know if this is the case? In light of the above comparison, it seems that the algorithm in [8] (that is also compared in the paper) has a lower theoretical complexity for large n_Y,n_X, i.e., the latter only has a complexity of O( (n_X+n_Y+kappa^3) log(1/eps) ). Even though the proposed algorithm is demonstrated to be faster from the experiments, explaining such gap from an analytical lens is also important. 2. Strong Convexity in the Risk Adverse learning Example The risk adverse learning is given as an example in the paper and the numerical experiments. Yet it seems that the problem itself does not satisfy Assumption 4.1 required in the analysis. From (4) and the discussion that follows, we know that (4) is a special case of (2), where the latter is a special case of (1) with *g(theta)=0*. However, Assumption 4.1 requires g(theta) to be strongly convex, which is not the case here. 3. Related Reference The linear convergence rate result proven in this paper seems to be related to: S. Du, W. Hu, "Linear Convergence of the Primal-Dual Gradient Method for Convex-Concave Saddle Point Problems without Strong Convexity", AISTATS 2019, which studies the linear convergence of a variance reduced primal dual algorithm with only a non-strongly-convex + strongly-concave structure, whose primal-dual variables are linearly coupled. This is similar to a special case for the setting of (8), where (8) has a nonlinear coupling.

[Author Response · NeurIPS 2019]

We thank all the reviewers for their constructive feedback. Our revision will incorporate all the points detailed below.

**R#1: Theoretical advantage over comp-SCGD [8].** In the table below, we compare our bound to [8], which shows
that we have better dependency on $\kappa$ (see our reply to reviewer #6 for how to derive the complexity in special cases).

**R#2: Bounded gradient assumption.** Bounded gradient is commonly assumed for the interior function $f_\theta$ in existing
compositional optimization works [8, 9, 17, 18] and C-SAGA. The key intuition can be seen from (8): the dual gradients,
which are linear combinations of $f_\theta$, would be Lipschitz if $f_\theta$ is Lipschitz (guaranteed by bounded $f_\theta'$).
**R#2: Novelty of SVRG + primal-dual w.r.t. [1,5].** We have discussed [1,5] in Sec. 6 (see lines 249-250, 252-253).
The algorithms in [1] are not for composition optimization but for general saddle-point problems. Applying their
algorithms to our saddle-point problem (8) will fail to capture its inherent special structures (e.g., dual decomposition).
In contrast, our algorithm fully exploits the dual decomposition and coordinate ascent structures in (8) to develop a
much better control variate compared to regular SVRG (see our discussion in Sec. 3.3.). Our new experiments (see
Fig-(a) below) shows the superiority of our algorithms over [1] (*Saddle-SVRG*). That is, our work is not a simple
combination of SVRG with primal-dual formulation but a novel algorithm that is carefully designed to solve (1). In
addition, we believe it is the first work to solve the more general compositional optimization (1) along this direction.
Regarding the work [5], it only considers MSPBE cost in policy evaluation, which can be viewed as a special case of (2)
with quadratic $\phi_i$ and linear $f_\theta$. The table below shows that our method has better dependency on $\kappa$ in this special case.
**R#2: Compare to other composition optimization methods.** The following table lists their complexity bounds (in
number of oracle calls) using our notation for different problem settings. First, ASCVRG [9] does not assume strong
convexity and thus cannot achieve linear convergence rate. (Its dependency on $\kappa$ has been dropped as it was not reported
in [9].) Second, none of the algorithms consider the general problem (1) as we did. Instead, they consider its special
case (2). Even in the special cases, our algorithm still have better (or comparable) complexity bound than other methods.
Note that our complexity for problem (2) is lower than (1) due to its structure (see our reply to reviewer #6).

| Methods | SVRPDA-I (Ours) | Comp-SVRG [8] | C-SAGA (minibatch: $\alpha=\frac{2}{3}$; batch-size=1: $\alpha=1$) | MSPBE-SVRG/SAGA [5] | ASCVRG [9] |
|---|---|---|---|---|---|
| General: problem (1) | $(n_X n_Y + n_X \kappa)\ln\frac{1}{\epsilon}$ | | | | |
| Special: problem (2) | $(n_X+n_Y+n_X\kappa)\ln\frac{1}{\epsilon}$ | $(n_X+n_Y+\kappa^3)\ln\frac{1}{\epsilon}$ | $(n_X+n_Y+(n_X+n_Y)^\alpha\kappa)\ln\frac{1}{\epsilon}$ | | $(n_X+n_Y)\ln\frac{1}{\epsilon}+\frac{1}{\epsilon^3}$ |
| Special: (2) & $n_X=1$ | $(n_Y+\kappa)\ln\frac{1}{\epsilon}$ | $(n_Y+\kappa^3)\ln\frac{1}{\epsilon}$ | $(n_Y+n_Y^\alpha\kappa)\ln\frac{1}{\epsilon}$ | $(n_Y+\kappa^2)\ln\frac{1}{\epsilon}$ | $n_Y\ln\frac{1}{\epsilon}+\frac{1}{\epsilon^3}$ |

**R#2: Large-scale experiments/more applications.** First, risk-averse learning is a real-world benchmark that was
widely used by existing compositional optimization works [8, 9, 17, 18]. As per your request, we add a policy evaluation
task with state-space size $S = 10$ (used in the C-SAGA paper) and $S = 10^4$ (10 times larger than the biggest one
experimented in C-SAGA). All the new results (figures below) further demonstrate the superiority of our algorithm
(even over the most recent C-SAGA work), where each baseline has been tuned to its best performance. Fig.(b) only
compares to C-SAGA because it was shown in their paper that it achieves the best performance on this task.

(a) Risk-averse learning (Europe INV dataset)    (b) MDP policy evaluation [same as C-SAGA paper (Zhang & Xiao, 2019) & x10 larger state space]

**R#2: Compare to C-SAGA.** We were unaware of C-SAGA paper because it appeared after the NeurIPS deadline.
However, we will be happy to include the comparison (Fig.(a)-(b) and the table above). C-SAGA mainly considers the
special case of $n_X = 1$ and extends it to (2). It seems that in the case (2), the comparisons would depend on $n_X$ and
$n_Y$. However, since we have derived bounds for a more general problem (1) and then specialize it, we would like to
highlight the technical difficulties that may arise in deriving our bounds, and the fact that it may not specialize to the best
bound for the simpler problem. We observe in our experiments that despite having much smaller memory requirement,
SVRPDA-II has a comparable/better performance with C-SAGA, while SVRPDA-I clearly beats C-SAGA.

**R#6: Compare Theorems 1 & 2 to [8]'s bound.** Our complexity bound $O((n_X n_Y + n_X \kappa) \ln(\frac{1}{\epsilon}))$ is for solving
the general problem (1), while [8] is only considering its special case (2). When applying our algorithm to (2), our
complexity could be reduced to $O((n_X + n_Y + n_X\kappa) \ln(\frac{1}{\epsilon}))$ (see the above table). This is because the complexity
for evaluating the batch quantities in (11) (see Algorithm 1) can be reduced from $O(n_X n_Y)$ in the general case (1)
to $O(n_X + n_Y)$ in the special case (2). To see this, note that $f_\theta$ and $n_{Y_i} = n_Y$ become independent of $i$ in (2) and
(11). This means that we can factor $U_0$ in (11) as $U_0 = \frac{1}{n_X n_Y} \sum_{j=0}^{n_Y-1} f_{\hat\theta}'(y_j) \sum_{i=0}^{n_X} w_i^{(0)}$, where the two sums can be
evaluated independently with complexity $O(n_Y)$ and $O(n_X)$, respectively. The other two quantities in (11) need only
$O(n_Y)$ due to their independence of $i$. In this case, our bound is better than that in [8] since $\kappa^2 > n_X$ generally holds.
**R#6: Strong convexity & reference by (Du & Hu).** We indeed added a small ($10^{-6}$) $L_2$-regularizaton in our
experiments. But we did observe that the algorithm can still converge linearly even without the strongly convex
regularization, similar to what is observed in (Du & Hu) for linear coupling. We will clarify this and add the reference.

[Meta-Review · NeurIPS 2019]

The authors propose a stochastic algorithm for generalized empirical composition problems of the form in eq. 1. The paper provides linear convergence rates for a problem (eq 1), which is more general than the problems (eq. 2) studied by the related work [8-9]. For the special case of the problem in eq. 2, the initial submission does not compare rates to [8-9]. Reviewers 2 and 6 had raised this concern and the author response provides convincing comparison. Addition of these comparisons to the revision is highly recommended.